# Tracheal tuft cells release ATP and link innate to adaptive immunity in pneumonia

Noran Abdel Wadood [1,14], Monika I. Hollenhorst[1,2,14], Mohamed Ibrahem Elhawy [1,14], Na Zhao [1], Clara Englisch[1], Saskia B. Evers[1], Mahana Sabachvili [1], Stephan Maxeiner [1,2], Amanda Wyatt [3], Christian Herr [4], Ann-Kathrin Burkhart [1,2,5], Elmar Krause [6], Daniela Yildiz[2,7], Anja Beckmann[1], Soumya Kusumakshi[3], Dieter Riethmacher [8], Markus Bischoff [9], Sandra Iden [1,2,5], Sören L. Becker [9], Brendan J. Canning [10], Veit Flockerzi [7], Thomas Gudermann [11,12], Vladimir Chubanov [11], Robert Bals[2,4,13], Carola Meier[1], Ulrich Boehm [2,3] & Gabriela Krasteva-Christ [1,2] ✉

Tracheal tuft cells shape immune responses in the airways. While some of these effects have been attributed to differential release of either acetylcholine, leukotriene C4 and/or interleukin-25 depending on the activating stimuli, tuft cell-dependent mechanisms underlying the recruitment and activation of immune cells are incompletely understood. Here we show that *Pseudomonas aeruginosa* infection activates mouse tuft cells, which release ATP via pannexin 1 channels. Taste signaling through the Trpm5 channel is essential for bacterial tuft cell activation and ATP release. We demonstrate that activated tuft cells recruit dendritic cells to the trachea and lung. ATP released by tuft cells initiates dendritic cell activation, phagocytosis and migration. Tuft cell stimulation also involves an adaptive immune response through recruitment of IL-17A secreting T helper cells. Collectively, the results provide a molecular framework defining tuft cell dependent regulation of both innate and adaptive immune responses in the airways to combat bacterial infection.

Airway tuft cells, including the brush cells in the trachea and the solitary chemosensory cells (SCCs) in the nasal mucosa elicit innate immune responses in the airways upon activation[1]. The tracheal tuft cells and nasal SCCs are considered chemosensory based on their responsiveness to bitter taste receptor ligands, including bacterial quorum-sensing molecules and formyl peptides[2–6]. Acute tuft cell stimulation with bacteria-derived molecules triggers cholinergic signaling dependent on activation of the transient receptor potential cation channel, subfamily M, member 5 channel (Trpm5), a member of the canonical bitter taste signaling cascade. Tuft cells express choline acetyltransferase (ChAT) for acetylcholine (ACh) synthesis[7]. The released ACh exerts paracrine effects on neighboring cells and sensory nerve fibers terminating adjacent to tuft cells, thus initiating

local innate immune responses and protective reflexes[3–6]. Tuft cell-mediated release of calcitonin gene-related peptide (CGRP) and Substance P from adjacent nerve endings also results in protective neurogenic inflammation characterized by plasma extravasation and neutrophil recruitment[4]. Tracheal tuft cells also modulate mucociliary clearance[5,6,8,9].

Mice lacking a functional taste signaling cascade in tuft cells manifest a more severe disease progression and a reduced survival rate upon acute *Pseudomonas aeruginosa* infection[4]. This may relate to a diminished tuft cell-dependent recruitment of neutrophils, monocytes, natural killer cells and alveolar macrophages in the early infection state (i.e., 4 h after bacterial contact). The impact of tuft cell activation on long-term protective responses in infection is unclear. It

is possible that tuft cell-dependent immune responses depend upon purinergic signaling[10]. In support of this hypothesis, tuft cells possess similar signaling mechanisms with type II taste cells including expression of a functional bitter taste signaling cascade and release ACh, which acts on nearby nerve fibers[4,5,7,11,12]. In taste buds chemosensory cells activate the bitter taste signaling cascade triggering the release of ATP through calcium homeostasis modulator 1/3 (Calhm1/3) channels and eventually binding to purinergic receptors on sensory nerves that innervate type II taste cells, thus conveying taste sensation to the nervous system[11,13–15].

While ATP plays a clear role in sensory signaling in the taste buds, no direct immunological function has been attributed to its release in this context. Interestingly, SCCs of the nasal mucosa respond to extracellular ATP. Stimulation of P2ry2 receptors in SCCs promotes a strongly polarized type 2 immune response[16,17], driven by the release of leukotriene C4 (LTC4) and IL-25 but not ACh[17]. Tuft cells are the source of IL-25 in tracheal epithelium of mice and upper airways of humans, with nasal SCCs in mice exhibiting higher *Il25* transcription levels than tracheal tuft cells[16,18,19]. These findings suggest that tuft cells fine-tune innate immunity by differential release of signaling molecules depending on the context, such as during infection or exposure to aeroallergens.

In this study, we investigated whether tracheal epithelial tuft cells release ATP upon activation of the bitter taste signaling cascade. Given the similarities between chemosensory cells in the taste bud and tuft cells, we hypothesized that ATP release from tuft cells may also be involved in tuft cell-mediated immune responses. Utilizing various in vivo and ex vivo models with particular emphasis on immune cell recruitment and activation, we demonstrate that ATP released by tuft cells plays a critical role in bacterial infections, highlighting functions of tuft cells in bridging innate and adaptive immune processes.

## Results

### Tracheal tuft cells release ATP

The Trpm5 channel in airway tuft cells plays an essential role in mediating innate immune responses[1,4,5]. Therefore, we assessed Trpm5 activity in primary tuft cells using patch-clamp techniques. We measured whole-cell currents in freshly isolated tuft cells from tissues of *Trpm5*[+/+]-*ChAT*-eGFP and *Trpm5*[−/−]-*ChAT*-eGFP mice. In both mouse strains, tuft cells can be identified by their green fluorescence. In the presence of an increased intracellular Ca[2+] concentration, we observed characteristic Trpm5 currents in all *Trpm5*[+/+] tuft cells examined. In line with previous studies[20,21], Trpm5 currents developed within ~3 min and afterwards decayed to baseline (Figs. 1A, Supplementary Fig. 1A). Of note, Trpm5 currents were not present in *Trpm5*[−/−] tuft cells (Fig. 1B, C) or *Trpm5*[+/+] tuft cells exposed to the Trpm5 antagonist triphenylphosphine oxide[22] (TPPO, Supplementary Fig. 1B).

Next, we asked whether stimulation with bitter compounds leads to an ATP release in the trachea. To specifically test whether tuft cells release ATP we used ATP sensor cells (Fig. 1D–H, Supplementary Fig. 1C–E), which react to ATP with an EC$_{50}$ of 83.9 μM (Supplementary Fig. 2A). We added the ATP sensor cells to epithelial cells isolated from *ChAT*-eGFP mice (Fig. 1D). We then recorded changes in fluorescence levels in the ATP sensor cells while performing patch-clamp experiments on tuft cells in the whole-cell mode. Upon Trpm5 channel activation in tuft cells with a high free Ca[2+] concentration in the intracellular solution, the fluorescence intensity in the ATP sensor cells increased simultaneously with a rise of Trpm5 currents in tuft cells, which was not the case after activation of tuft cells from *Trpm5*[−/−] mice (Fig. 1E–H). These data demonstrate that tuft cells release ATP upon Trpm5 channel activation. An ATP release from tuft cells was also detected by ELISA and in experiments with supernatants prepared from tracheae from wild-type mice treated with 1 mM denatonium—a taste receptor agonist specifically acting on tuft cells[4], but not from Trpm5-deficient (*Trpm5*[−/−]) mice (Fig. 1I). ATP release from explanted

denatonium-treated wild-type tracheae was also verified by the response of ATP sensor cells treated with these supernatants (Supplementary Fig. 2B). Tracheal supernatants from CNO-treated (clozapine-N-oxide at 100 μM) *Trpm5*-DREADD mice, characterized by expression of designer receptors exclusively activated by designer drugs (DREADD) in tuft cells, showed increased ATP levels compared to vehicle controls (Fig. 1I).

To address the ATP release mechanism in tuft cells, we studied the role of pannexin 1 (Panx1) channels. Panx1 channels have previously been identified as ATP release channels in various cell types, including taste cells[23–25]. More intriguingly, single-cell RNA-seq analyses from our group and others[5,18,26] have suggested *Panx1* expression in tuft cells[5,18,26]. Supernatants from denatonium- or vehicle-treated tracheae of Panx1-deficient (*Panx1*[−/−]) mice contained similar ATP concentrations (Fig. 1J). *Trpm5*-DTA mouse tracheae lacking tuft cells showed similar ATP levels after denatonium treatment compared to vehicle treated controls (Fig. 1J). In contrast to this, mice deficient of ChAT in tuft cells (*ChAT*[fl/fl]:*Trpm5*[cre]), showed increased ATP levels in tracheal supernatants after stimulation with denatonium compared to vehicle-treatment (Fig. 1J), indicating that the observed ATP release was independent from cholinergic signaling. This mouse model was validated by immunoblot and immunohistochemistry experiments and used throughout our study in order to distinguish between the effects induced by ATP and ACh, specifically to assess the potential involvement of tuft cell released ACh in the observed ATP release (Supplementary Fig. 3). These validations confirmed the absence of ChAT protein in tracheal tuft cells from *ChAT*[fl/fl]:*Trpm5*[cre] mice. Consistently, no ACh was detected in supernatants from *ChAT*[fl/fl]:*Trpm5*[cre] tracheae stimulated with denatonium (1 mM), while ACh was robustly detected in the supernatants collected from wild-type mice (Supplementary Fig. 3C). Furthermore, in *Panx1*[−/−] mice, denatonium-induced ACh release was not altered, indicating that ACh is not released via Panx1 (Supplementary Fig. 3D).

Because Panx1 is a potential contributor to ATP release from airway epithelial cells[27], we next investigated Panx1 reporter gene expression in the tracheal epithelium of *Panx1*[−/−] and *Panx1*[−/−]/*2*[−/−] reporter mice[28]. LacZ reporter signals co-localized with the tuft cell marker Trpm5 (Fig. 1K–O). We also detected LacZ signals in epithelial cells adjacent to tuft cells indicating that Panx1 and pannexin 2 (Panx2) expression is not restricted to tuft cells (Fig. 1K–O). Immunohistochemistry for Panx1 on tracheal tissue sections from *ChAT*-eGFP mice confirmed Panx1 expression in tuft cells (Supplementary Fig. 4A). Panx1 staining was absent in *Panx1*[−/−] mice (Supplementary Figs. 4B and C). LacZ-positive signals were also observed in other epithelial cells, sometimes clustering around tuft cells (Supplementary Figs. 4D–H). Quantification of tuft cells expressing LacZ in *Panx1*[−/−] tracheae revealed that around 40% of the Trpm5[+] tuft cells expressed Panx1 (Supplementary Fig. 4F). Taken together our results show a Trpm5-dependent ATP release from tuft cells via Panx1 channels expressed in a subpopulation of tuft cells.

### Tuft cells are activated by denatonium and express functional pannexins

To investigate Panx1 function in tuft cells, we performed whole-cell patch-clamp experiments using primary tuft cells isolated from *ChAT*-eGFP mice. Denatonium stimulation (1 mM) of wild-type (*Trpm5*[+/+]) tuft cells induced a large current with outwardly rectifying properties, which was inhibited in the presence of the Trpm5 antagonist TPPO (100 μM, Fig. 2A and C). The I/V relationship of the denatonium-induced current was similar to the I/V relationship observed in Trpm5-expressing HEK293 cells activated by Ca[2+][20,21]. Denatonium had no effect on tuft cell whole-cell currents in *Trpm5*[−/−]-*ChAT*-eGFP mice (Fig. 2B and C), confirming that the denatonium-induced currents in tuft cells were indeed due to Trpm5 activation.

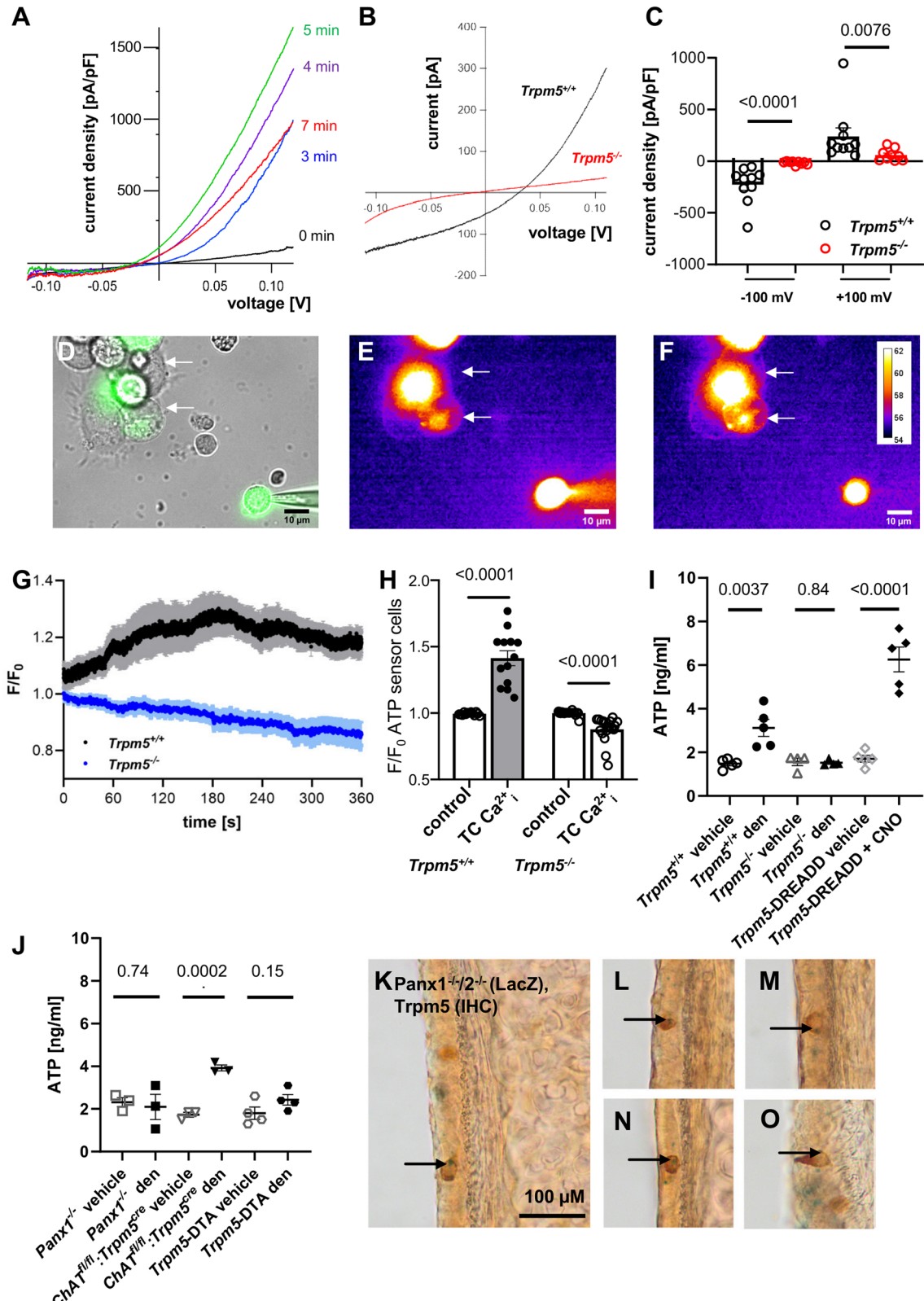

We then asked, whether tuft cells express functional pannexins and connexins, because Panx1 and several connexins including connexins 26, 30, 32, and 43 are permeable for ATP[29]. Since high $K^+$ concentrations activate Panx1 currents[30–32], we first studied the tuft cell response to KCl. KCl-induced currents were strongly diminished in the presence of the Panx1 antagonists 10 μM carbenoxolone and 300 μM probenecid in both wild-type ($Trpm5^{+/+}$) and $Trpm5^{-/-}$ mice (Fig. 2D

and E). In contrast, no significant difference was detected in KCl-induced currents between $Trpm5^{+/+}$ and $Trpm5^{-/-}$ mice (Fig. 2F). We next asked whether Trpm5 channel activation by $Ca^{2+}$ would affect the Panx1 currents. We recorded ramp currents in both wild-type ($Trpm5^{+/+}$) and $Trpm5^{-/-}$ tuft cells using either a low or a high $Ca^{2+}$ concentration in the intracellular solutions and applied carbenoxolone or probenecid (Fig. 2G). The $Ca^{2+}$-induced currents were

**Fig. 1 | Tuft cells release ATP and express pannexin 1. A** Representative I-V relationships of TRPM5 current amplitudes normalized to cell size (pA/pF) measured in a tuft cell at indicated time points after breaking the cell. **B** Representative current-voltage (I-V) relationships of whole-cell currents measured in a tracheal tuft cell from a *Trpm5*[+/+] or a *Trpm5*[−/−] mouse in the presence of 1 μM intracellular Ca²⁺. **C** Statistical analysis (two-tailed Mann-Whitney test) of currents illustrated in (**B**). Current amplitude densities measured at −100 mV and +100 mV in *Trpm5*[−/−] tuft cells (*n* = 9 cells) and *Trpm5*[+/+] tuft cells (*n* = 10 cells) are shown. **D** The bright field of a patched tuft cell (green, GFP + ) and ATP sensor cells (arrow) shown in E-F. **E**−**F** Representative images from a recording of the membrane fluorescence of two ATP sensor cells (arrow) before (**E**) and after (**F**) application of Ca²⁺ containing intracellular solution into a tuft cell through the patch pipette (for *n* = 5 mice/5 experiments). **G** The fluorescence intensity of all ATP sensor cells over the whole-cell recording of *Trpm5*[+/+] (black, *n* = 11 cells/5 mice/5 experiments) or *Trpm5*[−/−] tuft cells (blue, *n* = 9 cells/4 mice/4 experiments). **H** Statistical analyses of maximal fluorescence intensity of all ATP sensor cells responding to tuft cell stimulation in *Trpm5*[+/+] mice (*n* = 13 cells/5 mice/5 experiments) and *Trpm5*[−/−] mice (*n* = 19 cells/4 mice/4 experiments). Control = baseline fluorescence, tuft cell Ca²⁺ᵢ = fluorescence after tuft cell stimulation with intracellular Ca²⁺; two-tailed paired Student's *t*-test. **I** ATP levels in supernatants from denatonium- and vehicle-treated tracheae of *Trpm5*[+/+] mice (*n* = 5) and *Trpm5*[−/−] mice (*n* = 4), as well as in supernatants from clozapine-N-oxide (CNO, 100 μM, *n* = 5)- and vehicle-treated (*n* = 5) *Trpm5*-DREADD mice. Two-tailed unpaired Student's *t*-test. **J** ATP levels in supernatants from denatonium- and vehicle-treated tracheae from *Panx1*[−/−] (*n* = 3), *ChAT*[fl/fl]:*Trpm5*[cre] (*n* = 3), and *Trpm5*- DTA mice (*n* = 4). Two-tailed unpaired Student's *t*-test. **K**−**O** Immunohistochemistry for Trpm5 combined with LacZ-staining in the tracheal epithelium of *Panx1*[−/−]/2[−/−] mice (*n* = 3). Arrows indicate LacZ-staining (blue) in the nuclei of Trpm5⁺ cells (brown). Source data are provided in the Source Data file. Data in panels (**C, G, H, I** and **J**) are presented as mean ± SEM.

reduced in wild-type (*Trpm5*[+/+]) but not in *Trpm5*[−/−] mice in the presence of carbenoxolone (10 μM) and a low Ca²⁺ concentration (110 nM). The Ca²⁺-induced current was also significantly reduced by probenecid (300 μM) in the presence of the low (110 nM) as well as the high Ca²⁺ concentration (2.4 μM) in *Trpm5*[+/+] but not in *Trpm5*[−/−] mice. ATP released via Panx1 has been reported to activate the purinergic receptor P2rx7, leading to additional K⁺ efflux and further activation of Panx1. Given that single-cell RNA sequencing studies from our group and others have identified low levels of P2rx7 expression in tuft cells[5,18], we next investigated the role of P2rx7 in Trpm5-dependent currents. To assess this, we conducted additional whole-cell patch-clamp experiments, using the specific P2rx7 inhibitor AZ10606120[33]. Following tuft cell stimulation with elevated intracellular Ca²⁺, we found no difference in current density (pA/pF) in the presence or absence of the inhibitor (Fig. 2H). These findings suggest that P2rx7 does not significantly contribute to the Ca²⁺-induced currents in tuft cells, which are required for the Trpm5 activation that drives Panx1-dependent currents.

Additionally, we further investigated ATP release in the tracheal epithelium by recording currents after tuft cell stimulation using whole-mount tracheal preparations. The denatonium-induced transepithelial ion current was significantly reduced in *Panx1*[−/−] as well as in *Panx1*[−/−]*2*[−/−] double-deficient mice, while *Panx2*[−/−] mice showed a response comparable to wild-type controls (Supplementary Figs. 5A, B and C). When inhibiting calcium homeostasis modulator 1 (Calhm1) channels, which are responsible for the ATP release from type II cells in taste buds[14], with ruthenium red (20 μM apical and basolateral), the denatonium-induced effect remained unchanged (Supplementary Fig. 5D). In the presence of probenecid (200 μM, apical and basolateral) the denatonium-induced effect was significantly decreased in *Trpm5*[+/+] mice (Supplementary Fig. 5E). Application of suramin (100 μM, basolateral), a non-selective purinergic receptor antagonist[34], significantly reduced the denatonium-induced effect (Supplementary Fig. 5F), suggesting that the released ATP acts in a paracrine way on neighboring airway epithelial cells. Taken together, these data demonstrate that ATP is released from tuft cells via Panx1 channels.

### Chemogenetic tuft cell activation recruits neutrophils and dendritic cells

To study the tuft cell-induced effects on innate immunity longitudinally, we first established a model to specifically activate tuft cells in vivo. First, we performed Ca²⁺-imaging experiments in freshly isolated primary tuft cells from *Trpm5*-DREADD mice to confirm their specific activation by CNO. Intracellular Ca²⁺ levels increased in tuft cells in *Trpm5*-DREADD mice upon CNO application (Supplementary Figs. 6A, B). Next, we performed in vivo stimulation of tracheal tuft cells in *Trpm5*-DREADD mice. Using FACS analyses we detected higher numbers of neutrophils in bronchoalveolar lavage fluid (BALF) and in the trachea 30 min after in vivo tracheal application of CNO (Fig. 3A

and B, gating strategies are represented in Supplementary Fig. 7), consistent with our previous findings obtained 4 h after infection of mice with *P. aeruginosa*[4]. No difference in the blood cell composition was observed in blood samples of CNO-treated controls (Fig. 3C). Monocyte and interstitial macrophage numbers, but not alveolar macrophage numbers increased in BALF from CNO-treated *Trpm5*-DREADD animals (Fig. 3D, E and F). Thus, interstitial rather than alveolar macrophages seem to play a role in the acute tuft cell-induced immune response, in line with a study by Sanchez et al.[35]. Of note, dendritic cells (DCs) were also increased in BALF and lungs after CNO treatment (Fig. 3F and G).

### Tuft cells expand upon denatonium activation

DCs are antigen-presenting cells and provide an important link between the innate and the adaptive immune system. We next investigated the influence of tuft cells on dynamic DC recruitment and quantified tuft cells and DCs at different time points after tuft cell stimulation with denatonium. We observed a significant expansion of tuft cells in *Trpm5*[+/+] mice 24 h, 72 h, and 7 days after in vivo inhalation of 1 mM denatonium, reaching a maximum after 72 h when compared to the vehicle-treated controls (Supplementary Fig. 8A). In contrast, no changes in the tuft cell numbers were observed under the same treatment conditions in *Trpm5*[−/−] mice (Supplementary Fig. 8A). There was no significant difference between vehicle-treated *Trpm5*[+/+] and *Trpm5*[−/−] mice (Supplementary Fig. 8A). We next labeled immune cells (mostly DCs) with an antibody against CD11c on tracheal tissue sections[36]. While CD11c⁺ cell numbers did not differ between vehicle-treated *Trpm5*[+/+] and *Trpm5*[−/−] mice (Supplementary Fig. 8B), they were significantly increased in *Trpm5*[+/+] mice 72 h after denatonium treatment. This increase persisted in *Trpm5*[+/+] mice at 7 days (Supplementary Fig. 8B). In contrast, there were no changes in CD11c⁺ cell numbers in *Trpm5*[−/−] mice after denatonium treatment (Supplementary Fig. 8B). A positive medium linear correlation, with a Pearson's correlation coefficient of 0.4454, was detected between tuft cell and CD11c⁺ cell numbers in the tracheae of *Trpm5*[+/+] mice after denatonium inhalation, whereas no correlation was observed in *Trpm5*[−/−] mice (Pearson's r = 0.0432) (Supplementary Fig. 8C and D). These data indicate that tuft cells are involved in the regulation of DC function.

### Tuft cell activation in *P. aeruginosa* infection recruits dendritic cells in trachea and lung

To investigate the role of tuft cells in immune responses in the post-acute phase of infection or stimulation, we stimulated tuft cells in vivo by an infection with *P. aeruginosa* or with denatonium and analyzed the resulting immune profile in tracheae, lungs, and airway lymph nodes three days later using FACS (gating strategies are presented in Supplementary Fig. 9−12). This time point had been established as being crucial for survival following infection in our previous infection

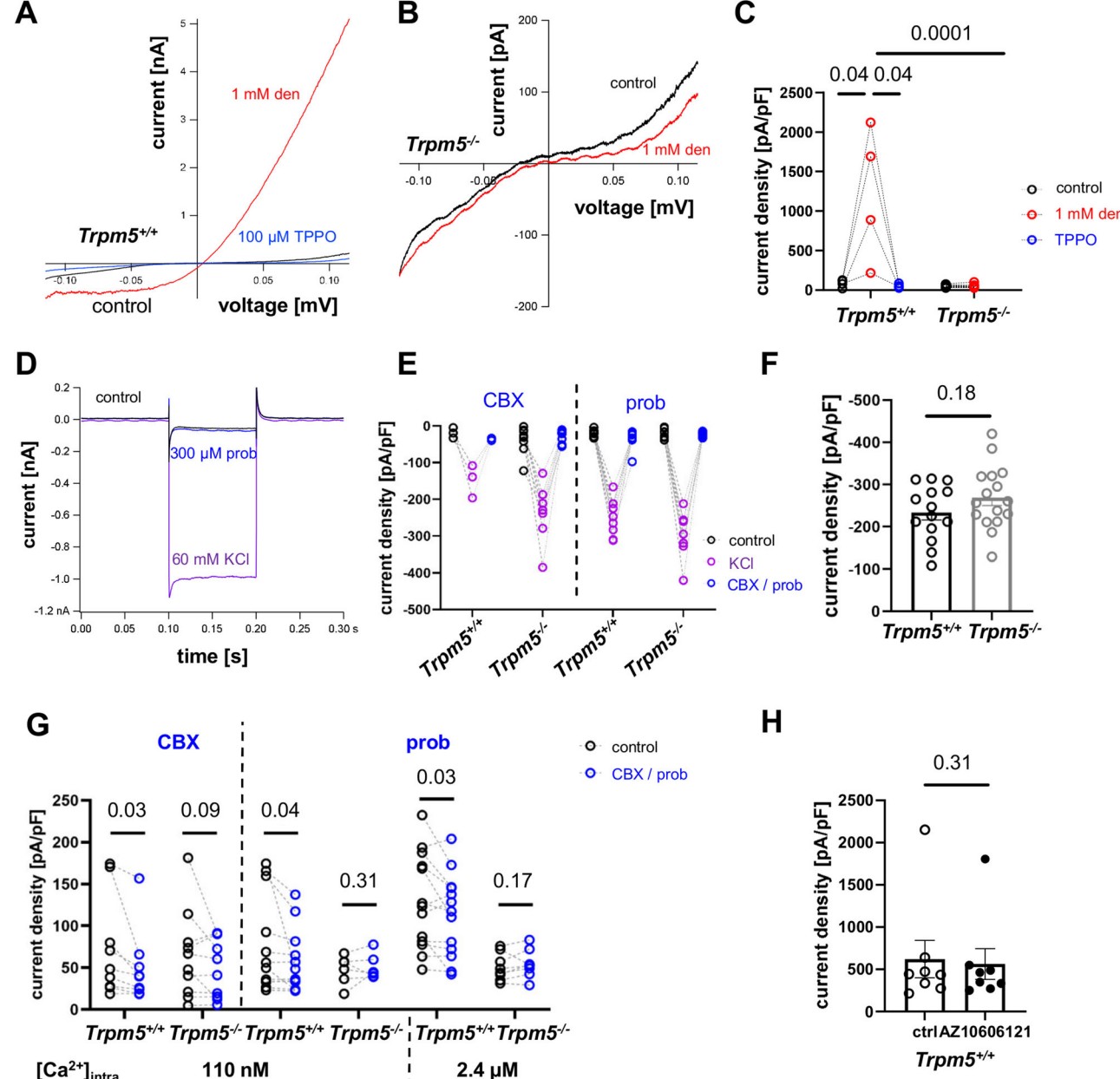

**Fig. 2 | Whole-cell patch-clamp recordings of mouse tracheal tuft cells.**
**A** Application of denatonium (1 mM) in $Trpm5^{+/+}$ tuft cells led to an outwardly rectifying current that was abolished by TPPO (100 μM). **B** Denatonium (1 mM) had no effect on the I/V curve recorded in $Trpm5^{-/-}$ tuft cells. Red = denatonium, black = baseline. **C** Current densities recorded at 100 mV were increased by denatonium in tuft cells of $Trpm5^{+/+}$ mice compared to before treatment and to currents in the presence of 100 μM TPPO ($n = 4$ cells/3 mice, paired Student's $t$-test). Currents at 100 mV of $Trpm5^{-/-}$ tuft cells ($n = 11$ cells/5 mice) perfused with 1 mM denatonium were reduced compared to tuft cells from $Trpm5^{+/+}$ mice (two-tailed unpaired Student's $t$-test). **D** Representative current traces at a holding potential of +30 mV hyper-polarized to −60 mV upon application of 60 mM KCl and addition of 300 μM probenecid (prob). **E** KCl-induced current densities were inhibited by carbenoxolone (CBX, 10 μM) or probenecid (prob, 300 μM) in tuft cells from $Trpm5^{+/+}$ and $Trpm5^{-/-}$ mice. $Trpm5^{+/+}$ CBX: $n = 3$ cells/3mice, prob: $n = 9$ cells/5mice, $Trpm5^{-/-}$

CBX: $n = 8$ cells/3mice, prob: $n = 8$ cells/3mice, two-tailed paired Student's $t$-test. **F** The KCl-induced current densities in $Trpm5^{+/+}$ ($n = 13$ cells/7mice) and $Trpm5^{-/-}$ ($n = 16$ cells/5mice) mice were identical (two-tailed unpaired Student's $t$-test). **G** Analyzed current densities at 100 mV upon a ramp depolarization. Carbenoxolone (CBX, 10 μM)-sensitive currents upon the activation of Trpm5 channels with 110 nM $[Ca^{2+}]_i$ were reduced in $Trpm5^{+/+}$ but not in $Trpm5^{-/-}$ mice. Trpm5 currents at 110 nM $[Ca^{2+}]_i$ and 2.4 μM $[Ca^{2+}]_i$ were sensitive to probenecid (prob, 300 μM) in $Trpm5^{+/+}$ but not in $Trpm5^{-/-}$ mice. CBX: $Trpm5^{+/+}$: $n = 9$ cells/6mice, $Trpm5^{-/-}$: $n = 10$ cells/5mice, prob, 110 nM Ca$^{2+}$: $Trpm5^{+/+}$: $n = 12$ cells/4mice, $Trpm5^{-/-}$: $n = 7$ cells/3mice, prob, 2.4 μM Ca$^{2+}$: $Trpm5^{+/+}$: $n = 13$ cells/4mice, $Trpm5^{-/-}$: $n = 8$ cells/3mice; one-tailed paired Student's $t$-test. **H** The Ca$^{2+}$-induced current densities in tuft cells (ctrl) were not inhibited by the P2x7 receptor inhibitor AZ10606121 (20 μM) ($n = 8$ cells/4 mice/4 experiments). Two-tailed Wilcoxon test. Source data are provided in the Source Data file. Data in panels **F** and **H** are presented as mean ± SEM.

experiments in which $Trpm5^{-/-}$ mice develop a more severe infection[4]. Three days after intratracheal denatonium treatment (1 mM) we observed a significant increase in DC numbers (CD11c$^+$, CD11b$^+$, F4/80$^-$) in tracheae of $Trpm5^{+/+}$ mice that was abolished in $Trpm5^{-/-}$ mice. This effect was also present in mice with deficient ACh synthesis specifically

in tuft cells ($Chat^{fl/fl}:Trpm5^{cre}$) (Fig. 4A), showing that the recruitment of DCs was independent from ACh released by airway tuft cells. Next, we found that denatonium-induced DC recruitment was significantly reduced in $Panx1^{-/-}$ mice when compared to wild-type ($Panx1^{+/+}$) controls (Fig. 4B). DC numbers in wild-type ($Trpm5^{+/+}$) mice

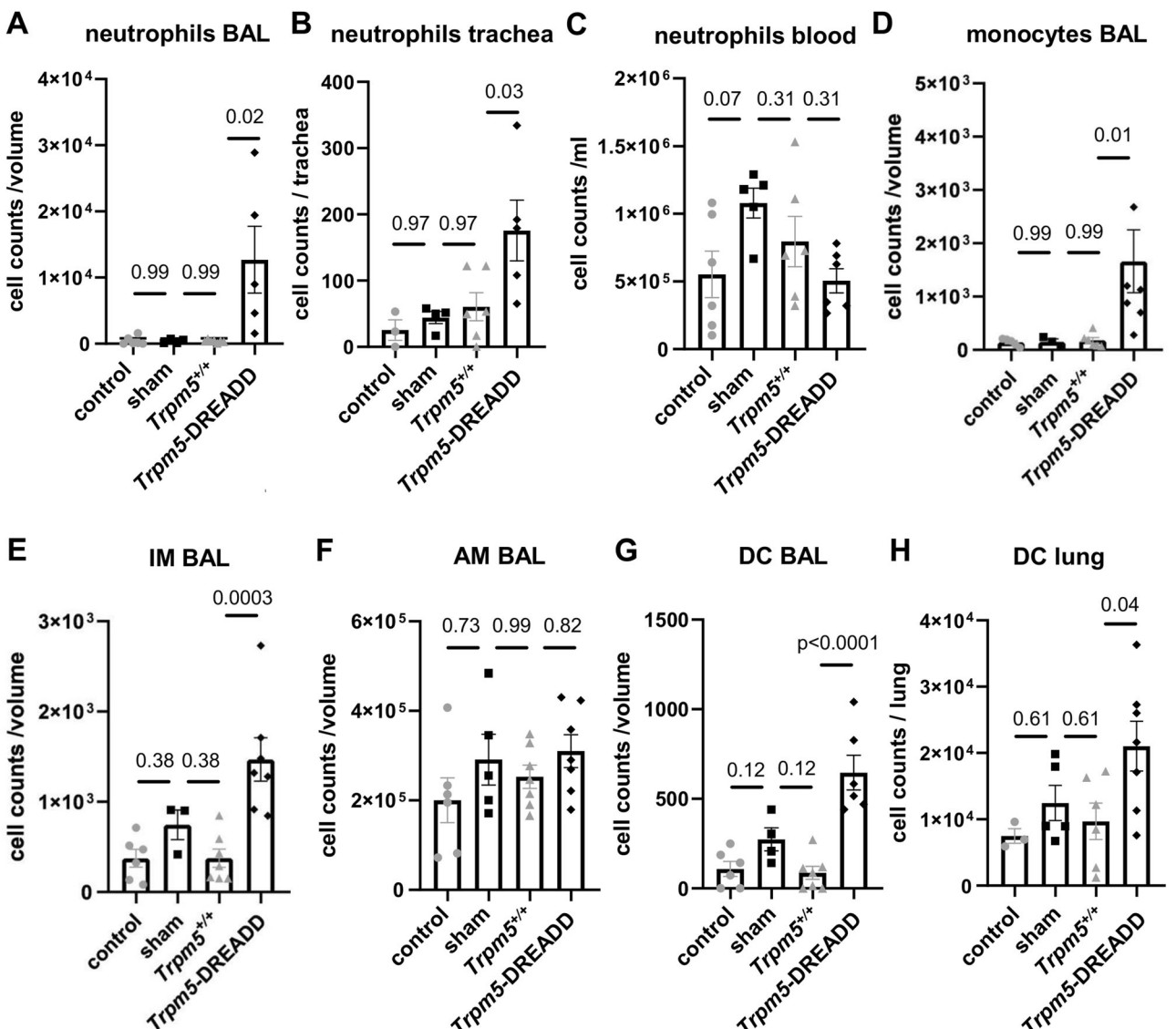

**Fig. 3 | Recruitment of immune cells after acute in vivo tuft cell stimulation.**
FACS analyses. **A** Stimulation of tuft cells intratracheally with 100 µM clozapine-N-oxide (CNO) in *Trpm5*-DREADD mice ($n = 5$) led to a significant increase in neutrophil numbers within bronchoalveolar lavage fluid (BALF, Kurskal-Wallis test) in (**B**) trachea (One-way ANOVA) but not in (**C**) blood (One-way ANOVA) after 30 min, compared to untreated *Trpm5*-DREADD control ($n = 6$), *Trpm5*-DREADD sham treated ($n = 4$) and CNO-treated *Trpm5*$^{+/+}$ mice ($n = 5$). **D** In *Trpm5*-DREADD mice treated with 100 µM CNO ($n = 7$), the numbers of monocytes were significantly increased in BAL samples after 30 min compared to CNO-treated *Trpm5*$^{+/+}$ mice ($n = 7$). sham-treated: $n = 3$, control=untreated: $n = 6$. Kruskal-Wallis test.
**E** Interstitial macrophage (IM) numbers increased in BAL samples of *Trpm5*-DREADD mice treated with 100 µM CNO ($n = 7$) compared to CNO-treated *Trpm5*$^{+/}$ $^{+}$mice ($n = 7$). sham-treated: $n = 5$, control=untreated: $n = 6$. One-way ANOVA. **F** The number of alveolar macrophages (AM) remained unchanged in BAL-samples of *Trpm5*-DREADD mice treated with 100 µM CNO ($n = 7$). *Trpm5*$^{+/+}$: $n = 7$, sham-treated: $n = 5$, control=untreated: $n = 6$. Kruskal-Wallis test. **G** The dendritic cell (DC) number was significantly increased in BALF samples of *Trpm5*-DREADD mice ($n = 6$) treated with 100 µM CNO after 30 min compared to CNO-treated *Trpm5*$^{+/+}$ mice ($n = 7$). sham-treated: $n = 4$, control=untreated: $n = 6$. One-way ANOVA. **H** In lungs of *Trpm5*-DREADD mice ($n = 7$) treated with 100 µM CNO the DC number was significantly increased after 30 min compared to CNO-treated *Trpm5*$^{+/+}$ mice ($n = 6$). sham-treated: $n = 5$, control=untreated: $n = 3$. One-way ANOVA. **A–H** Source data are provided in the Source Data file. Data in panels A–H are presented as mean ± SEM.

were significantly increased three days after *P. aeruginosa* infection when compared to non-infected control animals. The number of recruited DCs in *Trpm5*$^{-/-}$ as well as in *Panx1*$^{-/-}$ mice three days after infection was significantly lower compared to those in infected *Trpm5*$^{+/+}$ mice (Fig. 4C). We also observed a significant increase in DCs three days after intratracheal denatonium treatment in the lungs of *Trpm5*$^{+/+}$ and *Chat*$^{fl/fl}$:*Trpm5*$^{cre}$, but not *Trpm5*$^{-/-}$ mice (Fig. 4D). The denatonium-induced DC recruitment was significantly reduced in *Panx1*$^{-/-}$ mice when compared to controls (Fig. 4E). In line with this, *Trpm5*$^{+/+}$ mice infected with *P. aeruginosa* showed higher DC numbers than non-infected mice. DC numbers were increased in infected *Trpm5*$^{-/-}$ and *Panx1*$^{-/-}$ mice compared to non-infected controls, but highly reduced

when compared to infected *Trpm5*$^{+/+}$ mice (Fig. 4F). Similar results were obtained when the CD45$^{+}$ cell counts and the percentage of DCs were analyzed (Supplementary Figs. 13 and 14). The number of DCs was significantly increased in airway lymph nodes of *Trpm5*$^{+/+}$ mice three days after denatonium treatment, in contrast to *Trpm5*$^{-/-}$ mice, where no increase was observed (Fig. 4G). In denatonium-treated *Panx1*$^{-/-}$ mice, DCs were strongly reduced ($p = 0.059$) (Fig. 4H). Moreover, bitter taste signaling in tuft cells as well as ATP release via Panx1 is crucial for infection outcome, since we observed a reduced survival in *Trpm5*$^{-/-}$ as well as *Panx1*$^{-/-}$ mice, when compared to *Trpm5*$^{+/+}$ animals, and *Trpm5*$^{-/-}$ mice showed a greater weight loss 48 h post infection (Fig. 4I, Supplementary Fig. 15). This underlines the importance of tuft

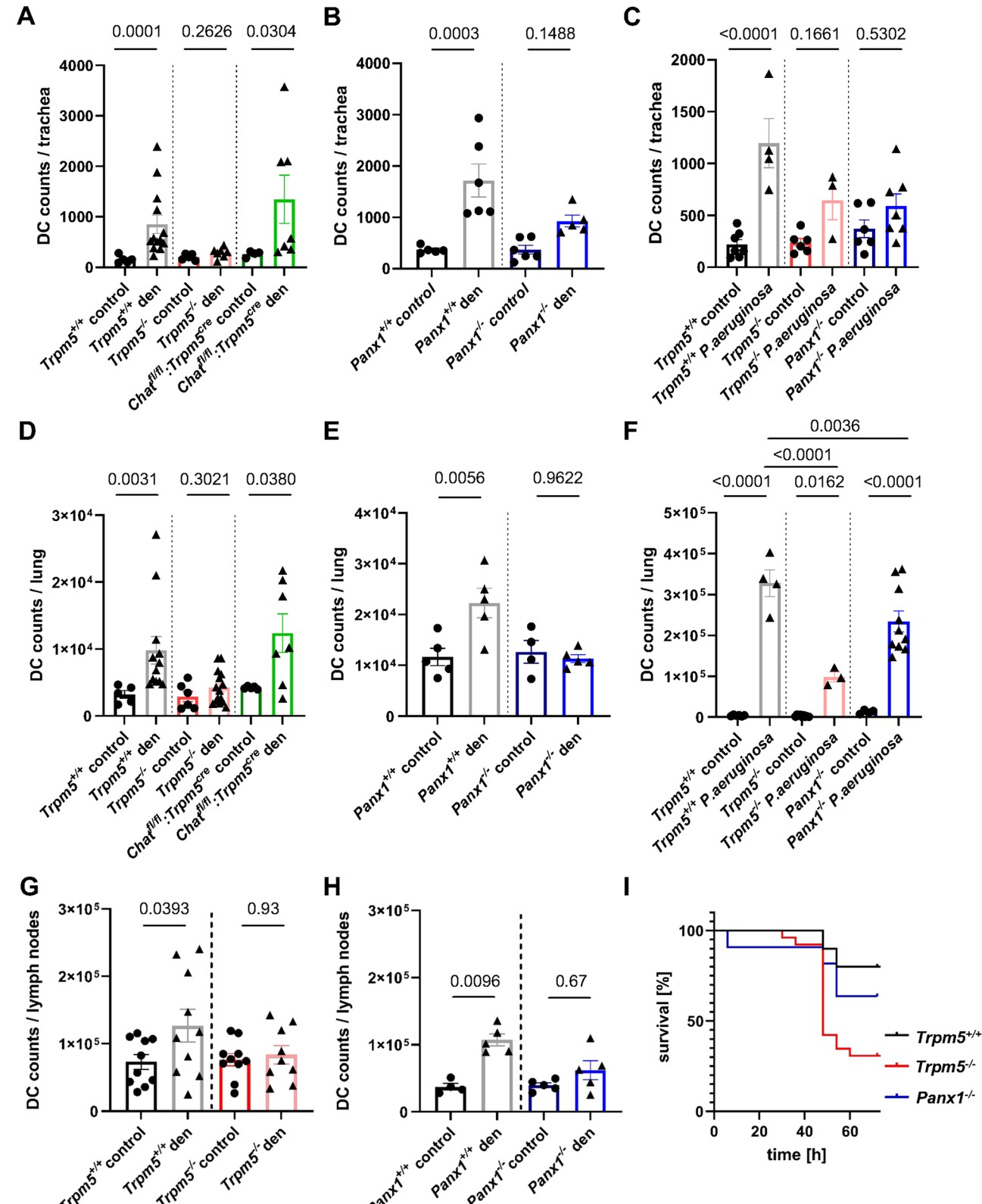

cell-dependent immune cell recruitment and specifically of DCs in the respiratory tract to infection.

**Airway dendritic cell activation depends on Trpm5 and Panx1**

Next, we investigated the activation status of dendritic cells (CD11c⁺CD86⁺) three days after tuft cell activation with denatonium or three days after infection with *P. aeruginosa* in tracheae and lungs. In

tracheae from *Trpm5⁺ᐟ⁺* and *Chat^fl/fl:Trpm5^cre* mice (ACh-deficient in tuft cells), we observed a significant increase in the percentage of CD86⁺ DCs after stimulation with denatonium, which was not found in *Trpm5⁻ᐟ⁻* mice (Fig. 5A). These findings corresponded to the percentage of activated DCs in the tracheae of denatonium-treated *Panx1⁻ᐟ⁻* mice which was significantly lower than in denatonium-treated *Panx1⁺ᐟ⁺* mice (Fig. 5B). In *Trpm5⁺ᐟ⁺* mice infected with *P. aeruginosa*, the

**Fig. 4 | FACS analyses of dendritic cell (DC) numbers three days after tuft cell stimulation by denatonium (1 mM) or infection with *Pseudomonas aeruginosa* NH57388A. A** DC (CD11b[+], CD11c[+], F4/80[-]) numbers in denatonium-treated or control tracheae of *Trpm5*[+/+] (control $n = 5$, den $n = 13$) *Trpm5*[-/-] mice (control $n = 5$, den $n = 7$) and *Chat*[fl/fl]:*Trpm5*[cre] mice (control $n = 4$, den $n = 7$) after three days. Kruskal-Wallis test. **B** DC numbers in tracheae of control or denatonium-treated *Panx1*[-/-] (control $n = 6$, den $n = 5$) and *Panx1*[+/+] mice (control $n = 6$, den $n = 6$). One-way ANOVA. **C** DC numbers three days post infection or in uninfected controls in *Trpm5*[+/+] (control $n = 7$, P.a. $n = 4$), *Trpm5*[-/-] (control $n = 6$, P.a. $n = 3$) and *Panx1*[-/-] mice (control $n = 6$, P.a. $n = 7$). One-way ANOVA. **D** DC numbers in lungs of denatonium-treated (1 mM) and control *Trpm5*[+/+] (control $n = 5$, den $n = 12$), *Trpm5*[-/-] mice (control $n = 6$, den $n = 15$) and *Chat*[fl/fl]:*Trpm5*[cre] mice (control $n = 5$, den $n = 7$). Kruskal-Wallis test. **E** DC numbers in lungs of denatonium-treated and control *Panx1*[-/-] (control $n = 4$, den $n = 5$) or *Panx1*[+/+] mice (control $n = 5$, den $n = 5$). One-way ANOVA. **F** DC numbers in lungs of infected *Trpm5*[+/+] ($n = 4$), *Trpm5*[-/-] ($n = 3$) and *Panx1*[-/-] mice ($n = 10$) and uninfected controls (*Trpm5*[+/+] $n = 6$, *Trpm5*[-/-] $n = 9$, *Panx1*[-/-] $n = 4$). One-way ANOVA. **G** DC numbers in airway lymph nodes of denatonium-treated *Trpm5*[+/+] ($n = 10$), control ($n = 10$) and *Trpm5*[-/-] mice (control $n = 10$, den $n = 9$). One-way ANOVA. **H** DC numbers in airway lymph nodes from denatonium-treated or control *Panx1*[+/+] ($n = 5$) and *Panx1*[-/-] mice ($n = 5$). Kruskal-Wallis test. **I** Survival rates of *Trpm5*[+/+] ($n = 10$), *Trpm5*[-/-] ($n = 26$) and *Panx1*[-/-] ($n = 11$) mice revealed decreased survival in *Panx1*[-/-] and *Trpm5*[-/-] mice compared to *Trpm5*[+/+] mice within the first 72 h after infection with *P. aeruginosa* NH57388A. Source data are provided in the Source Data file. Data in panels A–H are presented as mean ± SEM.

percentage of CD86[+] DCs in the trachea was significantly increased after three days compared to non-infected control animals, while this increase was completely abolished in *Trpm5*[-/-] and *Panx1*[-/-] mice (Fig. 5C). Similar findings were observed in the lungs of *Trpm5*[+/+] and *Chat*[fl/fl]:*Trpm5*[cre] mice, displaying a significant increase in CD86[+] DCs that was abolished in *Trpm5*[-/-] mice (Fig. 5D) and reduced in *Panx1*[-/-] mice (Fig. 5E). In lungs of *Trpm5*[+/+], *Trpm5*[-/-] and *Panx1*[-/-] mice infected with *P. aeruginosa*, the number of CD86[+] DCs was increased after three days (Fig. 5F). Yet, infected *Trpm5*[-/-] mice displayed less CD86[+] DCs in the lungs than infected *Trpm5*[+/+] mice (Fig. 5F). Taken together, these results demonstrate that tuft cells play a prominent role in local DC activation at the site of infection, which involves Trpm5 and Panx1 channels but is independent from ACh release from tuft cells.

## Denatonium activation of tuft cells leads to T$_H$17 recruitment in mouse lungs

Since we observed a higher recruitment and activation of DCs in lungs of *Trpm5*[+/+] mice when compared to *Trpm5*[-/-] mice and *Panx1*[-/-] mice after tuft cell activation with denatonium for three days, we next investigated the impact of these changes on adaptive immune responses. Higher T$_H$17 cell counts were determined in the lungs of *Trpm5*[+/+] mice after tuft cell activation when compared to controls or *Trpm5*[-/-] mice (Fig. 5G). Reduced numbers of T$_H$17 cells were recruited to the lungs of *Panx1*[-/-] mice compared to the denatonium-treated *Panx1*[+/+] mice (Fig. 5H). In addition, an increased percentage of cells showing the phenotype of T$_H$17 (CCR6[+] IL17A[+]) within CD4[+] cells were detected in the lungs of *Trpm5*[+/+] mice after denatonium activation of tuft cells (Supplementary Fig. 16A). These changes were not observed in *Trpm5*[-/-] mice (Supplementary Fig. 16A). The percentage of T$_H$17 cells within CD4[+] cells was reduced in denatonium-treated *Panx1*[-/-] mice when compared to denatonium-treated *Panx1*[+/+] mice (Supplementary Fig. 16B). Consistently, T$_H$17 cell counts were increased two days post infection with *P. aeruginosa* in the lungs of *Trpm5*[+/+], but not of *Trpm5*[-/-] mice (Supplementary Fig. 16C). Furthermore, the concentration of IL-17A, a key cytokine in host protective immunity to infection secreted for example by T$_H$17 cells, was elevated in the plasma of denatonium-treated *Trpm5*[+/+] mice compared to controls but not in plasma of *Trpm5*[-/-] mice (Fig. 5I). Denatonium-treated *Panx1*[-/-] mice showed lower levels of IL-17A in their plasma in comparison to the *Panx1*[+/+] mice (Fig. 5J). Since γδ T cells and ILC3 may also be a source of IL-17A, we investigated these cell types following tuft cell stimulation[37,38]. No differences in γδ T cell numbers were observed in the lungs or blood samples after denatonium treatment (Supplementary Figs. 16D and E). ILC3 numbers remained unchanged in lungs of both *Trpm5*[+/+] and *Trpm5*[-/-] mice three days after tuft cell activation with denatonium, compared to the respective untreated controls (Supplementary Fig. 16F). These data indicate that not only innate immune processes and processes bridging the innate and adaptive immunity are initiated and shaped by tuft cells, but also adaptive immune responses, which occur as early as day three post treatment.

## Activation, phagocytosis and migration of dendritic cells depends on ATP release by tuft cells

To investigate if the upregulation of CD86 in DCs depends on tuft cell activation and subsequent ATP release, we purified pulmonary DCs from the lungs of healthy *Trpm5*[+/+] mice. Subsequently, we co-cultured them with either primary tracheal epithelial cells from *Trpm5*-DREADD mice, *Trpm5*[+/+] mice or alone. After stimulation of the tracheal tuft cells with CNO (100 μM) we observed a higher percentage of CD86[+] DCs when compared to controls. This effect was abolished when we applied apyrase (5 U/ml), an ATP/ADP-hydrolyzing enzyme, along with the CNO (Fig. 6A–F). However, this effect remained unchanged when we added the nicotinic receptor antagonist mecamylamine (10 μM) and the muscarinic receptor antagonist atropine (1 μM) to the CNO-activated tuft cells. These data demonstrate that this effect is due to the ATP release rather than to ACh, which is also released after tuft cell activation. In support of this, apyrase (5 U/ml) did not change the ACh levels when added to a buffer solution containing 10 nM ACh detected by MALDI-TOF, indicating that apyrase (5U/ml) does not affect and degrade ACh per se (Supplementary Fig. 17A). Yet the concentration of apyrase used degraded ATP effectively, since the ATP concentration in the tracheal supernatants from denatonium-treated wild-type mice was robustly reduced (Supplementary Fig. 17B). Next, we performed co-culture experiments using DCs isolated from *Panx1*[-/-] mouse lungs and tracheal epithelial cells from *Trpm5*-DREADD mice to exclude the possibility that the reduction in DC activation in *Panx1*[-/-] mice was not due to the absence of the channel in DCs[39]. We observed an increase in the percentage of activated DCs (CD86[+]) after stimulation of tuft cells with CNO (Fig. 6G). In contrast, no change in CD86 was observed in control experiments in which *Panx1*[-/-] lung DCs were treated with CNO in the absence of tuft cells (Fig. 6G). These results further confirm that DC activation depends on ATP release from tuft cells via Panx1 rather than from DCs. MHCII expression levels in DCs were also upregulated after co-culture with activated tuft cells. However, the effect was abolished by apyrase, mecamylamine and atropine (Fig. 6H), which suggests that both ATP and ACh play a role in this upregulation. To determine the functional role of DC recruitment through tuft cell activation, we infected bone marrow-derived DCs with *P. aeruginosa* and performed a series of phagocytic assays (Fig. 6I). Prior to infection, DCs were treated with tracheal supernatants obtained from denatonium-treated *Trpm5*[+/+] and *Trpm5*[-/-] mice. The phagocytic capacity of DCs treated with *Trpm5*[+/+] tracheal supernatants was enhanced in comparison to those treated with *Trpm5*[-/-] tracheal supernatants (Fig. 6J). When apyrase was added to the tracheal supernatants obtained from *Trpm5*[+/+] mice before application to the DCs, the number of phagocytosed bacteria decreased significantly (Fig. 6J, K). These findings emphasize the importance of tuft cells as a source of ATP in the trachea which is also needed for enhancement of the phagocytic capacity of DCs and most probably contributes to the better outcome for *Trpm5*[+/+] mice following infection.

Next, we investigated the influence of tuft cells in DC migration, by assessing the expression of CCR7, a key regulator of DC

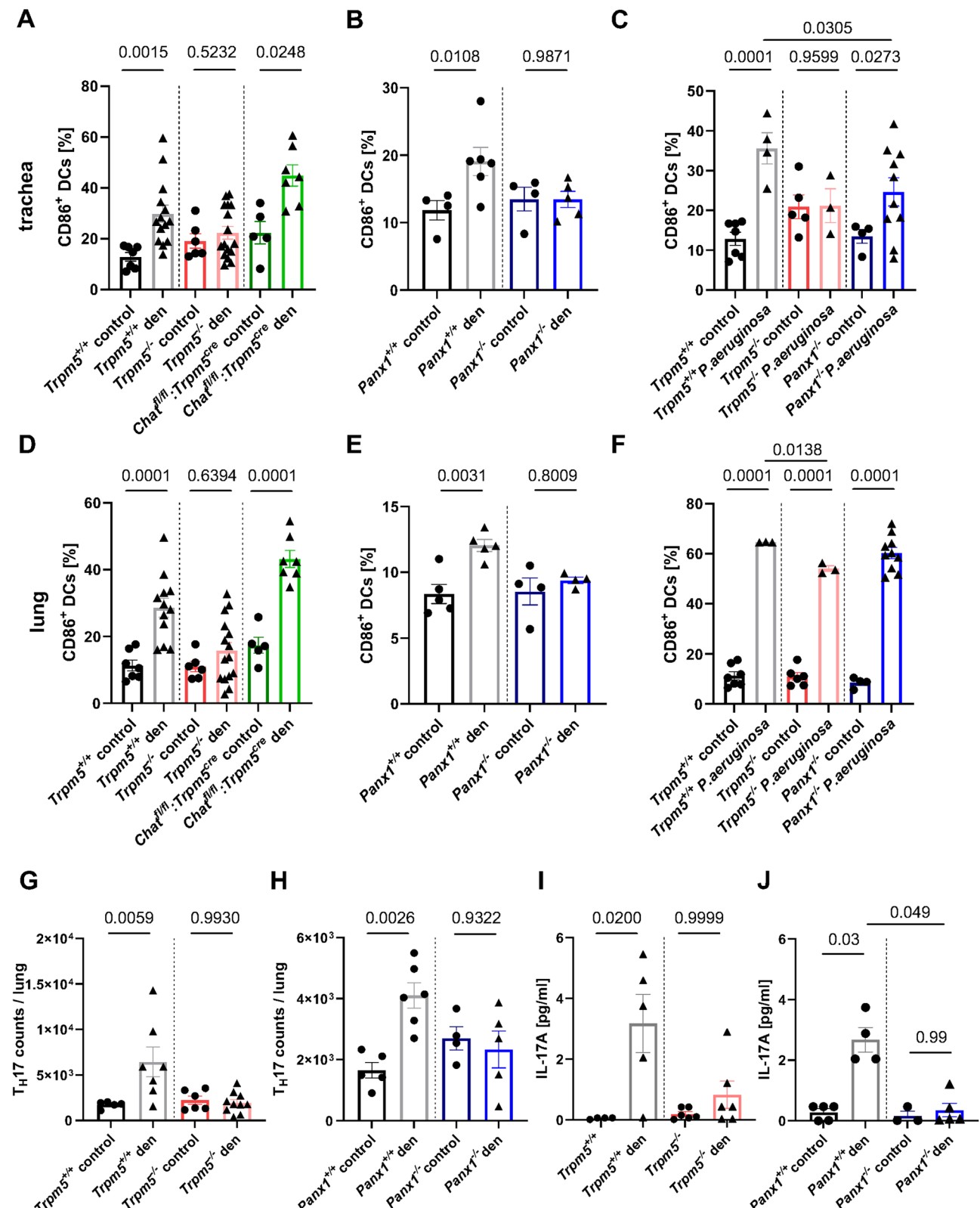

migration[40,41]. CCR7 was notably upregulated in DCs when they were co-cultured with CNO-treated tracheal epithelial cells, as well as when tuft cells were stimulated with denatonium in vivo (Fig. 7A and B and Supplementary Fig. 18). This increase in CCR7+ DCs was ATP-dependent, as it was abolished by apyrase in co-culture experiments (Fig. 7A). In support of the role of CCR7 in migration of DCs, we found an increase in CCR7-expressing DCs in lymph nodes in *Trpm5+/+* mice,

which was absent in *Trpm5−/−* mice (Fig. 7C). Moreover, we observed an increased expression of *Ccl21b* after denatonium-stimulation in tracheae and lungs from *Trpm5+/+* but not *Trpm5−/−* mice (Supplementary Figs. 19A and B) as well as an increase in *Ccl19* in the airway-draining lymph nodes (Supplementary Fig. 19C). In line with these observations, *Ccl21b* expression is mainly linked to peripheral tissues, such as the lung[42]. Since CCR7, CCL21 and CCL19 are crucial for DC migration into

**Fig. 5 | FACS analyses of the percentage of activated (CD11b⁺, CD11c⁺, F4/80⁻, CD86⁺) dendritic cells (DCs) and T$_H$17 cell recruitment after three days.**
**A** Percentage of CD86⁺ DCs in tracheae of denatonium-treated (1 mM) *Trpm5*⁺/⁺ (*n* = 14), controls (*n* = 7), *Trpm5*⁻/⁻ mice (control *n* = 6, den *n* = 16) and *Chat*^fl/fl^:*Trpm5*^cre^ mice (control *n* = 5, den *n* = 7). Kruskal-Wallis test. **B** Percentage of CD86⁺ DCs in tracheae of denatonium-treated or control *Panx1*⁺/⁺ (control *n* = 4, den *n* = 6) and *Panx1*⁻/⁻ mice (control *n* = 4, den *n* = 5). One-way ANOVA. **C** Percentage of CD86⁺ DCs in tracheae three days post infection or in uninfected controls in *Trpm5*⁺/⁺ (control *n* = 7, P.a. *n* = 4), *Trpm5*⁻/⁻ (control *n* = 5, P.a. *n* = 3) and *Panx1*⁻/⁻ mice (control *n* = 4, P.a. *n* = 10). One-way ANOVA. **D** Percentage of CD86⁺ DCs in lungs of denatonium-treated *Trpm5*⁺/⁺ (*n* = 12), control (*n* = 7), *Trpm5*⁻/⁻ mice (control *n* = 6, den *n* = 15) and *Chat*^fl/fl^:*Trpm5*^cre^ mice (control *n* = 5, den *n* = 7). One-way ANOVA. **E** Percentage of CD86⁺ DCs in lungs of denatonium-treated or control *Panx1*⁺/⁺

(*n* = 5) and *Panx1*⁻/⁻ (*n* = 4) mice. Unpaired Student's *t*-test. **F** Percentage of CD86⁺ DCs in lungs three days post infection or in uninfected controls of *Trpm5*⁺/⁺ (control *n* = 7, P.a. *n* = 3), *Trpm5*⁻/⁻ (control *n* = 6, P.a. *n* = 3) and *Panx1*⁻/⁻ mice (control *n* = 4, P.a. *n* = 10). One-way ANOVA. **G** FACS analysis of T$_H$17 cells in lungs of denatonium-treated *Trpm5*⁺/⁺ (*n* = 7) and *Trpm5*⁻/⁻ (*n* = 10) mice or controls (*Trpm5*⁺/⁺ *n* = 5, *Trpm5*⁻/⁻ *n* = 6). One-way ANOVA. **H** T$_H$17 cell counts from lungs of denatonium-treated and control *Panx1*⁺/⁺ (control *n* = 5, den *n* = 6) and *Panx1*⁻/⁻ mice (control *n* = 4, den *n* = 5). One-way ANOVA. **I** IL-17A levels in plasma samples in control or denatonium-treated *Trpm5*⁺/⁺ (control *n* = 4, den *n* = 5) and *Trpm5*⁻/⁻ (control *n* = 6, den *n* = 6) mice. Kruskal-Wallis test. **J** IL-17A levels in plasma samples of control (*n* = 5) or denatonium-treated *Panx1*⁺/⁺ mice (*n* = 4) or *Panx1*⁻/⁻ mice (control *n* = 3, den *n* = 5). Kruskal-Wallis test. Source data are provided in the Source Data file. Data in panels A–J are presented as mean ± SEM.

lymph nodes upon denatonium stimulation, these data align with our findings of increased DC numbers in lymph nodes (Fig. 4G), indicating that tuft cell stimulation indeed promotes enhanced migration of DCs to the secondary lymphatic organs.

To test if DC migration depends on tuft cell-released ATP, we performed scratch assays of DCs treated with different tracheal supernatants (Figs. 7D, Supplementary Fig. 20 and 21). Tracheal supernatants from *Trpm5*⁺/⁺ mice stimulated with denatonium (1 mM) increased DC migration (Fig. 7E). This effect was abolished in the presence of apyrase (5 U/ml) (Fig. 7E). In mice with impaired bitter taste signaling in tuft cells (*Trpm5*⁻/⁻) and in mice lacking tuft cells (*Trpm5*-DTA), the DC migration was reduced compared to controls (Fig. 7E). The wound closure of DCs induced by supernatants from *Trpm5*-DREADD mice stimulated with CNO (100 µM) was diminished upon apyrase (5 U/ml) treatment (Fig. 7E). These data demonstrate a crucial role for tuft cell-dependent ATP signaling in promoting DC migration (Fig. 7F).

## Discussion

We have delineated a role for ATP-dependent signaling in the later stages of tuft cell-induced innate immune responses in the airways, especially for DC recruitment and activation in response to pneumonia. Tuft cell-released ATP triggers innate and adaptive immune responses crucial to fight bacterial infections. Until now, tuft cell-induced innate immune responses have been attributed to ACh, CysLT or IL-25 release from tuft cells, depending on the tuft cell-activating stimulus, i.e., bacterial substances vs. aeroallergens[4–6,16–18]. Auto- and paracrine tuft cell signaling have been described for ACh released by a mechanism involving the Trpm5 channel[5,6]. Trpm5 has been established as a marker for tuft cells in the airway epithelium which opened further perspectives for studying tuft cell function in vivo[5,7,18,26,43,44]. By combining patch-clamp of tuft cells with imaging of ATP sensor cells we show that ATP is released from tuft cells in a Trpm5-dependent manner after their activation. Thus, tuft cells appear to be a previously unrecognized source of extracellular ATP.

Previously published RNA-seq analyses have identified *Panx1* expression in the population of tuft cells expressing the taste signaling components *Gnat3*, *Plcb2* and *Trpm5*[5,26]. Our patch-clamp results demonstrate the existence of functional Panx1 channels in tuft cells, which are responsible for the ATP release. In contrast to taste buds where activation of bitter taste signaling leads to ATP release[13] relying on Calhm1[14] or heteromeric Calhm1/3 ion channels[15], inhibition of Calhm1 channels by ruthenium red did not alter the denatonium-mediated ion transport. While we propose a direct release of ATP from tuft cells, we cannot exclude that a Ca²⁺ wave initiating from tuft cells could spread to neighboring cells via gap junction channels, causing these cells to release ATP. In line with this alternative theory, our previous RNAseq[5] and patch clamp results indicate that connexins in tuft cells form Ca²⁺- or ATP-permeable[29] gap junctions with adjacent cells. Independent of the ATP release mechanism, a purinergic feedback loop acting on tuft cells may exist, since tracheal tuft cells express

purinergic receptors, as do two different subpopulations of nasal SCCs[16].

We expand our previous findings that denatonium-evoked tuft cell stimulation induces neurogenic inflammation[4] and found that chemogenetic tuft cell activation leads to neutrophil, monocyte, and alveolar macrophage recruitment. In addition, DCs were increased in bronchoalveolar lavage fluid and the lungs. Moreover, three days after tuft cell stimulation with denatonium or in *P. aeruginosa* infections the DC population was increased and the recruited DCs were activated depending on both, ATP release and Trpm5-signaling, but not on cholinergic signaling, in tracheal tuft cells. This notion was supported by the reduced DC migration/recruitment in *Trpm5*⁻/⁻ and *Panx1*⁻/⁻ mice and as a result of exposure of apyrase, an ATP/ADP-hydrolyzing enzyme[45]. In addition, tuft cell-dependent ATP improved the chemotaxis of DCs during bacterial infections. We demonstrate that the increase in CD86⁺ DCs depends on ATP and not on ACh from tuft cells and is independent from the expression of Panx1 in DCs, while the expression of MHCII was mediated by ACh in addition to ATP. This is in line with the previously reported ACh release upon tuft cell activation[5,6] as well as expression of purinergic[46], muscarinic and nicotinic ACh receptors within DCs[47,48]. Of note, the increased expression of CD86 in DCs resulted in improved phagocytic capacity during bacterial infections. MHCII induces macrophage activation and antibody production from plasma cells[49] supporting the importance of the observed tuft cell-dependent ATP-mediated upregulation of both molecules for the ameliorated infection outcome.

The upregulation of CCR7 suggests a link between the observed DC activation and the increase in T$_H$17 cells in the adaptive immune system. DCs increase the expression of CCR7 when exposed to extracellular ATP[50]. This receptor regulates migratory speed of DCs to secondary lymphatic organs, as well as their chemotaxis[40]. The upregulation of CCL21 and CCL19, both ligands of CCR7[41], supports an increased migration and chemotaxis of DCs, since both chemokines are crucial for homing of DCs from peripheral organs (e.g., lungs) to secondary lymphatic organs, (e.g., lymph nodes)[41,51]. Furthermore, CCR7 plays a role in the differentiation and activation of T cells[41], since in the mesenteric lymph nodes CCR7⁺ CD11b⁺ DCs were responsible for priming of T$_H$17 cells[52]. Our observation of increased DC numbers in lymph nodes following tuft cell activation, along with the upregulation of CCR7 are suggestive of a mechanism for the increased differentiation and recruitment of T$_H$17 cells observed in our study. Additionally, the elevated levels of IL-17A further support the increased activation of T$_H$17 cells. Since we did not observe changes in γδ T cell or in ILC3 levels following tuft cell stimulation, it is unlikely that these cells are involved in the rise in IL-17A levels.

So far, tuft cell-mediated expansion of DCs was attributed to CysLT- and IL-25-dependent signaling resulting in aeroallergen-evoked type 2 inflammation[17], an experimental setting in which CysLTs also induced tuft cell expansion[18]. Here, we extend these findings by demonstrating DC expansion and even activation after tuft cell stimulation in bacterial infection. Moreover, the observed DC

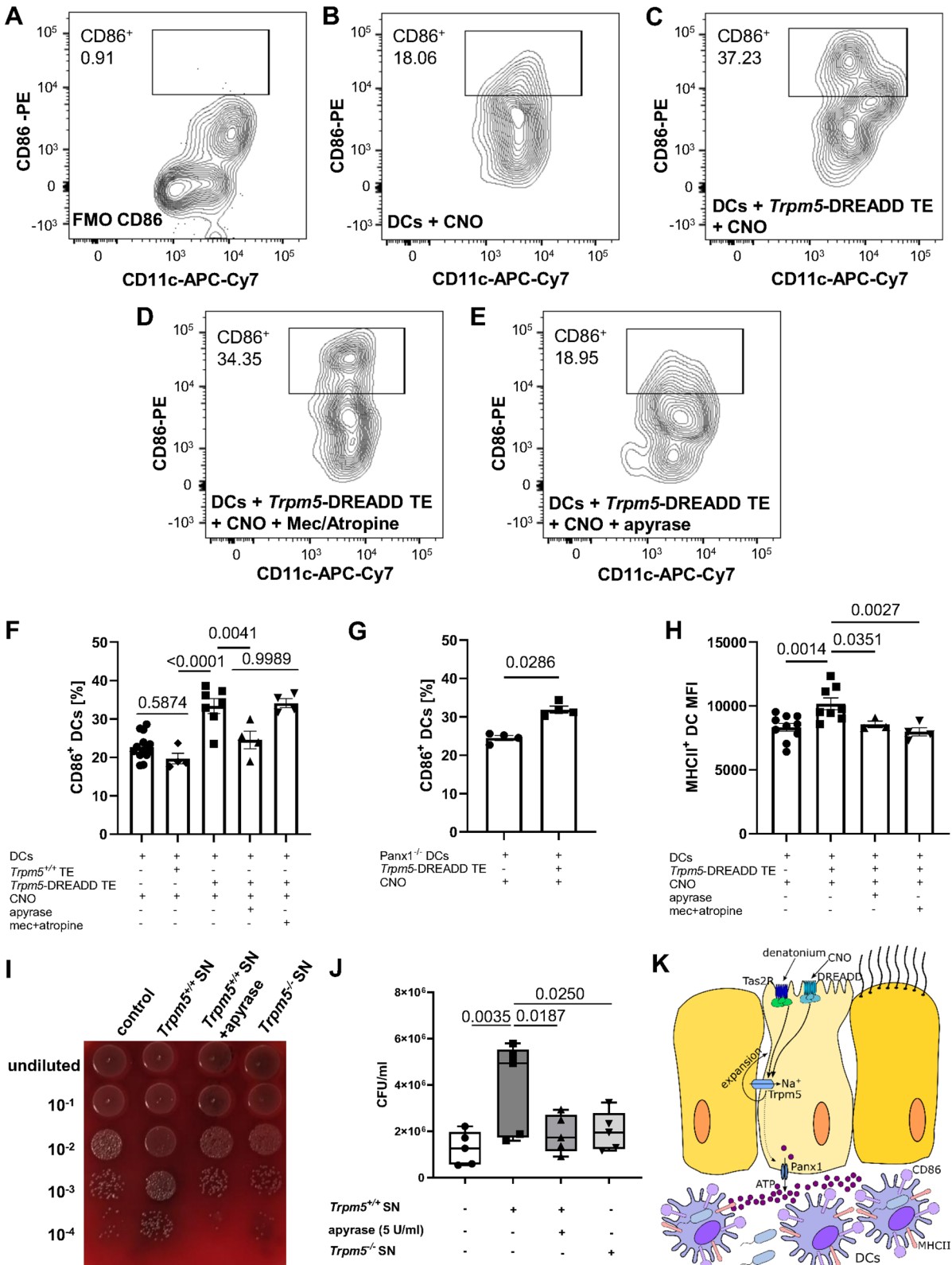

recruitment in pneumonia and to tuft cell denatonium stimulation was dependent on Trpm5 and Panx1 signaling and correlated with an increased tuft cell number. This suggests that ACh and ATP released from tuft cells are either directly responsible for this observed expansion or trigger a tuft cell-mediated release of CysLTs and IL-25, as proposed by Bankova et al.[18]. In their study, tuft cell expansion in response to aeroallergens was initiated by $LTC_4$ production and

secretion from tuft cells. $LTC_4$ converted to $LTE_4$, which then acts in an autocrine manner on $CysLT_3$ receptors, leading to IL-25 release and subsequent tuft cell expansion. IL-25 happens to regulate tuft cell differentiation and expansion in the gut, where parasitic helminths stimulate tuft cells to release IL-25, which in turn activates ILC2s to produce IL-13. IL-13 acts on gut crypt epithelial progenitor cells[53]. However, the exact mechanism underlying the tuft cell expansion

**Fig. 6 | Upregulation of CD86 on dendritic cells (DCs) is dependent on ATP released from tuft cells. A–F** FACS analysis of CD86 expression on sorted pulmonary DCs (CD45$^+$Ly6G$^-$F4/80$^-$CD11c$^+$) co-cultured with sorted *Trpm5*-DREADD tracheal epithelium (EpCAM$^+$CD45$^-$) ($n = 7$ mice). Clozapine-n-oxide (CNO) (100 μM) was used to stimulate *Trpm5*-DREADD tracheal epithelium in presence or absence of ATP hydrolyzing enzyme apyrase (5 U/ml) or nicotinic receptor antagonist mecamylamine (10 μM) and muscarinic receptor antagonist atropine (1 μM) in the co-culture system. The percentage of CD86-expressing DCs was significantly higher when DCs were co-cultured with CNO-stimulated *Trpm5*-DREADD tracheal epithelium compared to CNO treated DCs ($n = 10$ mice) or CNO treated *Trpm5*$^{+/+}$ tracheal epithelium. Treatment of CNO-stimulated *Trpm5*-DREADD tracheal epithelium with apyrase but not mecamylamine/atropine reduced the percentages of CD86$^+$ DCs ($n = 4$ mice). One-way ANOVA. **G** Co-culture of pulmonary DCs from *Panx1*$^{-/-}$ mice with CNO-stimulated *Trpm5*-DREADD tracheal tuft cells significantly increased the percentages of CD86 expressing DCs compared to CNO treated *Panx1*$^{-/-}$ DCs ($n = 4$ mice). Two-tailed Mann-Whitney test. **H** The median

fluorescence intensity (MFI) of MHCII in pulmonary DCs was significantly increased after being co-cultured with CNO-stimulated *Trpm5*-DREADD tracheal epithelium (DC: $n = 10$ mice, DC + TE: $n = 8$ mice). This was reduced when CNO-stimulated *Trpm5*-DREADD tracheal epithelium was treated with apyrase ($n = 3$ mice), or mecamylamine and atropine ($n = 4$ mice). One-way ANOVA. **I** Blood agar plate of phagocytosed *P. aeruginosa* NH57388A CFUs after treatment of DCs with supernatant (SN) of *Trpm5*$^{+/+}$ tracheae stimulated with 1 mM denatonium, with supernatant of *Trpm5*$^{+/+}$ tracheae stimulated with denatonium in the presence 5 U/ml apyrase and with supernatants of denatonium-treated *Trpm5*$^{-/-}$ tracheae and untreated control conditions. **J** Quantification of (A) ($n = 5$ mice). Data are shown as a box-and-whisker plot displaying the minimum and maximum values, with the median represented by a horizontal line. One-way ANOVA. **K** Schematic drawing. Activation of tuft cells with denatonium or CNO leads to an ATP-dependent increased in the number and activation of DCs. (* $p < 0.05$, ** $p < 0.01$). Source data are provided in the Source Data file. Data in panels (**F, G** and **H**) are presented as mean ± SEM.

observed in our study remains elusive. Nevertheless, it is tempting to speculate that the increased number of tuft cells may result in a greater release of signaling molecules in the airways and lungs, potentially affecting their function.

Overall, we propose a protective influence of tuft cell-released ATP in bacterial infections, since *Trpm5*$^{-/-}$ and *Panx1*$^{-/-}$ mice demonstrated worse infection outcomes. The observed increase in T$_H$17 cell differentiation and IL-17A concentrations after tuft cell stimulation emphasizes the importance of tuft cells in fostering a more favorable outcome in *P. aeruginosa* infections. This is in line with previous work which showed that IL-17 is crucial to fight infections with *Klebsiella pneumoniae*[54] or *Streptococcus pneumoniae*[55]. Furthermore, our findings of an early tuft cell-dependent increase of T$_H$17 cell numbers in infection supports the hypothesis, that these cells are necessary to boost the initial immune defense for clearing the bacteria and are downregulated afterwards to avoid detrimental effects due to chronic inflammation. T$_H$17 cells release IL-17A at the site of inflammation, leading to an increase in pro-inflammatory cytokines as well as antimicrobial peptides such as β-defensin-2[56,57], thereby supporting early protective immune responses necessary to combat bacterial infections. On the other hand, termination of T$_H$17-dependent IL-17A production is essential in order to prevent inflammatory diseases caused by a persistent presence of this cytokine[58].

Nevertheless, some open questions remain. A subset of vagal sensory neurons express the purinergic receptor P2rx2 and P2rx3[59,60]. It remains to be determined whether tuft cell-mediated ATP-dependent mucosal defense 1) depends on sensory neural signaling and 2) might further lead to the development of type 2 inflammation. Future studies will be needed to determine the mechanisms by which tuft cell-released ATP transmits the resolution of inflammation. A possible hypothetical scenario for the anti-inflammatory effect of tuft cell-released ATP might be the degradation of ATP by ectonucleotidases to its metabolite adenosine, a known anti-inflammatory mediator[46], once a sufficient number of immune cells are recruited. While these open questions remain, we regardless provide strong evidence that tuft cell-released ATP induces protective immune responses, which bridge innate and adaptive immunity and are crucial to fight bacterial infections.

## Methods
### Animals
All animal experimental and care procedures were conducted in accordance with the German guidelines for care and use of laboratory animals and approved by the animal welfare committee of Saarland (approval numbers 04/2018, 40/2018, 09/2021, 11/2024). Mice of both genders older than 8 weeks of age were used from the following genetic lines: *Trpm5*$^{-/-}$[61], *ChAT*-eGFP[62], *Trpm5*$^{-/-}$-*ChAT*-eGFP

(Tg$^{(RP23-268L19-EGFP)2Mik}$;Trpm5$^{tm1Dgen(129S5/SvEvBrd-C57BL/6)}$), *Trpm5*-DTA[6], *Trpm5*-DREADD (designer receptors exclusively activated by designer drugs)[63], *Trpm5*-DREADD-tGFP[63], *ChAT*$^{fl/fl}$:*Trpm5*$^{cre}$ B6;129-Trpm5$^{tm1.1(cre)Uboe}$;B6;129-Chat$^{tm1Jrs}$/J, Pannexin1$^{-/-}$ (*Panx1*$^{-/-}$)[64], Pannexin2$^{-/-}$ (*Panx2*$^{-/-}$)[28] and Pannexin1$^{-/-}$/2$^{-/-}$ (*Panx1*$^{-/-}$/2$^{-/-}$)[28] as well as wild-type controls. *Panx1*$^{-/}$, *Panx2*$^{-/-}$ and *Panx1*$^{-/-}$/2$^{-/-}$ mouse lines were generated and kindly provided by Hannah Monyer (Department of Clinical Neurobiology, University of Heidelberg, Heidelberg, Germany). *Panx1*$^{-/}$, *Panx2*$^{-/-}$ and *Panx1*$^{-/-}$/2$^{-/-}$ and *Panx1*$^{+/+}$ mouse lines were on a C57BL/6 J background and all other lines were kept in a mixed (129/SvJ and C57BL/6 J) background. To generate the *Trpm5*-DTA mice (Trpm5-IRES-Cre-R26:lacZbpAfloxDTA) Rosa26-DTA mice[65] were mated with *Trpm5*-IRES-Cre mice[66]. The *Trpm5*-DREADD mice were generated by breeding eRosa26-RFP-Dreadd mice with *Trpm5*-IRES-Cre mice[63]. The *Trpm5*$^{-/-}$-*ChAT*-eGFP mice were generated by breeding *Trpm5*$^{-/-}$ with *ChAT*-eGFP mice. The *ChAT*-eGFP, the *Trpm5*$^{-/-}$, the *Trpm5*$^{-/-}$-*ChAT*-eGFP, the *Panx1*$^{-/-}$, the *Panx2*$^{-/-}$ and the *Panx1*$^{-/-}$/2$^{-/-}$ mice were bred and housed in IVC cages in the animal facility of the Institute of Experimental Surgery of Saarland University under standardized 12 h day–night cycles. The *Trpm5*-DTA and the *Trpm5*-DREADD mice were bred and housed under SPF conditions with 12 h day-night cycles in the animal facility of the Institute of Experimental and Clinical Pharmacology of Saarland University. All mice were kept under standardized 12 h dark/light cycles with an ambient temperature of around $22 \pm 2$ °C and a humidity of $55 \pm 10\%$ Food and water were provided *ad libitum*. Mice of both genders were used in the experiments and no gender specific effects were studied. Mice were euthanized with an overdose of isoflurane followed by cervical dislocation or by an overdose of ketamine/xylazine according to the respective animal protocol.

### ATP sensor experiments
For ATP-release experiments tracheae from *Trpm5*$^{+/+}$ and *Trpm5*$^{-/-}$ mice were stimulated with 1 mM denatonium. The tracheae were then homogenized, centrifuged for 5 min at RT with 1500x g and the supernatants were frozen until further use. HEK293 cells (ATCC No. CRL-1573) were transiently transfected with an ATP sensor (pm-iATPSnFR1.1)[67] according to the manufacturer's protocol. The pm-iATPSnFR1.1 plasmid was a gift from Baljit Khakh (Addgene plasmid #102549; http://n2t.net/addgene:102549; RRID: Addgene_102549). The changes in pm-iATPSnFR1.1 fluorescence were measured at 525 nm emission before and after application of tracheal supernatants. In a different set of experiments, ATP-measurements were conducted simultaneously to whole-cell patch clamp experiments described below. A more detailed description on the ATP sensor experiments can be found in the Supplemental materials.

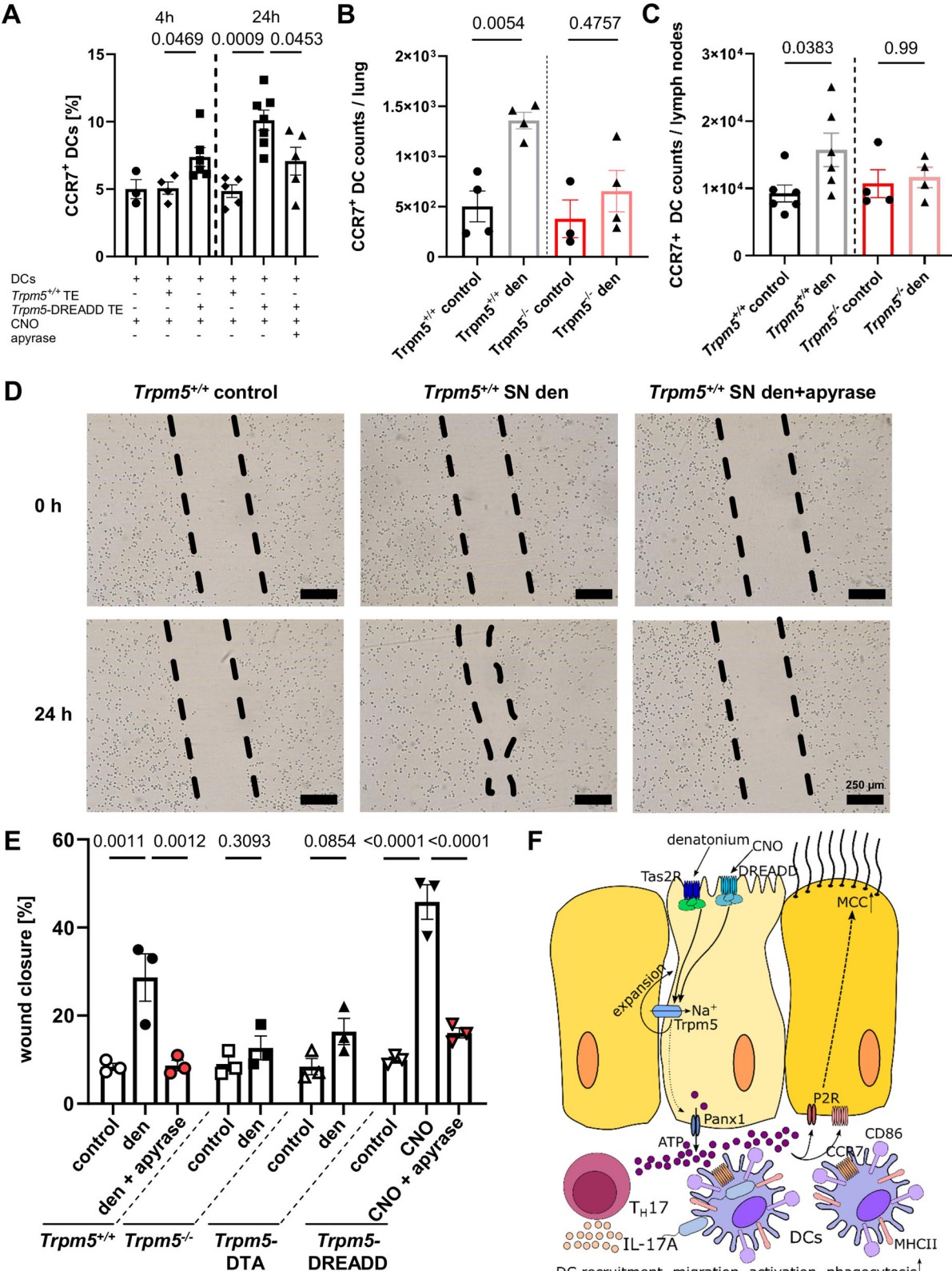

## ATP measurements

The concentration of ATP in tracheal supernatants was assessed using a commercially available ELISA kit according to the manufacturer's instructions (Cloud Clone Corp., Houston, TX, USA). A more detailed description can be found in the Supplemental materials.

## ACh measurements

ACh was measured in tracheal supernatants or buffer solution with MALDI-TOF (Matrix-assisted laser desorption/ionization-time of flight) mass spectrometry experiments. A detailed description of the procedure can be found in the Supplemental materials.

**Fig. 7 | Tuft cell-induced dendritic cell (DC) phagocytic activity and migration is dependent on ATP. A** Co-culture of CNO-stimulated *Trpm5*-DREADD tracheal epithelium (TE) with DCs significantly increased the percentage of CCR7-expressing DCs after 4 h ($n = 6$ mice) and 24 h ($n = 7$ mice) compared to CNO-treated Trpm5$^{+/+}$ TE co-culture with DCs (4 h: $n = 4$ mice, 24 h: $n = 5$ mice). The increase in % of CCR7$^+$ DCs after 24 h was abolished by apyrase (5 U/ml) ($n = 5$ mice). DCs cultured alone and treated with CNO ($n = 3$ mice) exhibited the same percentage of CCR7-expressing cells as DCs co-cultured with Trpm5$^{+/+}$ TE. One-way ANOVA. **B** The number of CCR7-expressing DCs in lungs increased significantly 3 days after intratracheal tuft cell stimulation with 1 mM denatonium in Trpm5$^{+/+}$ mice in contrast to Trpm5$^{-/-}$ mice ($n = 4$). One-way ANOVA. **C** In *Trpm5*$^{+/+}$ mice ($n = 6$) CCR7-expression DC counts in lymph nodes are increased 3 days after denatonium treatment, but not in *Trpm5*$^{-/-}$ mice ($n = 4$). Kruskal-Wallis test. **D** DC migration under different experimental conditions: untreated, treated with

supernatants (SN) from denatonium-treated *Trpm5*$^{+/+}$ mice without and with apyrase. **E** Quantification of (**D**). DCs treated with supernatants of unstimulated tracheae of *Trpm5*$^{+/+}$, *Trpm5*-DREADD, *Trpm5*$^{-/-}$ and *Trpm5*-DTA mice as well as with supernatants of tracheae stimulated with 1 mM denatonium of *Trpm5*$^{+/+}$ mice with and without apyrase (5 U/ml), of denatonium-treated *Trpm5*$^{-/-}$ and *Trpm5*-DTA tracheae, and from CNO-treated (100 μM) *Trpm5*-DREADD tracheae with and without apyrase. Wound closure was evaluated 24 h after induction of the scratch ($n = 3$). One-way ANOVA or two-tailed unpaired Student's *t*-test. **F** Graphical summary. Stimulation of tuft cells with denatonium or CNO activates the Trpm5 channel, resulting in ATP release via pannexin 1 and tuft cell expansion. The released ATP drives DC recruitment, migration, activation, and phagocytosis, promotes T$_H$17 cell recruitment and IL-17A secretion. Source data are provided in the Source Data file. Date in panels A, B, C and E are presented as mean ± SEM.

## Immunohistochemistry

Immunohistochemistry experiments on tracheal sections were performed as described previously[5]. To demonstrate co-localization of the tuft cell marker Trpm5 together with beta-galactosidase (lacZ) by immunohistochemical means, bright field documentation was performed as previously published using lacZ histochemistry in combination with immunohistochemistry[68,69]. A detailed description of the used procedures and the antibodies used can be found in the Supplemental materials.

## Patch-clamp experiments

Whole-cell patch-clamp experiments were performed on isolated tuft cells. A detailed description on the cell isolation, the patch-clamp procedure and the experimental conditions is given in the Supplemental materials.

## Ussing chamber experiments

Ussing chamber experiments were performed as described previously[70]. For further details, see the Supplemental materials section.

## Immunoblot analysis

For immunoblot analysis of ChAT expression in the tracheal epithelium of *ChAT*$^{fl/fl}$:*Trpm5*$^{cre}$ mice and wild-type controls, sample preparation as well as immunoblot analysis were performed as has been reported previously[71]. Further details are described in the Supplemental materials section.

## In vivo tuft cell stimulation

By performing a minimal invasive surgery, the upper part of the trachea was exposed and either vehicle or 1 mM denatonium (Sigma-Aldrich) or 100 μM CNO (HelloBio) were applied through the median cricothyroid ligament. Animals were sacrificed 24 h, 72 h or 7 days after denatonium inhalation by an overdose of ketamine/xylazine. Tracheal sections were used for immunohistochemistry while tracheae, lungs, bronchoalveolar lavage fluid, blood and lymph nodes were isolated for FACS experiments as described in the Supplemental materials.

## Infection of mice with *P. aeruginosa*

Mice were infected with the mucoid *P. aeruginosa* strain NH57388A (provided by Niels Hoiby, Department of Clinical Microbiology, Rigshospitalet, University of Copenhagen, Denmark) as described previously[4]. For further details, please see the Supplemental materials.

## FACS analyses

Lungs, tracheae, blood, bronchoalveolar lavage fluid and airway lymph nodes from CNO or from denatonium treated or *P. aeruginosa*-infected mice were collected and processed as described previously[4]. A description of the organ collection, preparation, FACS analyses and the antibodies used is given in the Supplemental materials.

## Cytokine measurements

Plasma cytokine IL-17A was measured using luminex discovery assay (mouse premixed multi analyte kit, R&D systems, Minneapolis, MN, USA) according to the manufacturer's protocol. For further details see the Supplemental materials.

## Isolation of mouse lung dendritic cells and tracheal epithelium

*Trpm5*$^{+/+}$ and *Trpm5*-DREADD mice were euthanized by an overdose of ketamine/xylazine. Lungs of *Trpm5*$^{+/+}$ mice and tracheae from *Trpm5*$^{+/+}$ or *Trpm5*-DREADD mice were collected and processed to obtain single cell suspensions for the FACS analyses. A detailed description of the FACS procedure, the antibodies, the gating strategy, the co-culture of tracheal epithelial cells and DCs and the experimental conditions can be found in the Supplemental materials.

## Phagocytosis assay

Bone marrow-derived dendritic cells (BM-DCs) were isolated from *Trpm5*$^{+/+}$ mice, cultured for 11 days, infected with the *P. aeruginosa* strain NH57388A, treated with tracheal supernatants obtained under different stimulating conditions, lyzed to harvest phagocytosed bacteria and evaluated for CFU. A detailed description of the procedure is given in the Supplemental materials.

## Migration assay

BM-DCs were isolated and allowed to proliferate followed by application of a linear scratch. Then BM-DCs were incubated with tracheal supernatants stimulated under different experimental conditions and pictures were taken at the beginning of each experiment and after 24 h. A detailed description can be found in the Supplemental materials.

## Quantitative RT-PCR

Experiments were performed as described previously[71]. A detailed description of the quantitative RT-PCR of *Ccl21b* and *Ccl19* can be found in the supplement.

## Statistical analyses

Each experiment included at least three biological replicates. For histological evaluation, a minimum of three animals were included, with multiple sections or samples per animal analyzed. The exact sample numbers and statistical tests are specified in the respective figure legends. Each experiment was repeated at least three times. For the quantification of DCs and tuft cells in tissue sections the cells were counted manually based on their specific fluorescence using a fluorescence microscope. For tracheae, five sections covering a distance of 1 mm (200 μm distance between each section) were evaluated. In Ussing chamber experiments, all values were calculated over 1 cm$^2$ tissue area. Statistical analyses were performed with GraphPad Prism 10.2.2 (GraphPad software Inc., La Jolla, CA, USA). Data are depicted as mean ± standard error of the mean with single values and an n-number of measured tracheae or cells. Shapiro-Wilk test and Kolmogorov-

Smirnov test were used to assess normal distribution of the data. Normally distributed data sets with more than two groups were then analyzed for significant differences with a one-way ANOVA and non-normally distributed data with a Kruskal-Wallis test both followed by post-hoc analyses. Direct comparisons between two groups were performed using the unpaired Student's $t$-test for normally distributed data and with the Mann-Whitney test for non-normally distributed data ($\alpha = 0.05$). Paired samples were analyzed with the paired Student's $t$-test for normally distributed data and with the Wilcoxon test for non-normally distributed data. For correlation analysis, Pearson's correlation coefficient was calculated, and a linear regression was performed. All data points represent measurements from distinct samples. The level of significance was set at $p \leq 0.05$.

### Reporting summary

Further information on research design is available in the Nature Portfolio Reporting Summary linked to this article.

## Data availability

Data are available in the article and its Supplementary files or from the corresponding author upon request. Source data are provided with this paper.

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

## Acknowledgements

The authors would like to thank Nora Aouragh, Alexander Grißmer, Aline Herges, Praveen Kumar, Andrea Rabung, Alaa Salah and Maxim Zimmer (Institute of Anatomy and Cell Biology, Saarland University, Homburg) and Joanna Zaisserer and Anna Erbacher (Walter-Straub-Institute for Pharmacology and Toxicology, LMU Munich) for excellent technical assistance. The authors would like to thank Gudrun Wagenpfeil (Institute for Medical Biometry, Epidemiology and Medical Informatics, Saarland University, Homburg) for help with sample size estimation. The software Inkscape version 0.92.3 was used to create Figs. 6K, 7F and Supplementary Fig. S5A. This study was supported by the German Research Foundation (DFG) SFB TRR152 grants P22 to GKC, P11 to UB, Z02 to VF and UB and P15 to VC and TG, SFB 894 P2 to UB as well as by DFG KR 4338/1-2 to GKC and DFG ID79/2-2 to SI, a Hom-For2022 and a HomFor2024 grant from Saarland University to MIH and a HomFor2023 Nachwuchsförderung and HomForexzellent2023 grant from Saarland University to MIE. TG was supported by Research Training Group 2338 (DFG). BJC was supported by a grant (R21 ES036349) from the National Institutes of Health (NIH, Bethesda, Maryland).

## Author contributions

N.A.W., M.I.H., M.I.E., N.Z., C.E., S.B.E., M.S., S.M., C.H. performed the experiments and analyzed the data. A.K.B., A.W., S.K., D.R., A.B., R.B., C.H., D.Y., B.J.C., M.B., V.F., T.G., V.C., C.M., E.K., S.I., S.L.B., U.B. provided experimental tools and the transgenic animals. N.A.W., M.I.H., M.I.E., N.Z., C.E., U.B., and G.K.C. wrote the manuscript. N.A.W., M.I.H., M.I.E., N.Z., C.E., S.B.E., S.M., G.K.C. interpreted the data. G.K.C. conceived and designed the study. All authors approved of the final version of the manuscript.

## Funding

## Competing interests
The authors declare no competing interests.

## Additional information

[1]Institute of Anatomy and Cell Biology, Saarland University, Homburg, Germany. [2]Center for Gender-Specific Biology and Medicine (CGBM), Saarland University, Homburg, Germany. [3]Experimental Pharmacology, Center for Molecular Signaling (PZMS), Saarland University, Homburg, Germany. [4]Department of Internal Medicine V-Pulmonology, Allergology, Intensive Care Medicine, Saarland University Hospital, Homburg, Germany. [5]Cell and Developmental Biology, Center of Human and Molecular Biology (ZHMB), Saarland University, Faculty of Medicine, Homburg, Germany. [6]Department of Cellular Neurophysiology, Center for Integrative Physiology and Molecular Medicine (CIPMM), Saarland University, Homburg, Germany. [7]Institute for Experimental and Clinical Pharmacology and Toxicology, Center for Molecular Signaling (PZMS), Saarland University, Homburg, Germany. [8]Department of Biomedical Sciences, School of Medicine, Nazarbayev University, Astana, Kazakhstan. [9]Institute for Medical Microbiology and Hygiene, Saarland University, Homburg, Germany. [10]The Johns Hopkins Asthma and Allergy Center, Baltimore, MD, USA. [11]Walther-Straub Institute of Pharmacology and Toxicology, LMU Munich, Munich, Germany. [12]Comprehensive Pneumology Center, a member of the German Center for Lung Research (DZL), Munich, Germany. [13]Helmholtz Institute for Pharmaceutical Research Saarland (HIPS), Helmholtz Centre for Infection Research (HZI), Saarbrücken, Germany. [14]These authors contributed equally: Noran Abdel Wadood, Monika I. Hollenhorst, Mohamed Ibrahem Elhawy. ✉e-mail: Gabriela.Krasteva-Christ@uks.eu

