## [Transparent Peer Review file · Nature Communications]

Tracheal tuft cells release ATP and link innate to adaptive immunity in pneumonia

Corresponding Author: Professor Gabriela Krasteva-Christ

Version 0:

Reviewer comments:

Reviewer #1

(Remarks to the Author)

Tracheal tuft cells (TC) have recently been demonstrated to shape immune responses in the airways. While some of the downstream immune response events after TC activation have been attributed to differential release of either acetylcholine, leukotriene C4 or interleukin (IL)-25, depending on the activating stimulus, the molecular mechanisms underlying the TC-mediated recruitment and activation of immune cells are incompletely understood. In this manuscript Krasteva-Christ G. and team observed that *Pseudomonas aeruginosa* infection activates murine TC, which releases ATP via pannexin 1 channels. Taste signaling through the Trpm5 channel was found to be essential for bacterial TC activation and ATP release. The authors further demonstrate that activated TCs recruit dendritic cells in the trachea and lung. Subsequent dendritic cell activation, phagocytosis and migration was revealed to be dependent on ATP released by TCs. In addition the authors present further evidence that stimulation of TCs also impinges on the adaptive immune system by recruiting IL-17A secreting T helper cells. In summary, this study provides a novel molecular framework of how tuft cells shape both innate and adaptive immune responses in the airways to combat bacterial infection.

General remarks

This is a very well designed and conducted study detailing novel mechanistic insights and in vivo relevance of how tracheal tuft cells release pro-inflammatory ATP and subsequent innate and adaptive immune activation. All major claims are well substantiated by the data presented.

Specific comments

Some minor adjustments and clarifications would help to contextualize the findings and improve the clarity of the study.

- a) While the finding that TC release ATP is intriguing and clearly associated to increased pulmonary cellular infiltrate it would be necessary to better contextualize how ATP release synergizes with known chemokine gradients for the cellular increases observed. A better contextualization would improve the clarity and strength of the manuscript.
- b) the introduction is a bit incoherent in its flow – it is elusive what the hypothesis was put forward that led the authors to study ATP release. Some modifications would make the flow more coherent.
- c) Many tools and methods used are not explained and contextualized in the flow of the manuscript (especially results section); e.g. why using denatonium, why this dose, why pannexin1-deficient mice,; while this maybe be obvious to TC experts, it might not be that clear to the broader immunological and mucosal immunology audience. As such it would be favorable to better integrate short justifications/explanations throughout the manuscript.
- d) the authors discuss a pulmonary source of IL-25 but do not cite the studies that highlight that.
- e) the increase in Th17 cells in the lungs is striking; however, this begs the question are other IL-17A/F-producing cell types elevated/changed in an TC-ATP-specific manner? If the authors have evidence for ILC3 or gd-T cell/IEL it would be favourable to include.
- f) Figure A: scale bars in A-C are missing; Statistics in F&H is strangely added and should be revised;
- g) Figure 3-5: gating strategies of analyzed pulmonary cell populations should be added to supplement. Data of cell frequencies should be also shown as absolute numbers but also as # of total CD45+ cells.
- h) Figure 6: Activation of DC is multi-layered – are other activation markers of DC elevated?

Reviewer #2

(Remarks to the Author)

The paper by Wadood et al addresses the function of tuft cells in shaping the immune response in the upper respiratory tract. The authors propose a model whereby taste signaling via Trpm5 would result in ATP release by tuft cell with subsequent dendritic cell activation, phagocytosis and migration. Tuft cell activation via Trpm5 would also affect the adaptive branch of the immune system by recruiting IL-17A secreting CD4 T cells to respond to bacterial infection. The paper convincingly shows how wildtype tuft cells release ATP following stimulation with denatonium in vitro as well as in tracheal supernatants from CNO-activated Trpm5-DREADD mice. ATP release by tuft cells was shown to depend on pannexin 1, which colocalized with Trpm5 in tuft cells. Trpm5-induced K⁺ efflux underlies pannexin 1 dependent currents.

Comment 1

It would be interesting to know if the activity of the ATP-gated P2X7 receptor is involved in K⁺ efflux, a point that could be addressed by pharmacological inhibition of P2X7R.

The authors used Trpm5-DREADD mice to show intracellular increases in Ca²⁺ in tuft cells and recruitment of neutrophils in BALF and trachea following CNO inhalation. Monocytes (but not macrophages) and DCs were increased as well.

Comment 2

It could be interesting to address whether a switch between interstitial (IM) and alveolar (AM) macrophages (CD64⁺ and CD170⁺, respectively) has occurred. With unaltered absolute counts, there might be an increase in AM and a corresponding decrease in IM.

In vivo inhalation of denatonium was used to show the expansion of tuft cells and increases of DCs in the trachea of Trpm5^{+/+} but not Trpm5^{-/-} mice at 24 h after stimulation. However, at 72 h CD11c⁺ cells were significantly increased also in Trpm5^{-/-} cells. A positive correlation was shown between the increase of TC and DCs in the trachea after denatonium inhalation.

Comment 3

How would the authors explain the increase of DCs in Trpm5^{-/-} mice at 72 h after denatonium stimulation? Also, I would tone down the final statement of the paragraph: "These data suggest that immune cell recruitment depend on TC number and activation"

Trpm5 was shown to contribute to the increase of DCs in mice infected with *P. aeruginosa*. Notably, Trpm5^{-/-} mice infected with *P. aeruginosa* showed significantly impaired survival with respect to Trpm5 proficient animals. A less pronounced decrease in mice survival was observed with Panx1^{-/-} mice. Experiments in Trpm5 and Panx1 proficient vs deficient mice treated with denatonium led the authors hypothesizing that Trpm5 dependent ATP release promoted the recruitment of Th17 cells in the lungs.

Comment 4

Did you check for differences in Th17 cells in Trpm5^{+/+} vs Trpm5^{-/-} mice infected with *P. aeruginosa*?

IL-17A concentrations were shown to be elevated in the plasma of denatonium-treated Trpm5^{+/+} mice compared to controls but not in plasma of Trpm5^{-/-} mice

Comment 5

Variations of gamma/delta T cells in the lung after denatonium inhalation should be investigated. They represent a substantial proportion of lymphocytes in the lung and are the major producers of IL-17A at early stage of infections (doi:10.1111/imm.12764). Furthermore, gd T cells are sensitive to ATP.

Upregulation of CD86 in either wildtype or Panx1^{-/-} DCs cocultured with TCs from Trpm5-DREADD mice stimulated with CNO showed that DCs activation was dependent on Trpm5-dependent ATP release by TCs. MHCII upregulation was also shown, however, it was abolished by mecamlamine and atropine (in addition to apyrase), thereby complicating the interpretation of Trpm5 dependent DC activation.

Comment 6

Contour plots in fig 6 are difficult to compare and more details on panel A-F should be provided in the figure legend. Please provide the gating strategy for all flow cytometry experiments.

Finally, in vitro phagocytic assays with DCs infected with *P. aeruginosa* revealed the enhanced phagocytic activity of DCs treated with tracheal supernatants from Trpm5^{+/+} vs Trpm5^{-/-} mice treated with denatonium. Analogously, DCs migration was shown to be enhanced by tracheal supernatants from Trpm5 proficient mice.

General comment

A major issue I see in this study is the lack of connection of tuft cell activation with an antigen specific response and the possible polarization of a CD4 T cell response toward the Th17 program, if I understood correctly the hypothesis of the paper. Alternatively, the mere recruitment of Th17 by TC stimulation via Trpm5 activation seems an "inappropriate"

immunopathological stimulus. It is clear that Trpm5 mediated ATP release by TCs can condition both innate immune cells and enrich Th17 cells in the lung. However, these evidences are difficult to integrate in the physiology or pathophysiology of an adaptive immune response.

Minor comments

1. Please provide details on the ELISA test used for ATP measurements
2. The legend of Fig 1 J-M should be more detailed and arrows explained
3. In fig S3C, statistical significance seems impossible (please detail which test was used)

Reviewer #3

(Remarks to the Author)

The manuscript by Abdel Wadood and collaborators provides a molecular mechanism initiated by Tuft cells to build an innate and adaptive immune response by the airway epithelium in response to *Pseudomonas aeruginosa* infection. The activation of the bitter taste sensor TRPM5 in tuft cells triggers Panx1-dependent ATP release which lead to dendritic cells activation and recruitment to the trachea and the lung. The strength of the manuscript is the use of a variety of in vivo and ex vivo models to support this concept. Although the amount of work is appreciated, the combination of so many models makes some figures difficult to interpret. Importantly, appropriate controls are not always shown and some experimental conditions are not well justified. This critically undermines some of the conclusions.

Major comments:

- 1) The introduction should bring a clear justification of evaluating ATP as mediator of Tuft cells' response. The links between TasR, TRPM5 and ATP release by CALHM or Pannexins is indeed well described in taste buds.
- 2) The statistical analysis is sometimes difficult to understand. For instance, how the statistical significance is reached in Fig. S3C after Suramin treatment with such a large variability? In contrast in Fig. 3C the difference between sham and control is not significant? The statistical tests used should be indicated in the figure legends. Are the bars showing SD or SEM? Moreover, n should be indicated for each experimental condition (and not showing only the higher n...).
- 3) According to Methods (ATP sensor experiments and ATP measurements paragraphs), ATP was determined by ELISA from homogenized tracheas stimulated with denatonium. Thus, one can understand that ATP in supernatants from homogenized trachea mostly corresponds to intracellular ATP. Please explain how intracellular ATP is excluded or revise Methods to avoid confusion. How ATP release from tracheas is normalized? Can you estimate the ATP concentration in these supernatants? The number of TCs in tracheas is likely to be different because of the differential TCs expansion described in Fig. S5.
- 4) X-gal staining is poor and difficult to interpret. Without quantification/scoring and with only a single representative image provided, the interpretation of these data is ambiguous. In situ hybridization could be useful.
- 5) The immunofluorescence in Fig. S2A2 is not convincing. Validation of the antibody by IF on Panx1,2 KO mice should be shown. Please explain the endogenous eGFP fluorescence in all tracheal cells in Fig. S2A1?
- 6) Fig.3: how immune cells were detected? FACS? The methodology is not clearly described (BALF is also not described in Methods...). The writing can be improved to help the readers. How neutrophil counts were normalized in the tracheas?
- 7) Important controls are missing to support the data in Figs. 4 and 5. Please add non-treated conditions for panels B/C and the control condition without *P. aeruginosa* for Panx1-/- in panels C/F.
- 8) L.291-295: sentences hard to read.
- 9) The DC migration assay is not convincing. DCs treated with tracheal supernatants from different mice cannot be compared to non-treated DCs! The migration phenotype induced by each supernatant should be compared to its own control genotype! Is there any impact of the different supernatant on the proliferation of DCs? Why different confluences are observed between experiments? An impact on the proliferation could explain differences in wound repair.
- 10) A more detailed explanation on Tuft cells' expansion, and how it could be regulated by TRPM5, should be added in the Discussion.

Minor comments:

- 1) Better images should be shown in Fig. 1A-C. eGFP fluorescence should be visualized in Fig1B-C. The data do not show that ATP released by TCs is responsible for Ca²⁺ increase but point only to a correlation. Data using TCs from Panx1-/- and Trpm5-/- mice could be used.
- 2) Only 13 cells were analyzed in Fig. 1F?
- 3) Brush cells should be replaced by tuft cells for nomenclature coherence through the text (legends of Fig. S2 and S4).
- 4) Patch-clamp quantification should be shown for Fig. 1D. Please normalize the current intensity to the capacitance (pA/pF). Differences in cells' size (membrane surface) may explain the variability observed in Fig. 2C.
- 5) The closing sentence in L.153-154 is an overstatement as the Ca²⁺ data are not clear cut.
- 6) It is surprising that a small proportion of Tuft cells can release sufficient amount of ATP, despite the dilution imposed by the Ussing system, to activate purinergic-induced I_{sc} currents. Please show traces of the recordings.
- 7) Line 171-172: this sentence does not seem at the right place!
- 8) Fig. S4: how TCs were identified, as they were not eGFP-positive? Please clarify how Ca²⁺ recording was made as not only TCs are likely monitored! The CNO concentration is different from the one mentioned in the methods (60 μ M vs 100 μ M).
- 9) Why TRPM5-DTA mice were not described or used before. This model would have been useful to support previous conclusions.
- 10) The loading control (GAPDH or b-actin) is missing in Fig. 6A.

11) The scenario for the resolution of inflammation does not sound realistic.

Reviewer #4

(Remarks to the Author)

Version 1:

Reviewer comments:

Reviewer #1

(Remarks to the Author)

The authors addressed all my comments and queries and significantly improved the manuscript. All claims are justified by the data. I recommend to accept the manuscript for publication in its current form.

Reviewer #2

(Remarks to the Author)

The authors carefully and thoroughly addressed all my concerns.

Reviewer #3

(Remarks to the Author)

This Reviewer thanks the authors for the solid revision of their manuscript. There is, however, some major and minor points that still need to be addressed. They are listed below with the sequence of the manuscript.

1) Abstract, l.49: "stimuli".

2) Fig.1A: thank you for normalizing the current in the I-V shown in A (and some other Figures). The point of normalization is to compare I-V relations in TCs with different genotypes for quantitative analysis. Please correct Fig.2E, 2F and 2G.

3) L.127-129: It is stated that Ca²⁺ changes are recorded in the ATP sensor cells while performing patch-clamp experiments on TCs. I guess you mean recording of FOF1-ATPase fluorescence changes upon ATP binding, not Ca²⁺. Please explain or correct. The legend of Fig.1E is also misleading. An additional control/representative image using trpm5^{-/-} TCs should be shown. The quality of the images is very low, so that the fluorescence at the membrane is barely visible.

4) Legend of Fig.1H: there are no marked cells in C). Does the trace represent the average of two recordings? Data are shown as F1/F0 while they are expressed as deltaF in supplemental Fig.S2. Which one is correct as it does not mean the same. Does time 0 represents when the whole-cell configuration was established in TCs? The legend needs to be improved or remove panel H.

5) Line 169-172. Can you speculate on how 40% of TCs expressing Panx1 lead to such an increase in ATP release? Is there any evidence that Panx1 expression is regulated by denatonium?

6) Supplemental Methods, ATP sensor experiments, l.10: what means exactly tracheas were homogenized? In the reply to Reviewer 3, you state that tracheas are not homogenized for extracellular ATP measurement. Are they or not homogenized? Please, clarify.

7) L.136-138: I believe that tracheal supernatants were not treated with denatonium but tracheas were, and supernatants collected. Please correct sentence.

8) It is stated that activation of tuft cells leads to trpm5-dependent K efflux, resulting in depolarization of the plasma membrane and opening of Panx1, independently of P2X7 receptor activation (answer to Reviewer 2). If I am correct, K efflux should lead to hyperpolarization, not depolarization of the cell membrane, so this mechanism does not hold. The link between Trpm5 and Panx1 is likely more complex than proposed here. Please explain.

9) The synergy between Trpm5 and Panx1 is not fully supported by the experiment shown in Fig. 2G. Why 2.4 μM intracellular Ca²⁺ was applied here instead of 1 μM, as used in Fig.1B? How can you distinguish between Trpm5 and Panx1-mediated currents using this protocol? Why CBX reduced the current in low calcium and probenecid only in high calcium? Better contextualization and interpretation of these experiments are needed.

10) Fig.S5 is bringing even more confusion. Denatonium-induced I_{sc} integrates responses from many channels and transporters, including Trpm5 and Panx1. Probenecid will block Panx1 current, and thus denatonium-induced I_{sc} will be smaller; this does not imply (and does not exclude either) a paracrine effect of released ATP on purinergic receptors.

Similarly, the effect of suramin on denatonium-induced Isc does not imply that extracellular ATP comes from Panx1 channel. These data using whole-mount tracheal preparations, which are nice by the way, do not confirm that ATP is released via pannexin channel after TC stimulation, as stated in I.206-207. The data are consistent with the idea that ATP released via pannexins acts in a paracrine manner. I would suggest toning down some of the conclusions reported in this paragraph.

11) Fig.S3: Panel B is not convincing at all! If the shadow fluorescent signal in WT is considered positive for ChAT, why it is not in Chatfl:Trpm5^{-/-} tracheas? I can see very similar background fluorescence.

12) The gating strategy for flow cytometric (please correct spelling in legend, as well as in legends of Fig.S11 and S12) identification is shown in Fig.S8; it should be shown as Fig.S7 to be coherent with the Figure numbering in the text (Fig.S8 is mentioned after Fig.S6 and before Fig.S7).

13) Fig.3 and Fig.4: Please explain how the data are normalized as counts per "trachea" or per "lung".

14) Controls for Fig. 5J are missing.

15) Fig.S7: The effects of denatonium at 24h, 72h and 7d were compared to vehicle at 24h only. This is not correct and matched controls should be used.

Please speculate on the mechanisms by which TCs activation by denatonium induces their intrinsic expansion within 24h. DCAMKL1 and CD11c legends are shown in black then not visible on the images!

16) Reply to Reviewer 3 regarding TRPM5-DTA mice. At least, the effect of stimulation of TRPM5-DTA tracheas with denatonium on ATP release should be shown.

17) Thank you for sharing additional experiments on DC proliferation. Since the representative images are not very clear and some wounds even enlarge over time, controls should be presented to support the migration data. Maybe a line in Methods can be added to exclude the possibility of DC proliferation.

Please align the positioning of bottom legends (Trpm5^{+/+...}) in Fig. 7E to the corresponding conditions.

Reviewer #4

(Remarks to the Author)

Version 2:

Reviewer comments:

Reviewer #3

(Remarks to the Author)

All my concerns were adequately addressed.

Reviewer #4

(Remarks to the Author)

Point-by-point response to the reviewers' comments

We are very grateful for the constructive reviewers' comments. We have addressed all reviewers' comments. In the last months, we have performed many additional experiments and the obtained results add substantial information that strengthens our concept and previous findings. The new experiments include additional patch clamp experiments, FACS control experiments, *in vivo* and *ex vivo* investigation of activation of DCs, especially regarding the activation marker CCR7 and the chemokines Ccl21b and Ccl19 following TC activation. We have also addressed the possibility of alternative IL-17-producing cells other than T_H17 cells, namely $\gamma\delta$ -T cells and ILC3. Additionally, we have changed the introduction part of the manuscript to improve the cohesiveness, as suggested by reviewer 1 and 3. The new results added substantial information to our manuscript and strengthen our hypothesis. They are presented now in the following figures and tables: Figures 1A, B, G, 2H, 4, 5 and 7, Supplementary Figures 1, 2, 4, 5, 6, 7, 8, 9, 10, 11, 12, 14, 16 and 17, and are discussed extensively. We hope that our revision will meet your approval and can now be accepted for publication in *Nature Communications*.

Reviewer #1 (Remarks to the Author):

Tracheal tuft cells (TC) have recently been demonstrated to shape immune responses in the airways. While some of the downstream immune response events after TC activation have been attributed to differential release of either acetylcholine, leukotriene C4 or interleukin (IL)-25, depending on the activating stimulus, the molecular mechanisms underlying the TC-mediated recruitment and activation of immune cells are incompletely understood. In this manuscript Krasteva-Christ G. and team observed that *Pseudomonas aeruginosa* infection activates murine TC, which releases ATP via pannexin 1 channels. Taste signaling through the Trpm5 channel was found to be essential for bacterial TC activation and ATP release. The authors further demonstrate that activated TCs recruit dendritic cells in the trachea and lung. Subsequent dendritic cell activation, phagocytosis and migration was revealed to be dependent on ATP released by TCs. In addition the authors present further evidence that stimulation of TCs also impinges on the adaptive immune system by recruiting IL-17A secreting T helper cells. In summary, this study provides a novel molecular framework of how tuft cells shape both innate and adaptive immune responses in the airways to combat bacterial infection.

General remarks

This is a very well designed and conducted study detailing novel mechanistic insights and *in vivo* relevance of how tracheal tuft cells release pro-inflammatory ATP and subsequent innate and adaptive immune activation. All major claims are well substantiated by the data presented.

Reply: We thank the reviewer for his comment.

Specific comments

Some **minor** adjustments and clarifications would help to contextualize the findings and improve the clarity of the study.

a) While the finding that TC release ATP is intriguing and clearly associated to increased pulmonary cellular infiltrate it would be necessary to better contextualize how ATP release synergizes with known chemokine gradients for the cellular increases observed. A better contextualization would improve the clarity and strength of the manuscript.

Reply: We thank the reviewer for this valuable comment. We agree, that it would be very interesting to know more about chemokine gradients which play a role for the recruitment of dendritic cells and the T cells in our experimental setting. As suggested, we performed additional experiments, in which we investigated the CCR7 receptor, an activation marker for dendritic cells to which the chemokine CCL21 and CCL19 can bind and included the results in the revised version of the manuscript (revised Fig. 7 A-C and Supplementary Fig. 18). CCL21 is a well-known chemokine from a concentration gradient that guide the migration and recruitment of antigen-presenting dendritic cells and naïve T-cells that express the chemokine receptor CCR7 to the lymph nodes, thereby priming the immune responses (Forster, Davalos-Misslitz, & Rot, 2008, PMID: 18379575). In particular, CCR7-dependent DC migration from peripheral tissues to lymphoid tissues (Riol-Blanco et al., 2005, PMID: 15778365, Liu et al., 2021, PMID: 34302064) is essential for host defence against pathogens and immune tolerance of harmless self- or nonself-antigens (Iwasaki & Medzhitov 2015, PMID: 25789684; Liu et al., 2021, PMID: 34302064). Dendritic cells (DCs) have been reported to upregulate CCR7 expression and acquire migratory properties when exposed to extracellular ATP, prompting us to investigate whether airway DCs would exhibit the same effect following tuft cell activation and subsequent ATP release (La Sala et al., 2002, PMID: 11861288; Sáez PJ et al., 2017, PMID: 29162744). We observed in co-culture experiments of dendritic cells and tracheal tuft cells that CCR7+ dendritic cell numbers increased after tuft cell stimulation. Furthermore, dendritic cell numbers in the lungs and the tracheae increased after *in vivo* stimulation of tuft cells followed by elevated numbers of dendritic cells in the draining lymph nodes. Indeed, dendritic cell migration was enhanced after stimulation with supernatants collected from stimulated tracheal tuft cells. The increase in CCR7+ dendritic cell numbers was ATP-dependent and depended on activation of tuft cells, thus strongly indicating a direct relation or rather synchronisation between tuft cell activation and ATP-dependent recruitment of dendritic cells in the trachea and lung. The increased number of CCR7+ dendritic cells in the trachea, the lung and in the lymph nodes (please see revised Fig. 7A-C and Supplementary Fig. 18) is suggestive for a CCL21 gradient, a CC motif chemokine mainly involved in migration of dendritic cells (Summers et al., 2022, PMID: 35322079). As part of the revision, we investigated the levels of this chemokine by qPCR. We found that *Ccl21b* expression increased robustly in tracheae and lungs of *Trpm5*^{+/+} after tuft cell stimulation but not in *Trpm5*^{-/-} mice (Supplementary Fig. 19A, B). Moreover, CCR7 activation on dendritic cells plays an important role for the dendritic cell-induced T cell differentiation (Probst Brandum et al., 2021, PMID: 34361107, Liu et al., 2021, PMID: 34302064), thereby linking the increased numbers of dendritic cells to the observed increase and activation in T_H17 cells in our study. Furthermore, we detected an increased expression of *Ccl19* in the airway draining lymph nodes obtained from the wild-type mice three days after the intratracheal application of 1 mM denatonium (Supplementary Fig. 19C). CCL19 is a ligand of CCR7 allowing homing of dendritic cells from peripheral organs, e.g., lungs to secondary lymphoid organs, e.g., lymph nodes (Yan et al., 2019, PMID: 31632965). The

critical role of CCL19 was demonstrated by the failure of dendritic cells to migrate to lymph nodes when the CCL19 was antagonized in the skin (Robbiani et al., 2000, PMID: 11114332). Our findings clearly emphasize the importance of the CCL19-CCR7 axis, activated downstream of tracheal TC activation, in the recruitment and homing of dendritic cells.

Fig. 7: Tuft cell (TC)-induced dendritic cell (DC) phagocytic activity and migration is dependent on ATP. **(A)** Co-culture of CNO-stimulated *Trpm5*-DREADD tracheal epithelium (TE) with DCs significantly increased the percentage of CCR7-expressing DCs after 4h (n=6) and 24h (n=7) compared to CNO-treated *Trpm5*^{+/+} TE co-culture with DCs (n=4-5). The increase in % of CCR7⁺ DCs after 24h was abolished by apyrase (5 U/ml) (n=5). DCs cultured alone and treated with CNO (n=3) exhibited the same percentage of CCR7-expressing cells as DCs co-cultured with *Trpm5*^{+/+} TE. One-way ANOVA. **(B)** The number of CCR7-expressing DCs in lungs increased significantly 3 days after intratracheal TC stimulation with 1 mM denatonium in *Trpm5*^{+/+} mice in contrast to *Trpm5*^{-/-} mice (n=4). One-way ANOVA. **(C)** In *Trpm5*^{+/+} mice (n=6) CCR7-expression DC counts in lymph nodes are increased 3 days after denatonium treatment, but not in *Trpm5*^{-/-} mice (n=4). Kruskal-Wallis test.

Fig. S18: FACS analyses of CCR7⁺ DC numbers in tracheae from *Trpm5*^{+/+} and *Trpm5*^{-/-} mice revealed an increase in CCR7⁺ DCs three days after treatment with denatonium in *Trpm5*^{+/+} but not *Trpm5*^{-/-} mice. (n = 3-4)

Fig. S19: Relative expression of *Ccl21b* in whole tracheae (A) and lungs (B) of *Trpm5*^{+/+} and *Trpm5*^{-/-} mice 3 days after the intratracheal application of 1 mM denatonium (n=3, each point represent the average of two technical replicates). (C) Relative expression of *Ccl19* in airway draining lymph nodes (LN) of *Trpm5*^{+/+} and *Trpm5*^{-/-} mice 3 days after the intratracheal application of 1 mM denatonium (n=3, each point represent the average of two technical replicates). (* p<0.05, p **<0.01, *** p<0.001; one-way ANOVA).

b) the introduction is a bit incoherent in its flow – it is elusive what the hypothesis was put forward that led the authors to study ATP release. Some modifications would make the flow more coherent.

Reply: We apologize for this incoherence and thank the reviewer for his/her/their comment. We are happy to provide more detailed information about the rationale which led us to study the hypothesis of ATP release from TC in the introduction part of our manuscript (please see line 105-107). Additionally, we have added more information about ATP release and pannexin 1 function in taste cells. We hope that the flow of information is more coherent. The text reads now as follows (line 88-94):

In support of this hypothesis, TCs possess similar signaling mechanisms with type II taste cells including expression of a functional bitter taste signaling cascade and release ACh, which acts on nearby nerve fibers^{4,5,7,11,12}. In taste buds chemosensory cells activate the bitter taste signaling cascade triggering the release of ATP through calcium homeostasis modulator 1/3 (CALHM1/3) channels and eventually binding to purinergic receptors on sensory nerves that innervate type II taste cells, thus conveying taste sensation to the nervous system^{11,13-15}. While ATP plays a clear role in sensory signaling in the taste buds, no direct immunological function has been attributed to its release in this context.

c) Many tools and methods used are not explained and contextualized in the flow of the manuscript (especially results section); e.g. why using denatonium, why this dose, why pannexin 1-deficient mice,; while this maybe be obvious to TC experts, it

might not be that clear to the broader immunological and mucosal immunology audience. As such it would be favorable to better integrate short justifications/explanations throughout the manuscript.

Reply: We thank the reviewer for this comment. We followed his/her/their suggestion and included more details on the study design and background in the results section of our manuscript. We specifically addressed the examples mentioned by the reviewer. Please find changes made in the revised version of the results section:

line 135: ... with 1 mM denatonium - a taste receptor agonist specifically acting on TCs⁴,...

line 142-145: To address the ATP release mechanism in TCs, we studied the role of pannexin 1 (Panx1) channels. Panx1 channels have previously been identified as ATP release channels in various cell types, including taste cells²³⁻²⁵. More intriguingly, single-cell RNA-seq analyses from our group and others^{5,18,26} have suggested *Panx1* expression in TCs^{5,18,26}.

Line 150-154: This newly generated mouse model was validated by immunoblot and immunohistochemistry experiments and used throughout our study in order to distinguish between the effects induced by ATP and ACh, specifically to assess the potential involvement of TC-released ACh in the observed ATP release (Supplementary Fig. 3).

We hope that the rationale behind our experiments and their design is now clear and convincing to the readership.

d) the authors discuss a pulmonary source of IL-25 but do not cite the studies that highlight that.

Reply: We apologize that we missed the citation of some studies, which described tuft cells as a source of IL-25 in the airways. In the original version of the manuscript, we cited in the introduction a study from Ualiyeva et al. (2021, PMID: 34932383) that describes the role of IL-25 originating from tuft cells for the regulation of lung type 2 inflammation. Unfortunately, we overlooked the incorporation of the original study from the same group describing for the first-time mouse tracheal tuft cells as a source of IL-25 in the airways (Bankova et al. 2018, PMID: 30291131). We have now included this citation in the revised version of the manuscript (lines 101, 145, 199, 414, 417, 469, 474). Additionally, we cite a study by Kohanski et al. (2018, PMID: 29778504), who showed that human nasal tuft cells are a prominent source of IL-25. We also included the information that mouse nasal tuft cells have been shown to have higher transcript levels of IL-25 than tracheal ones (Ualiyeva et al. 2020, PMID: 31953256). The respective sentence in the introduction section reads now as follows (line 99-101): "TCs are the source of IL-25 in tracheal epithelium of mice and upper airways of humans, with nasal SCCs in mice exhibiting higher *Il25* transcription levels than tracheal TCs^{16,18,19}."

e) the increase in Th17 cells in the lungs is striking; however, this begs the question are other IL-17A/F-producing cells types elevated/changed in an TC-ATP-specific manner? If the authors have evidence for ILC3 or gd-T cell/IEL it would be favourable to include.

Reply: We appreciate this interesting comment that helped to improve the manuscript. We agree with the reviewer that it is of interest to analyse the $\gamma\delta$ -T and ILC3 cell numbers and their dependency on ATP released upon tuft cell stimulation. Therefore, we performed additional *in vivo* experiments followed by FACS analyses and included the results in the

revised version of Supplementary Fig. 16D, E and F. We did not observe changes in the number of $\gamma\delta$ -T and ILC3 cells in wild-type mice three days after intratracheal treatment with denatonium. We also included additional data of infection experiments investigating T_H17 cell counts in the lung (Supplementary Fig. 16C). In these experiments we observed an increase in T_H17 cells in the lungs of wild-type mice two days after infection, which was not the case in *Trpm5*^{-/-} mice, and therefore tuft cell specific. Taken together, due to the unchanged number of $\gamma\delta$ -T and ILC3 cells, it appears more likely that T_H17 cells are primarily responsible for the elevated IL-17A levels seen in our study.

Fig. S16: (A) Percentages of T_H17 cells as a fraction of CD4+ cells in lungs of denatonium-treated *Trpm5*^{+/+} and *Trpm5*^{-/-} mice or controls (n=5-10). One-way ANOVA. (B) Percentages of T_H17 cells in denatonium-treated or control *Panx1*^{+/+} (n=4-6) and *Panx1*^{-/-} mice (n=4-5). Unpaired Student's t-test (C) FACS analyses of T_H17 cell counts in lungs of *Trpm5*^{+/+} and *Trpm5*^{-/-} mice 2 days (2d pi) and 3 days post infection (3d pi) with *P. aeruginosa*, compared to untreated controls showed increased T_H17 cell levels 2d pi in *Trpm5*^{+/+} mice (n=3-8). (* p<0.05, ** p<0.01, one-way ANOVA) (D, E) $\gamma\delta$ T cell levels in *Trpm5*^{+/+} mice remained unaltered in the lungs and in the blood tree days after treatment with denatonium compared to untreated control animals in FACS analyses (n=4 for control and denatonium-treated animals). (ns: not significant, unpaired Student's t-test) (F) FACS analyses of lungs of denatonium-treated *Trpm5*^{+/+} and *Trpm5*^{-/-} mice revealed no changes in ILC3 numbers after denatonium treatment compared to controls. (one-way ANOVA).

f) Figure A: scale bars in A-C are missing; Statistics in F&H is strangely added and should be revised;

Reply: We thank the reviewer for this valuable comment. We have now added the scale bars in Figures 1B and C and changed the colour of the scale bar in Figure 1A (revised Fig. 1D-F). Additionally, we have revised Figure 1F (now Fig. 1G), and hope that the updated version is clearer. In the experiment, we compared the increase in fluorescence at the membrane of ATP sensor cells to their baseline fluorescence, following the elevation of intracellular Ca²⁺ levels in *Trpm5*^{+/+} tuft cells positioned near the ATP sensor cells. For statistical analyses, we applied a paired Student's t-test and observed a significant increase in fluorescence. Given that the pmATPSnFR1.0 sensor specifically localises to the plasma membrane and responds to changes in ATP concentrations in the extracellular space with fast fluorescence increases (Lobas et al., 2019, PMID: 30755613), we assume that the observed effects indicate ATP release from tuft cells. During the revision process, we performed additional control

experiments with *Trpm5*^{-/-} tuft cells alongside ATP sensor cells (revised Fig. 1G). In this experimental setting, no changes in fluorescence were detected in ATP sensor cells, indicating that *Trpm5* activation is indispensable for ATP release from tuft cells. In Figure 1I-J, we compared the ATP levels in supernatants from vehicle-treated tracheae of different mouse strains with those from denatonium- or CNO-treated mice of the corresponding mouse strains, using an unpaired Student's t-test. We hope that our explanations clarify the figure and support our conclusions.

Fig. 1 Tuft cells (TCs) release ATP and express pannexin 1. (A) Representative I-V relationships of normalized TRPM5 current amplitudes (pA/pF) measured in a TC at indicated time points after

breaking the cell. (B) Representative current-voltage (I-V) relationships of whole-cell currents measured in a tracheal TC from a *Trpm5*^{+/+} mouse or a *Trpm5*^{-/-} mouse in the presence of 1 μ M intracellular Ca²⁺. (C) Statistical analysis (Unpaired Student's *t*-test) of currents illustrated in (A). Current amplitudes measured at -100 mV and +100 mV in *Trpm5*^{-/-} TCs (n=9 cells) and *Trpm5*^{+/+} TCs (n=10 cells) are shown. (D) The bright field of a patched TCs and ATP sensor cells. (Green cells: GFP-expressing TC) (E) Ca²⁺ imaging showing the beginning of recording, and (F) example of maximal fluorescent changes in cell membranes (arrow). (G) The statistics of maximal fluorescence intensity of all ATP sensor cells responding to TC stimulation in *Trpm5*^{+/+} mice (n=13 cells/5 mice/5 experiments) and *Trpm5*^{-/-} mice (n=19 cells/4 mice/4 experiments). Control: baseline fluorescence of ATP sensor cells, TC Ca²⁺; fluorescence in ATP sensor cells after TC stimulation with intracellular high Ca²⁺; paired Student's *t*-test (H) The fluorescence intensity of marked cells in (C) over the whole cell recording. (I) Supernatants from denatonium-treated tracheae of *Trpm5*^{+/+} mice (n=5) revealed higher ATP levels than those from *Trpm5*^{-/-} mice (n=5) and vehicle-treated tracheae (n=4-5) quantified by ELISA assays. Supernatants from clozapine-N-oxide (CNO, 100 μ M) treated *Trpm5*-DREADD mice (n=5) contained higher ATP concentrations after stimulation compared to vehicle treated controls (n=5). unpaired Student's *t*-test (J) Supernatants from denatonium-treated tracheae from *Panx1*^{-/-} mice (n=3) revealed the same ATP levels as vehicle-treated controls (n=3) quantified by ELISA assays. Tracheal supernatants from denatonium-treated Chat^{fl/fl}:*Trpm5*^{cre} mice (n=3) contained higher ATP levels than vehicle treated tracheae (n=3). unpaired Student's *t*-test (K-O) Immunohistochemistry for Trpm5 (TC marker) combined with LacZ-staining suggest partial but not exclusive co-expression of Trpm5 (brown) and Panx1/2 (perinuclear blue signal) in the tracheal epithelium of *Panx1*^{-/-}/*2*^{-/-} mice. Arrows indicate LacZ staining in the nuclei of Trpm5+ cells. (** p<0.01, ** p<0.01, *** p<0.001)

g) Figure 3-5: gating strategies of analyzed pulmonary cell populations should be added to supplement. Data of cell frequencies should be also shown as absolute numbers but also as # of total CD45+ cells.

Reply: We thank the reviewer for this comment. We followed the reviewer's recommendation and added the gating strategy to the supplementary figures (Fig. S8-12). Additionally, we included the percentages of the different immune cell population related to the total CD45+ cell number (Supplementary Fig. 13 and 14). Nevertheless, we would like to stay with the absolute number of cells in the figures in the main body of the manuscript. We think that the absolute cell number provides a more accurate picture about the recruitment of immune cells into peripheral organs than their relative proportions, since the percentage of a certain immune cell type could be influenced by any fluctuation in the number of other cell types. We hope that this proceeding meets the reviewer's approval. Please refer to the supplemental data for the respective figures.

h) Figure 6: Activation of DC is multi-layered – are other activation markers of DC elevated?

Reply: We thank the reviewer for this comment which has helped to improve the manuscript. We agree with the reviewer that several factors contribute to DC activation. We now performed additional experiments in which we investigated CCR7 levels in dendritic cells. We found a significant upregulation of CCR7+ dendritic cells after specific stimulation of tuft cells with CNO when dendritic cells were co-cultured with epithelial cells for 4 and 24 hours. This effect was ATP-mediated since it was abolished by the application of apyrase (ATP depletion). Additionally, when we stimulated tracheal tuft cells with denatonium *in vivo*, we observed a prominent increase in CCR7+ dendritic cells in lungs and tracheae from *Trpm5*^{+/+} mice as well as in the draining lymph nodes. The increase in *Ccl19* and *Ccl21b* expression together with the increase in CD86 and CCR7 on dendritic cells in tracheae, lungs and

draining lymph nodes strongly indicates a role for tuft cells in regulation of dendritic cell function.

Please see also our response to your comment **a**. The new results are included in the revised Fig. 7 A-C, and Supplementary Fig. 18 and 19.

Fig. 7: Tuft cell (TC)-induced dendritic cell (DC) phagocytic activity and migration is dependent on ATP. **(A)** Co-culture of CNO-stimulated Trpm5-DREADD tracheal epithelium (TE) with DCs significantly increased the percentage of CCR7-expressing DCs after 4h (n=6) and 24h (n=7) compared to CNO-treated Trpm5^{+/+} TE co-culture with DCs (n=4-5). The increase in % of CCR7⁺ DCs after 24h was abolished by apyrase (5 U/ml) (n=5). DCs cultured alone and treated with CNO (n=3) exhibited the same percentage of CCR7-expressing cells as DCs co-cultured with Trpm5^{+/+} TE. One-way ANOVA. **(B)** The number of CCR7-expressing DCs in lungs increased significantly 3 days after intratracheal TC stimulation with 1 mM denatonium in Trpm5^{+/+} mice in contrast to Trpm5^{-/-} mice (n=4). One-way ANOVA. **(C)** In Trpm5^{+/+} mice (n=6) CCR7-expression DC counts in lymph nodes are increased 3 days after denatonium treatment, but not in Trpm5^{-/-} mice (n=4). Kruskal-Wallis test.

Fig. S18: FACS analyses of CCR7⁺ DC numbers in tracheae from Trpm5^{+/+} and Trpm5^{-/-} mice revealed an increase in CCR7⁺ DCs three days after treatment with denatonium in Trpm5^{+/+} and Trpm5^{-/-} mice. (n = 3-4)

Fig. S19: Relative expression of Ccl21b in whole tracheae **(A)** and lungs **(B)** of Trpm5^{+/+} and Trpm5^{-/-} mice 3 days after the intratracheal application of 1 mM denatonium (n=3, each point represents the average of two technical replicates). **(C)** Relative expression of Ccl19 in airway draining lymph nodes (LN) of Trpm5^{+/+} and Trpm5^{-/-} mice 3 days after the intratracheal application of 1 mM denatonium (n=3, each point represents the average of two technical replicates). (* p<0.05, **<0.01, *** p<0.001; one-way ANOVA).

Reviewer 2:

Reviewer #2 (Remarks to the Author):

The paper by Wadood et al addresses the function of tufts cells in shaping the immune response in the upper respiratory tract. The authors propose a model whereby taste signaling via Trpm5 would result in ATP release by tuft cell with subsequent dendritic cell activation, phagocytosis and migration. Tuft cell activation via Trpm5 would also affect the adaptive branch of the immune system by recruiting IL-17A secreting CD4 T cells to respond to bacterial infection. The paper convincingly shows how wildtype tuft cells release ATP following stimulation with denatonium in vitro as well as in tracheal supernatants from CNO-activated Trpm5-DREADD mice. ATP release by tuft cells was shown to depend on pannexin 1, which colocalized with Trpm5 in tuft cells. Trpm5- induced K⁺ efflux underlies pannexin 1 dependent currents.

Comment 1: It would be interesting to know if the activity of the ATP-gated P2X7 receptor is involved in K⁺ efflux, a point that could be addressed by pharmacological inhibition of P2X7R.

Reply: We thank the reviewer for the recommendation. We would like to clarify that KCl/K⁺ was used in our experimental setting to provide evidence for the presence of functional pannexin 1 channels in tuft cells (Jackson et al., 2014, PMID: 24694658). Extracellular K⁺ accumulation at concentrations near 100 mM was reported to cause significant membrane depolarization, which opens Panx1 channels (Silverman et al., 2009, PMID: 19416975; Whyte-Fagundes et al., 2018, PMID: 28735901). In our experiments KCl-induced currents could be abolished by carbenoxolone or probenecid, indicating that pannexin 1 is active in tuft cells.

It has been reported that ATP released via pannexin 1 activates the purine receptor P2x7, leading to additional K⁺ efflux and further activation of pannexin 1 (Iglesias et al., 2008 PMID: 18596211; Locovei et al., 2006 PMID: 16364313). We conducted additional whole-cell patch clamp experiments to pharmacologically inhibit the P2x7 receptor using its specific inhibitor, AZ10606121 (Karmakar et al., 2016, PMID: 26877061). After increasing intra-cellular Ca²⁺ concentration in tuft cells, we detected Trpm5 currents. No significant changes in current density (pA/pF) were observed in the presence or absence of the P2x7 receptor inhibitor. Based on these results we propose that the activation of tuft cells leads to Trpm5-dependent K⁺ efflux, resulting in depolarisation of the plasma membrane and opening of pannexin 1, which does not involve P2x7 receptor activation. Furthermore, the low expression of P2x7 receptor in tuft cells, as revealed in single-cell RNAseq studies from our group and from others (Hollenhorst et al., 2020, PMID: 31914675; Bankova et al., 2018, PMID: 30291131), supports our conclusions. We included this data in the revised version of Fig. 2H.

The authors used Trpm5-DREADD mice to show intracellular increases in Ca²⁺ in tuft cells and recruitment of neutrophils in BALF and trachea following CNO inhalation. Monocytes (but not macrophages) and DCs were increased as well.

Comment 2: It could be interesting to address whether a switch between interstitial (IM) and alveolar (AM) macrophages (CD64+ and CD170+, respectively) has occurred. With unaltered absolute counts, there might be an increase in AM and a corresponding decrease in IM.

Reply: This is indeed a very interesting comment. At least three major macrophage populations are found in the lung, distinguished by their anatomical location: alveolar macrophages within the air-exposed space of the alveolus and two to three interstitial macrophage populations within the interstitial regions of the lung (Mass E et al., 2023, PMID: 36922638). In the original version of the manuscript, we included only the alveolar macrophage population, not the total macrophage population. This is because alveolar macrophages represent the major macrophage population in the lung and are considered the

primary gatekeepers and housekeepers of alveolar homeostasis (Mass E et al., 2023, PMID: 36922638). We have now added an additional panel for interstitial macrophages in Figure 3E and specified the macrophage populations as IM for interstitial macrophages and AM for alveolar macrophages. Further analysis of our data revealed a difference between interstitial and alveolar macrophages in bronchoalveolar lavage fluid (BALF) of *Trpm5*-DREADD mice. While there was no increase in alveolar macrophages following tuft cell stimulation, the number of interstitial macrophages were elevated. These findings correspond to previous observations that interstitial macrophages can migrate into the alveolar space under inflammatory conditions (Bain et al., 2022, PMID: 35017701). In influenza infection interstitial macrophages play a more prominent role in the efferocytosis of the alveolar epithelium than alveolar macrophages (Sanchez Santos Rizzo Zuttion et al., 2024, PMID: 38207122). The role of the macrophage population following tuft cell activation in pneumonia certainly warrants investigation in future studies.

We now included the data on interstitial macrophages in the revised version of the manuscript in Figures 3E and F and added a brief comment in line 224-227 that reads: "Monocyte and interstitial macrophage numbers, but not alveolar macrophage numbers increased in BALF from CNO-treated *Trpm5*-DREADD animals (Fig. 3, D, E and F). Thus, interstitial rather than alveolar macrophages seem to play a role in the acute TC-induced immune response, in line with a study by Sanchez et al³⁵."

Fig. 3(E) Interstitial macrophage (IM) numbers increased in BAL samples of *Trpm5*-DREADD mice treated with 100 μ M CNO (n=7) compared to CNO-treated *Trpm5*^{+/+} mice (n=7). sham-treated: n=5, control=untreated: n=6. (F) The number of alveolar macrophages (AM) remained unchanged in BAL-samples of *Trpm5*-DREADD mice treated with 100 μ M CNO (n=7). *Trpm5*^{+/+}: n=7, sham-treated: n=5, control=untreated: n=6.

In vivo inhalation of denatonium was used to show the expansion of tuft cells and increases of DCs in the trachea of *Trpm5*^{+/+} but not *Trpm5*^{-/-} mice at 24 h after stimulation. However, at 72 h CD11c+ cells were significantly increased also in *Trpm5*^{-/-} cells. A positive correlation was shown between the increase of TC and DCs in the trachea after denatonium inhalation.

Comment 3: How would the authors explain the increase of DCs in *Trpm5*^{-/-} mice at 72 h after denatonium stimulation? Also, I would tone down the final statement of the paragraph: "These data suggest that immune cell recruitment depend on TC number and activation"

Reply: We thank the reviewer for this comment. We evaluated additional samples, e.g., tracheal sections from *Trpm5*^{+/+} and *Trpm5*^{-/-} mice at the 3-day time point and observed that there was no increase in dendritic cells, as assessed by CD11c staining, in the *Trpm5*^{-/-} mice at any time point. In contrast, increased dendritic cell numbers were observed in the *Trpm5*^{+/+} tracheae (please see Fig. S7B). Additionally, there is no correlation between tuft cell and dendritic cell numbers in the *Trpm5*^{-/-} mice, unlike the tracheal sections from *Trpm5*^{+/+} mice (Fig. S7C and D). We believe that these results - the observed correlation of tuft cell and dendritic cell numbers exclusively in *Trpm5*^{+/+} mice and the lack of changes in tuft cell and dendritic cell numbers in *Trpm5*^{-/-} mice - suggest that dendritic cell recruitment depends on tuft cell activation and numbers. We suspect that the high number of dendritic cells previously observed in *Trpm5*^{-/-} mice was due to a technical issue with the tissue perfusion in

one mouse. This has been confirmed by re-evaluating samples from all animals. We excluded the sections from this particular mouse from the final dataset and statistical analyses and included samples from a new experiment. We hope that the results and our approach to addressing this issue will satisfy the reviewer and alleviate their concerns. Additionally, we revised the final statement of the paragraph to focus on dendritic cell recruitment and tuft cell activation, as tuft cell activation clearly induces dendritic cell recruitment, which was absent in *Trpm5*^{-/-} mice. The sentence now reads (line 257-258): “These data indicate that TCs are involved in the regulation of DC function.”

Please find the new Supplementary Figure 7 below:

Fig. S7: Dendritic cell numbers correlate with tuft cell numbers. **(A)** Immunofluorescence staining for tuft cells (DCAMKL1) in tracheal cross sections and quantification of DCAMKL1⁺ cells in *Trpm5*^{+/+} and *Trpm5*^{-/-} mice treated with 1 mM denatonium after 24h, 72h, 7d or with vehicle (PBS) after 24h (n=20 sections of 4 mice for each experimental condition). **(B)** Immunofluorescence staining of dendritic cells (CD11c) in tracheal cross sections and quantification of CD11c⁺ cells in *Trpm5*^{+/+} and *Trpm5*^{-/-} mice treated with 1 mM denatonium after 24h, 72, 7d or with vehicle (PBS) after 24h (n=20 sections of 4 mice for each experimental condition). **(C)** Correlation analysis of tuft cell and dendritic cell numbers of *Trpm5*^{+/+} mice revealed a moderate positive linear correlation with Pearson's r=0.4454. **(D)** There was no correlation between tuft cell and dendritic cell numbers in *Trpm5*^{-/-} mice (r=0.0432). A, B: * p<0.05, ** p<0.01, one-way ANOVA.

Trpm5 was shown to contribute to the increase of DCs in mice infected with *P. aeruginosa*. Notably, *Trpm5*^{-/-} mice infected with *P. aeruginosa* showed significantly impaired survival with respect to *Trpm5* proficient animals. A less pronounced decrease in mice survival was observed with *Panx1*^{-/-} mice. Experiments in *Trpm5* and *Panx1* proficient vs deficient mice treated with denatonium led the authors hypothesizing that *Trpm5* dependent ATP release promoted the recruitment of Th17 cells in the lungs.

Comment 4: Did you check for differences in Th17 cells in *Trpm5*^{+/+} vs *Trpm5*^{-/-} mice infected with *P. aeruginosa*?

Reply: Indeed, we checked for differences in T_H17 cells in Trpm5^{+/+} and Trpm5^{-/-} mice infected with *P. aeruginosa* and observed an increase in T_H17 cell counts in the lungs of Trpm5^{+/+} mice two days post infection that was completely absent in Trpm5^{-/-} mice. Interestingly, the increased number in T_H17 was reduced back to control levels at three days post infection in Trpm5^{+/+} mice while which it remained unaltered in Trpm5^{-/-} mice.

It is well known that once T_H17 cells reach the site of inflammation, they release IL-17 to stimulate the expression of pro-inflammatory cytokines like granulocyte-macrophage colony-stimulating factor, granulocyte-colony stimulating factor, IL-6 and tumor necrosis factor-alpha (TNF-α) (Xu & Cao, 2010, PMID: 20383173). In addition, IL-17 also promotes the secretion of CXC chemokines, which attracts neutrophils *in vivo* and stimulates the production of antimicrobial peptides, such as β-defensin providing defence against a wide range of microorganisms (Kao, et al., 2004, PMID: 15322213; Ganz, 1999, PMID: 10577203). A persistent secretion of IL-17 is involved in many inflammatory diseases (Thomas, et al., 2023, PMID: 36902294). Thus, it is tempting to speculate that this early tuft cell-dependent increase in T_H17 cell numbers in infection is necessary to give the immune system a sufficient boost for clearing the bacteria.

We now included these data in the revised version of the manuscript in Supplementary Fig. S16C.

C

Fig. S16C: (C) FACS analyses of T_H17 cell counts in lungs of Trpm5^{+/+} and Trpm5^{-/-} mice 2 days (2d pi) and 3 days post infection (3d pi) with *P. aeruginosa*, compared to untreated controls showed increased T_H17 cell levels 2d pi in Trpm5^{+/+} mice (n=3-8). (* p<0.05, ** p<0.01, one-way ANOVA)

IL-17A concentrations were shown to be elevated in the plasma of denatonium-treated Trpm5^{+/+} mice compared to controls but not in plasma of Trpm5^{-/-} mice

Comment 5: Variations of gamma/delta T cells in the lung after denatonium inhalation should be investigated. They represent a substantial proportion of lymphocytes in the lung and are the major producers of IL-17A at early stage of infections (doi:10.1111/imm. 12764). Furthermore, γδ-T cells are sensitive to ATP.

Reply: We are grateful for this interesting comment which helped to improve the manuscript. We agree with the reviewer that it is of interest to analyse the γδ-T cell numbers and their dependency on ATP released upon tuft cell stimulation. We now performed additional *in vivo* experiments in Trpm5^{+/+} mice followed by FACS analyses, in which we investigated the γδ-T cell counts three days after intratracheal treatment with denatonium and included these data in Supplementary Fig. S16D and E. Surprisingly, we did not observe any changes in the γδ-T cell counts in the lungs or the blood after treatment of the mice with denatonium. We also included additional data of infection experiments about T_H17 cell counts in the lung (Supplementary Fig. S16C). In these experiments we observed an increase in T_H17 cells in the lungs of wild-type mice 2 days after infection, which was not the case in Trpm5^{-/-} mice, and therefore tuft cell specific. Taken together, the unchanged number of γδ-T cells indicates that the responsible cells for the increased IL-17A levels we observed in our study are due to increased numbers and activation of T_H17 cells.

Additionally, we also did not observe any changes also in the numbers of other IL-17A producing cells e.g. ILC3 (Supplementary Figure S16F). We included the information at page 14/line 335-340 and in Supplementary Fig. S16. It now reads: "Since $\gamma\delta$ T cells and ILC3 may also be a source of IL-17A, we investigated these cell types following TC stimulation^{37,38}. No differences in $\gamma\delta$ T cell numbers were observed in the lungs or blood samples after denatonium treatment (Supplementary Fig. 16D and E). ILC3 numbers remained unchanged in lungs of both *Trpm5*^{+/+} and *Trpm5*^{-/-} mice three days after TC activation with denatonium, compared to the respective untreated controls (Supplementary Fig. 16F)."

Upregulation of CD86 in either wildtype or *Panx1*^{-/-} DCs cocultured with TCs from *Trpm5*-DREADD mice stimulated with CNO showed that DCs activation was dependent on *Trpm5*-dependent ATP release by TCs. MHCII upregulation was also shown, however, it was abolished by mecamlamine and atropine (in addition to apyrase), thereby complicating the interpretation of *Trpm5* dependent DC activation.

Comment 6: Contour plots in fig 6 are difficult to compare and more details on panel A-F should be provided in the figure legend. Please provide the gating strategy for all flow cytometry experiments.

Reply: We regret the oversight that the gating strategies were not included in the submitted manuscript. We also added the gating strategies for other flow cytometry experiments in the supplemental data. They can be found in Supplementary Fig. S8-12.

We have now included more detailed description of panels A-F in Figure 6. We hope that it is now easier to understand. The figure legend now reads: "Upregulation of CD86 on dendritic cells (DCs) is dependent on ATP released from tuft cells (TCs). (A-F) FACS analysis of CD86 expression on sorted pulmonary DCs (CD45⁺Ly6G⁻F4/80⁺CD11c⁺) cocultured with sorted *Trpm5*-DREADD tracheal epithelium (EpCAM⁺CD45⁻) (n=7). Clozapine-n-oxide (CNO) (100 μ M) was used to stimulate *Trpm5*-DREADD tracheal epithelium in presence or absence of ATP hydrolyzing enzyme apyrase (5 U/ml) or nicotinic receptor antagonist mecamlamine (10 μ M) and muscarinic receptor antagonist atropine (1 μ M) in the coculture system. The percentage of CD86-expressing DCs was significantly higher when DCs were cocultured with CNO stimulated *Trpm5*-DREADD tracheal epithelium compared to CNO treated DCs (n=10) or CNO treated *Trpm5*^{+/+} tracheal epithelium. Treatment of CNO-stimulated *Trpm5*-DREADD tracheal epithelium with apyrase but not mecamlamine/atropine reduced the percentages of CD86⁺ DCs (n=4). One-way ANOVA."

For better comparison, we performed changes in Figure 6 and 7 (see below). We now show in the revised Figure 6 the upregulation of CD86 and MHCII on dendritic cells (FACS analyses) and the enhanced phagocytotic activity of dendritic cells following activation of tuft cells and release of ATP. In the revised Figure 7 we focused on migratory activity of dendritic cells treated with supernatants from stimulated tuft cells.

Revised Figure 6: Upregulation of CD86 on dendritic cells (DCs) is dependent on ATP released from tuft cells (TCs). (**A-F**) FACS analysis of CD86 expression on sorted pulmonary DCs (CD45+ Ly6G-F4/80-CD11c+) cocultured with sorted *Trpm5*-DREADD tracheal epithelium (EpCAM+ CD45-) (n=7). Clozapine-n-oxide (CNO) (100 μ M) was used to stimulate *Trpm5*-DREADD tracheal epithelium in presence or absence of ATP hydrolyzing enzyme apyrase (5 U/ml) or nicotinic receptor antagonist mecamylamine (10 μ M) and muscarinic receptor antagonist atropine (1 μ M) in the coculture system. The percentage of CD86-expressing DCs was significantly higher when DCs were cocultured with CNO stimulated *Trpm5*-DREADD tracheal epithelium compared to CNO treated DCs (n=10) or CNO treated *Trpm5*^{+/+} tracheal epithelium. Treatment of CNO-stimulated *Trpm5*-DREADD tracheal epithelium with apyrase but not mecamylamine / atropine reduced the percentages of CD86⁺ DCs (n=4). One-way ANOVA. (**G**) Co-culture of pulmonary DCs from *Panx1*^{-/-} mice with CNO-stimulated *Trpm5*-DREADD tracheal tuft cells significantly increased the percentages of CD86 expressing DCs compared to CNO treated *Panx1*^{-/-} DCs (n=4). Unpaired Student's *t*-test. (**H**) The median fluorescence intensity (MFI) of MHCII in pulmonary DCs was significantly increased after being co-cultured with CNO-stimulated *Trpm5*-DREADD tracheal epithelium (n=8-10). This was reduced when CNO-stimulated *Trpm5*-DREADD tracheal epithelium was treated with apyrase (n=3), or mecamylamine and atropine (n=4). One-way ANOVA. (**I**) Blood agar plate of phagocytosed *P. aeruginosa* NH57388A CFUs after treatment of DCs with supernatant (SN) of *Trpm5*^{+/+} tracheae stimulated with 1 mM denatonium, with supernatant of *Trpm5*^{+/+} tracheae stimulated with denatonium in the presence 5 U/ml apyrase and with supernatants of denatonium-treated *Trpm5*^{-/-} tracheae and untreated control conditions. (**J**) Quantification of (A) (n=5). One-way ANOVA (**K**) Schematic drawing. Activation of TCs with denatonium or CNO leads to increased numbers and activation of DCs dependent on ATP. (* $p < 0.05$, ** $p < 0.01$)

Revised Figure 7: Tuft cell (TC)-induced dendritic cell (DC) phagocytic activity and migration is dependent on ATP. (A) Co-culture of CNO-stimulated *Trpm5*-DREADD tracheal epithelium (TE) with DCs significantly increased the percentage of CCR7-expressing DCs after 4h (n=6) and 24h (n=7) compared to CNO-treated *Trpm5*^{+/+} TE co-culture with DCs (n=4-5). The increase in % of CCR7⁺ DCs after 24h was abolished by apyrase (5 U/ml) (n=5). DCs cultured alone and treated with CNO (n=3) exhibited the same percentage of CCR7-expressing cells as DCs co-cultured with *Trpm5*^{+/+} TE. One-way ANOVA. (B) The number of CCR7-expressing DCs in lungs increased significantly 3 days after intratracheal TC stimulation with 1 mM denatonium in *Trpm5*^{+/+} mice in contrast to *Trpm5*^{-/-} mice (n=4). One-way ANOVA. (C) In *Trpm5*^{+/+} mice (n=6) CCR7-expression DC counts in lymph nodes are increased 3 days after denatonium treatment, but not in *Trpm5*^{-/-} mice (n=4). Kruskal-Wallis test. (D) DC migration under different experimental conditions: untreated, treated with supernatants (SN) from denatonium-treated *Trpm5*^{+/+} mice without and with apyrase, and from denatonium-treated *Trpm5*^{-/-} mice. (E) Quantification of (C). DCs treated with supernatants of unstimulated tracheae of *Trpm5*^{+/+}, *Trpm5*-DREADD, *Trpm5*^{-/-} and *Trpm5*-DTA mice as well as with supernatants of tracheae stimulated with 1 mM denatonium of *Trpm5*^{+/+} mice with and without apyrase (5 U/ml), of denatonium-treated *Trpm5*^{-/-} and *Trpm5*-DTA tracheae, and from CNO-treated (100 μM) *Trpm5*-DREADD tracheae with and without apyrase. Wound closure was evaluated 24 h after induction of the scratch (n=3). One-way ANOVA or unpaired Student's *t*-test. (F) Schematic drawing. Activation of TCs with denatonium or CNO leads to activation of the *Trpm5* channel, resulting in ATP release via pannexin 1 and TC expansion. The released ATP leads to an increased recruitment, migration, activation, and phagocytosis of DCs, recruitment of Th17 cells and IL-17A secretion and to paracrine activation of mucociliary clearance. (* p<0.05, ** p<0.01)

Finally, *in vitro* phagocytic assays with DCs infected with *P. aeruginosa* revealed the enhanced phagocytic activity of DCs treated with tracheal supernatants from Trpm5^{+/+} vs Trpm5^{-/-} mice treated with denatonium. Analogously, DCs migration was shown to be enhanced by tracheal supernatants from Trpm5 proficient mice.

General comment: A major issue I see in this study is the lack of connection of tuft cell activation with an antigen specific response and the possible polarization of a CD4 T cell response toward the Th17 program, if I understood correctly the hypothesis of the paper. Alternatively, the mere recruitment of Th17 by TC stimulation via Trpm5 activation seems an “inappropriate” immunopathological stimulus. It is clear that Trpm5 mediated ATP release by TCs can condition both innate immune cells and enrich Th17 cells in the lung. However, these evidences are difficult to integrate in the physiology or pathophysiology of an adaptive immune response.

Reply: We thank the reviewer for this comment. Indeed, we have used denatonium as a model substance for specific tuft cell activation, allowing us to mechanistically study ATP release from tuft cells. In this context, denatonium mimics bacterial substances that activate tuft cells (Hollenhorst et al., 2022, PMID: 35954259; Lossow et al., 2016, PMID: 27226572). Based on our previous and current work, we propose that under both physiological and pathophysiological conditions, tuft cells detect changes in the microenvironment – such as the presence of quorum-sensing molecules, formyl peptides, and others - and initiate a defence program involving both innate and the adaptive immunity. We have now evaluated the recruitment of T_H17 cells also in the lungs of animals infected with *P. aeruginosa*. In this model the stimulation of the immune system is more complex, nevertheless immune responses mediated by tuft cell can be successfully studied (Hollenhorst et al., *JCI* 2022, PMID: 35503420). Please find the results in Supplementary Fig. S16C. In these experiments, in the lungs of Trpm5^{+/+} animals we observed an increase in T_H17 cells already two days post infection, which dropped back to baseline control levels at three days post infection. In contrast to this, we did not observe an increase in T_H17 cells in Trpm5^{-/-} mice. This further underlines our conclusion that T_H17 cell recruitment was dependent on tuft cell activation, since the Trpm5 channel is exclusively expressed in tuft cells in the airway epithelium. It is tempting to speculate that the early transient increase in T_H17 cells is an important stimulus to initially boost the adaptive immunity and that the subsequent return to baseline levels is responsible for the prevention of harmful chronic immune responses. Our observation of TC-dependent T_H17 cell recruitment induced by *P. aeruginosa* infection, a different stimulus than denatonium, in which antigen specific responses are clearly involved, helps to address the concerns of the reviewer regarding the TC-dependent activation of adaptive immune responses. Furthermore, we performed additional experiments and observed increased CCR7 levels on dendritic cells, as well as elevated expression of its ligands Ccl21a and Ccl19, in the lungs, tracheae and lymph nodes after tuft cell stimulation. CCR7 is known to play an important role in the migration of dendritic cells into secondary lymphatic organs, such as the lymph nodes, as observed in our study, and for differentiation and activation of T cells (Liu et al., 2021, PMID: 34302064). Therefore, these data support our conclusion in the paper that activation of tuft cells may also induce an adaptive immune response in addition to the innate immune response via dendritic cell activation, which in turn migrate to lymph nodes, where they activate cells of the adaptive immune system, such as T_H17 cells. Interestingly, ATP has previously been shown to be sufficient to induce T_H17 differentiation when added to a dendritic cells-naïve CD4⁺ cells co-culture system (Sáez PJ et al., 2017, PMID: 29162744). Additionally, ATP has been shown to be an appropriate stimulus for T_H17 cell differentiation from CD4⁺ cells in the intestinal lamina propria in the absence of pathogenic bacteria (Atarashi et al., 2008, PMID: 18716618). Is it possible that denatonium induces T_H17 cell differentiation through tuft cell-released ATP, which acts on dendritic cells and CD4⁺ cells.

We hope, that the results from our *P. aeruginosa* infection experiments help to mitigate the reviewer's concerns regarding our conclusions.

Minor comments

1. Please provide details on the ELISA test used for ATP measurements

Reply: We apologize for the insufficient description of ELISA experiments. We have now provided a more detailed description of the protocol as well as the product number of the assay that was used in the supplemental methods in the paragraph "ATP measurements". The respective paragraph (page 3 line 10 in the supplement) now reads:

"Briefly, tracheas isolated from *Trpm5*^{+/+}, *Trpm5*^{-/-} and *Panx1*^{-/-} mice were stimulated with 1 mM denatonium (Sigma-Aldrich) or with 100 μ M clozapine-N-oxide (CNO, HelloBio, Bristol, UK) for tracheas explanted from *Trpm5*-DREADD mice. Stimulation was performed after tracheal explantation in 200 μ l of a solution containing 5.6 mM KCl, 136 mM NaCl, 10.7 mM glucose, 10 mM HEPES, 1 mM MgCl₂ and 2.2 mM CaCl₂ for 5 min at 37°C. Supernatants used for ATP measurements were collected by centrifugation at 1000g for 5 min. For the ELISA assay, 50 μ l of each sample was added to a well of a 96 well plate, followed by 50 μ l of detection reagent A. After mixing, the plate was incubated for 1 h at 37°C and then washed 3 times with wash buffer. Next, 100 μ l of detection reagent B were added, and the plate was incubated for 30 min at 37°C, followed by 5 washes. Then, 90 μ l of substrate solution were added, and the plate was incubated for approximately 20 min at 37°C. Finally, 50 μ l of stop solution was added, and the plate was read at a wavelength of 450 nm. Analyses were performed using an EnSight Multimode Plate Reader (Perkin Elmer™, Rodgau-Jügesheim, Germany)."

2. The legend of Fig 1 J-M should be more detailed and arrows explained

Reply: We followed the reviewer's recommendation and have explained the arrows in Fig. 1 J-M (now Figure 1 K-O). We indicated that the brown staining represents the *Trpm5*⁺ cells and the LacZ staining for *Panx1/2* is represented by the blue staining of the perinuclear region. The legend now reads as follows (line 868-872): "(K-O) Immunohistochemistry for *Trpm5* (TC marker) combined with LacZ-staining suggest partial but not exclusive co-expression of *Trpm5* (brown) and *Panx1/2* (perinuclear blue signal) in the tracheal epithelium of *Panx1*^{-/-/2}^{-/-} mice. Arrows indicate LacZ staining in the nuclei of *Trpm5*⁺ cells." Additionally, we quantified the number of LacZ⁺ cells in the tracheae of *Panx1*^{-/-} mice (included in Supplementary Fig. S4F). We found that approximately 40% of *Trpm5*⁺ tracheal tuft cells are pannexin 1 expressing cells (LacZ⁺ cells). This finding corresponds to our and other groups' single-cell RNAseq data from tuft cells (Hollenhorst et al., 2020 PMID: 31914675, Bankova et al., 2018, PMID: 30291131).

We hope that the revised version of the figure help to reconcile the reviewer's concerns.

Supplementary Fig S4F: (F) Quantification of double-positive LacZ⁺/*Trpm5*⁺ TCs *Panx1*^{-/-} mouse tracheas (n=4).

3. In fig S3C, statistical significance seems impossible (please detail which test was used)

Reply: We have now changed the former Fig. S3 which is now Fig. S5. We added representative current traces (I_{SC}) for apical application of 1 mM denatonium (den) to $Panx1^{+/+}$ and $Panx1^{-/-}$ mouse trachea and separated the previous Fig. S3C into three graphs, which are now Fig. S5D, E, and F. We performed two more experiments with the purinergic receptor antagonist suramin and included the data in the revised version of the manuscript. For statistical comparison we used the paired Student's t-test and compared the control denatonium effect with the effect observed in the presence of the respective inhibitors (Fig. S3D, E and F). Previously, we have shown that repeated apical application of 1 mM denatonium and denatonium + vehicle (in our case H_2O) does not change the denatonium-induced current in the trachea (Hollenhorst et al. 2022, PMID: 35954259). Therefore, we didn't perform the vehicle controls again. The experiments with the inhibitors were conducted as follows: First, 1 mM denatonium was applied apically, then the trachea was washed. Next, the respective inhibitor was applied (suramin, probenecid or Ruthenium Red) and after a short incubation, denatonium was applied again in the presence of the inhibitor. The denatonium-induced current changes were significantly reduced in the presence of suramin or probenecid, but not with Ruthenium Red when the effects were compared to the effect in the same trachea with denatonium applied alone (control).

Fig. S5: Tuft cell released ATP modulates transepithelial ion transport. **(A)** Schematic drawing of the action of denatonium on ATP release. **(B)** Representative current traces (I_{SC}) of a wildtype ($Panx1^{+/+}$) and of a $Panx1^{-/-}$ mouse trachea upon apical application of 1 mM denatonium (den). **(C)** In $Panx1^{-/-}$ (n=4) and $Panx1^{-/-}2^{-/-}$ mice (n=4) the denatonium-induced current (1 mM, apical, DI_{SC}) was reduced compared to wild type controls ($Panx1^{+/+}$, n=11, *** p<0.001), while the denatonium-induced effect in $Panx2^{-/-}$ mice (n=4) was similar to wildtype mice. (one-way ANOVA) **(D)** The CALHM1-inhibitor Ruthenium Red (RuR, 20 μM , apical and basolateral) did not change the denatonium-induced current compared to control conditions (n=4, ns: not significant, paired Student's t-test). **(E)** Application of the pannexin inhibitor probenecid (200 μM , apical and basolateral) led to a significant reduction of the denatonium-induced current in wildtype mice (n=5, * p<0.05, paired Student's t-test). **(F)** In the presence of the non-selective P2 purinergic receptor antagonist suramin (100 μM , basolateral) the denatonium-induced effect was diminished (n=6, * p<0.05, paired Student's t-test).

Reviewer 3:

Reviewer #3 (Remarks to the Author):

The manuscript by Abdel Wadood and collaborators provides a molecular mechanism initiated by Tuft cells to build an innate and adaptive immune response by the airway epithelium in response to *Pseudomonas aeruginosa* infection. The activation of the bitter taste sensor TRPM5 in tuft cells triggers Panx1-dependent ATP release which lead to dendritic cells activation and recruitment to the trachea and the lung. The strength of the manuscript is the use of a variety of in vivo and ex vivo models to support this concept. Although the amount of work is appreciated, the combination of so many models makes some figures difficult to interpret. Importantly, appropriate controls are not always shown and some experimental conditions are not well justified. This critically undermines some of the conclusions.

We thank the reviewer for their comments that have helped to improve the manuscript.

Major comments:

1) The introduction should bring a clear justification of evaluating ATP as mediator of Tuft cells' response. The links between TasR, TRPM5 and ATP release by CALHM or Pannexins is indeed well described in taste buds.

Reply: We thank the reviewer for this comment and have changed the introduction by adding more information describing taste bud signalling. We have also now explained the rationale behind our study of ATP release from TCs. For more details about the performed changes please also see the revised version of the introduction part of our manuscript (page 4/5 line 87-110). We hope that the justification for studying ATP as a TC mediator is now clearer.

2) The statistical analysis is sometimes difficult to understand. For instance, how the statistical significance is reached in Fig. S3C after Suramin treatment with such a large variability? In contrast in Fig. 3C the difference between sham and control is not significant? The statistical tests used should be indicated in the figure legends. Are the bars showing SD or SEM? Moreover, n should be indicated for each experimental condition (and not showing only the higher n...).

Reply: We thank the reviewer for this comment. In the former Fig. S3C, now Supplementary Fig. S5D-F in the revised version of the manuscript, we used the paired Student's *t*-test to compare the control effect of apical application of 1 mM denatonium to the denatonium-induced current changes in the presence of the respective inhibitors. Since the control denatonium-effect and the denatonium-effect in the presence of a given inhibitor are measured in the same trachea after a wash-out of denatonium after the first application, it is appropriate to use the paired Student's *t*-test for these analyses. In our previous work, we have shown that repeated apical application of 1 mM denatonium has no influence on the denatonium-induced current changes to the second application of denatonium (Hollenhorst et al., 2022, PMID: 35954259). We also performed two additional experiments with suramin and included them in Supplementary Fig. S5F of the revised version of the manuscript. In the revised version of the manuscript, we analysed the data in new Supplementary Fig. S5C by one-way ANOVA test, since in this data set, we compared the denatonium-induced current changes in different experimental groups (mouse strains) to the same control group of mice. In order to avoid false positive results, we used a one-way ANOVA test followed by the Bonferroni test for multiple comparisons and not the Student's *t*-test. For comparison of the data sets in Fig. 3C we used one-way ANOVA, since blood samples from different mice were compared. We have now included the information about the statistical tests in all figure legends. The bars in our figures show SEM as mentioned in the paragraph 'Statistical analyses' on line 635 in the methods section. We have also now included the respective n-number for experiments in all figure legends. We hope, these explanations help to mitigate the reviewer's concerns.

3) According to Methods (ATP sensor experiments and ATP measurements paragraphs), ATP was determined by ELISA from homogenized tracheas stimulated with denatonium. Thus, one can understand that ATP in supernatants from homogenized trachea mostly corresponds to intracellular ATP. Please explain how intracellular ATP is excluded or revise Methods to avoid confusion. How ATP release from tracheas is normalized? Can you estimate the ATP concentration in these supernatants? The number of TCs in tracheas is likely to be different because of the differential TCs expansion described in Fig. S5.

Reply: We are sorry about this confusion, however, as mentioned in the material and methods section, we used tracheal supernatants for the ATP measurements and did not use homogenized tracheas. Therefore, we are measuring only the extracellular ATP that is released into the fluid and not the intracellular ATP. We measure the ATP concentration in the supernatants using a commercial ELISA kit according to the manufacturer's recommendation. The concentrations are depicted as ng/ml (please see revised Fig. 1I and J). All supernatants that we processed for ATP-measurements had identical volumes. For these experiments the explanted tracheas were stimulated for 5 minutes *ex vivo*.

[text redacted]. Acute preparation of the trachea (1-2 hours) has no influence on the tracheal function (Hollenhorst et al., 2020, PMID: 31914675). Due to the short period of cultivation of the trachea in these experiments (5 min), we don't expect that there will be an expansion of tuft cells. **[text redacted]**

In our *in vivo* and *ex vivo* models we have observed no newly generated tuft cells 30 min after stimulation with tuft cell agonist. Since this topic is not directly related to the current manuscript, we prefer to not include this data set here. Therefore, we can conclude that the differences in ATP levels are not due to tuft cell expansion.

4) X-gal staining is poor and difficult to interpret. Without quantification/scoring and with only a single representative image provided, the interpretation of these data is ambiguous. In situ hybridization could be useful.

Reply:

As suggested by the reviewer, we have now quantified the LacZ staining in *Panx1*^{-/-} mice and observed a co-localization of LacZ with Trpm5 in 40% of the cells. This quantification is included in Supplementary Figure S4F of the revised version of the manuscript and we have integrated our findings in the text (page 8/line 169-170). The LacZ has only a perinuclear localization, therefore, the staining is restricted to a small area and does not extend to the cytoplasm of the whole cells. We hope that the quantification helps to better interpret the data. We refrained from performing in situ hybridisation for *Panx1* on *ChAT*-eGFP tissue, since this is not trivial and the protocol cannot be established in our hands within the short revision period. In addition, we propose that beta-galactosidase (LacZ-) staining of tissues derived from mice in which the bacterial lacZ gene has been introduced as a reporter gene is a well-recognised technique for studying gene expression at cellular level and therefore find that in situ is not necessary in this case. The beta-galactosidase staining comprises a simple and robust workflow to determine promoter activity of a gene of interest. In our study, we combined LacZ-staining immunohistochemistry as we have successfully done in previous studies (Maxeiner et al., 2003; PMID: 12809690). Moreover, the obtained results from the LacZ staining in *Panx1*^{-/-} mice correspond to the published work on the gene expression of tuft cells (Hollenhorst et al., 2020, PMID: 31914675; Bankova et al., 2018, PMID: 30291131; Montoro et al., 2018, PMID: 30069044).

5) The immunofluorescence in Fig. S2A2 is not convincing. Validation of the antibody by IF on *Panx1,2* KO mice should be shown. Please explain the endogenous eGFP fluorescence in all tracheal cells in Fig. S2A1?

Reply: We thank the reviewer for this comment. We have now validated the antibody in *Panx1*^{-/-} mouse tracheas and repeated the staining in *ChAT*-eGFP tracheas. We observed that the *Panx1*-immunoreactive cells in the tracheal epithelium were ChATeGFP+ tuft cells. In this mouse strain eGFP is expressed under the promotor for choline acetyltransferase, the synthesising enzyme of acetylcholine (ACh) (Tallini et al., 2006, PMID: 16940431). ChAT is one of the marker genes for tuft cells (Hollenhorst et al., 2020, PMID: 31914675; Bankova et al., 2018, PMID: 30291131; Montoro et al., 2018, PMID: 30069044). The information about the mouse strains can be found in the methods section of the manuscript (page 21/ line 511-532). We also included a short note on this within the results of the manuscript (line 116-118). This reads as follow: “We measured whole-cell currents in freshly isolated TCs from tissues of *Trpm5*^{+/+}-*ChAT*-eGFP and *Trpm5*^{-/-}-*ChAT*-eGFP mice. In both mouse strains, TCs can be identified by their green fluorescence.”

We also performed double-labelling experiments for *Panx1* and tuft cell marker (*ChAT* and α -gustducin) and found *Panx1*-immunofluorescence colocalized with the tuft cells markers. *Panx1* staining was absent in *Panx1*^{-/-} mouse tracheas (Supplementary Fig. S4). We included representative images from the newly performed staining in the revised Supplementray Fig. S4A-C.

revised Fig. S4A-C: (A) Immunofluorescence staining revealed overlay of pannexin 1 (*Panx1*) staining (white) and tuft cells (*ChAT*, cholineacetyltransferase, green, *Gnat3*, red) in a subpopulation of cells (arrows), but also some additional *Panx1* staining in *ChAT*-negative cells (star). (B-C) There was no *Panx1* staining (white) in tracheal sections of *Panx1*^{-/-} mice stained for *Gnat3* (A) or for *ChAT* (C) as TC markers (red, arrows).

6) Fig.3: how immune cells were detected? FACS? The methodology is not clearly described (BALF is also not described in Methods...). The writing can be improved to help the readers. How neutrophil counts were normalized in the tracheas?

Reply: We are sorry that there was no clear description of the collection of the bronchoalveolar lavage fluid (BALF) in the manuscript. We have now included a description

of BALF in the supplemental material and methods section of the manuscript (Supplement page 11/line 19-21).

The cell counts represented in Figure 3 are from the immune cells detected with FACS. We now added this information to the figure legend of Figure 3 and in line 227. Please see the revised figure legend (line 898). The neutrophil counts in the tracheas were not normalized. The data represented in the figure are the absolute cell numbers per trachea. We included this information in the labelling of the y-axis.

7) Important controls are missing to support the data in Figs. 4 and 5. Please add nontreated conditions for panels B/C and the control condition without *P. aeruginosa* for *Panx1*^{-/-} in panels C/F.

Reply: We thank the reviewer for this comment which has helped to improve the manuscript. We have now added the control conditions in the revised version of figures 4 and 5, such as controls for the dendritic cell counts and for the percentage of CD86+ dendritic cells in tracheas, lungs and lymph nodes.

The following controls were added:

- For the denatonium treated mice:
 - DC cell counts in tracheas, lungs and lymph nodes from *Panx1*^{+/+} and *Panx1*^{-/-} animals
 - DC cell counts in tracheas and lungs from *Chat*^{fl/fl}:*Trpm5*^{cre} mice
 - Percentage of CD86+ DCs of tracheas and lungs from *Panx1*^{+/+} and *Panx1*^{-/-} animals
 - Percentage of CD86+ DCs in tracheas and lungs from *Chat*^{fl/fl}:*Trpm5*^{cre} mice
 - T_H17 numbers and percentages in tracheas, lungs and lymph nodes from *Panx1*^{+/+} and *Panx1*^{-/-} animals
- For the infected mice:
 - DCs in tracheas and lungs from *Panx1*^{-/-} animals
 - Percentage of CD86+ DCs of tracheas and lungs from *Panx1*^{-/-} animals

For the afore mentioned, please refer to the revised versions of Fig. 4 and Fig. 5.

Figure 4: FACS analyses of dendritic cell (DC) numbers three days after tuft cell (TC) stimulation by denatonium (1 mM) or infection with *Pseudomonas aeruginosa* NH57388A. **(A)** DC (CD11b⁺, CD11c⁺, F4/80⁻) numbers in denatonium-treated or control tracheae of *Trpm5*^{+/+} (n=5-7) *Trpm5*^{-/-} mice (n=4-5) and *Chat*^{fl/fl}:*Trpm5*^{cre} mice (n=4-7) after three days. One-way ANOVA. **(B)** DC numbers in tracheae of control or denatonium-treated *Panx1*^{-/-} (n=5-6) and *Panx1*^{+/+} mice (n=6). One-way ANOVA. **(C)** DC numbers three days post infection or in uninfected controls in *Trpm5*^{+/+} (n=4-7), *Trpm5*^{-/-} (n=3-5) and *Panx1*^{-/-} mice (n=4-10). One-way ANOVA. **(D)** DC numbers in lungs of denatonium-treated (1 mM) and control *Trpm5*^{+/+} (n=5-12), *Trpm5*^{-/-} mice (n=6-15) and *Chat*^{fl/fl}:*Trpm5*^{cre} mice (n=5-7). One-way ANOVA. **(E)** DC numbers in lungs of denatonium-treated and control *Panx1*^{-/-} (n=4-5) or *Panx1*^{+/+} mice (n=5). One-way ANOVA. **(F)** DC numbers in lungs of infected *Trpm5*^{+/+} (n=4), *Trpm5*^{-/-} (n=3) and *Panx1*^{-/-} mice (n=10) and uninfected controls (n=4-9). One-way ANOVA. **(G)** DC numbers in airway lymph nodes of denatonium-treated *Trpm5*^{+/+} (n=7), control (n=8) and *Trpm5*^{-/-} mice (n=5-7). One-way ANOVA. **(H)** DC numbers in airway lymph nodes from denatonium-treated or control *Panx1*^{+/+} (n=5) and *Panx1*^{-/-} mice (n=5). One-way ANOVA. **(I)** Survival rate of *Trpm5*^{+/+} (n=10), *Trpm5*^{-/-} (n=26) and *Panx1*^{-/-} (n=11) mice revealed decreased survival rates in *Panx1*^{-/-} and *Trpm5*^{-/-} mice after infection with *P. aeruginosa* NH57388A in the first 72 hours. (* p<0.05, ** p<0.01, *** p<0.001)

8) L.291-295: sentences hard to read.

Reply: We have changed these sentences and hope they are now easier to read and more understandable. The sentences now read (page 15 line 362-368): “Next, we performed co-culture experiments using DCs isolated from *Panx1*^{-/-} mouse lungs and tracheal epithelial cells from *Trpm5*-DREADD mice to exclude the possibility that the reduction in DC activation in *Panx1*^{-/-} mice was not due to the absence of the channel in DCs³⁹. We observed an increase in the number of activated DCs (CD86⁺) after stimulation of TCs with CNO (Fig. 6G). In contrast, no change in CD86 was observed in control experiments where *Panx1*^{-/-} lung DCs were treated with CNO in the absence of TCs (Fig. 6G).”

9) The DC migration assay is not convincing. DCs treated with tracheal supernatants from different mice cannot be compared to non-treated DCs! The migration phenotype induced by each supernatant should be compared to its own control genotype! Is there any impact of the different supernatant on the proliferation of DCs? Why different confluences are observed between experiments? An impact on the proliferation could explain differences in wound repair.

Reply: We have followed the reviewer’s recommendation and performed additional experiments in which we used the corresponding genotype as a control for each experimental group. These data are now included in the revised version of Fig. 7. The newly added data support our previous conclusion that tuft cell-released ATP induces dendritic cell migration.

As proof the impact of TC stimulation on DC proliferation we have performed three additional experiments in which we quantified DC cell numbers after stimulation with supernatants for 24 h. [text redacted] the cell counts did not change significantly.

[text redacted]

As the result does not change the conclusions of our manuscript and these experiments are not at its focus, we would like to exclude these findings from this manuscript.

[figure redacted]

10) A more detailed explanation on Tuft cells’ expansion, and how it could be regulated by TRPM5, should be added in the Discussion.

Reply: We have followed the reviewer’s recommendation and changed the discussion accordingly. We added discussion regarding a possible regulatory mechanism for tuft cell

expansion in the airways, nevertheless, these are hypothetical assumptions, since the exact mechanism behind the tracheal tuft cell expansion in response to activation with tastants or bacterial products remains to be elucidated. The paragraph to tuft cell expansion (line 467-483) now reads as follows: "So far, TC-mediated expansion of DCs was attributed to CysLT- and IL-25-dependent signaling resulting in aeroallergen-evoked type 2 inflammation¹⁷, an experimental setting in which CysLTs also induced TC expansion¹⁸. Here, we extend these findings by demonstrating DC expansion and even activation after TC stimulation in bacterial infection. Moreover, the observed DC recruitment in pneumonia and to TC denatonium stimulation was dependent on Trpm5 and Panx1 signaling and correlated with an increased TC number.

[text redacted]

Minor comments:

1) Better images should be shown in Fig. 1A-C. eGFP fluorescence should be visualized in Fig1B-C. The data do not show that ATP released by TCs is responsible for Ca²⁺ increase but point only to a correlation. Data using TCs from Panx1^{-/-} and Trpm5^{-/-} mice could be used.

Reply: We thank the reviewer for this comment which has helped to improve the manuscript. As suggested, we have included visualization of eGFP fluorescence in the new Fig. 1D. Furthermore, we performed experiments with Trpm5^{-/-} TCs (from *Trpm5^{-/-}ChATeGFP* mice) and included the results in Fig. 1F. The increase in intracellular Ca²⁺ in these TCs did not evoke a response in ATP sensor cells, indicating that ATP is released from TCs. Unfortunately, to the best of our knowledge, there are currently no *Panx1^{-/-}* mice available in which tuft cells can be specifically identified and studied *in vivo*. This certainly an issue we would like to address in our future studies.

Revised Figure 1: Tuft cells (TCs) release ATP and express pannexin 1. (A) Representative I-V relationships of normalized TRPM5 current amplitudes (pA/pF) measured in a TC at indicated time points after breaking the cell. (B) Representative current-voltage (I-V) relationships of whole-cell currents measured in a tracheal TC from a *Trpm5*^{+/+} mouse or a *Trpm5*^{-/-} mouse in the presence of 1 μ M intracellular Ca^{2+} . (C) Statistical analysis (Unpaired Student's *t*-test) of currents illustrated in (A). Current amplitudes measured at -100 mV and +100 mV in *Trpm5*^{-/-} TCs (n=9) and *Trpm5*^{+/+} TCs (n=10) cells are shown. (D) The bright field of a patched TCs and ATP sensor cells. (Green cells: GFP-expressing TC) (E) Ca^{2+} imaging showing the beginning of recording, and (F) example of maximal fluorescent changes in cell membranes (arrow). (G) The statistics of maximal fluorescence intensity of all ATP sensor cells responding to TC stimulation in *Trpm5*^{+/+} mice (n=13 cells/5 mice/5 experiments) and *Trpm5*^{-/-} mice (n=19 cells/4 mice/4 experiments). Control: baseline fluorescence of ATP sensor cells, TC Ca^{2+} ; fluorescence in ATP sensor cells after TC stimulation with intracellular high Ca^{2+} ; (H) The time course of maximal fluorescence intensity of all ATP sensor cells responding to TC stimulation in *Trpm5*^{+/+} mice. (I) The statistics of ATP release from TCs in *Trpm5*^{+/+} mice (n=13 cells/5 mice/5 experiments) and *Trpm5*^{-/-} mice (n=19 cells/4 mice/4 experiments) under various conditions. (J) The statistics of ATP release from TCs in *Panx1*^{-/-} mice (n=13 cells/5 mice/5 experiments) and *ChAT*^{fl/fl}; *Trpm5*^{fl/fl} mice (n=19 cells/4 mice/4 experiments) under various conditions. (K-O) Immunohistochemistry (IHC) images showing the expression of KPanx1^{+/2+} (LacZ) and Trpm5 in tracheal TCs. Scale bar = 100 μ m.

paired Student's *t*-test (H) The fluorescence intensity of marked cells in (C) over the whole cell recording. (I) Supernatants from denatonium-treated tracheae of *Trpm5*^{+/+} mice (n=5) revealed higher ATP levels than those from *Trpm5*^{-/-} mice (n=5) and vehicle-treated tracheae (n=4-5) quantified by ELISA assays. Supernatants from clozapine-N-oxide (CNO, 100 μM) treated *Trpm5*-DREADD mice (n=5) contained higher ATP concentrations after stimulation compared to vehicle treated controls (n=5). unpaired Student's *t*-test (J) Supernatants from denatonium-treated tracheae from *Panx1*^{-/-} mice (n=3) revealed the same ATP levels as vehicle-treated controls (n=3) quantified by ELISA assays. Tracheal supernatants from denatonium-treated *ChAT*^{fl/fl}:*Trpm5*^{cre} mice (n=3) contained higher ATP levels than vehicle treated tracheae (n=3). unpaired Student's *t*-test (K-O) Immunohistochemistry for Trpm5 (TC marker) combined with LacZ-staining suggest partial but not exclusive co-expression of Trpm5 (brown) and Panx1/2 (perinuclear blue signal) in the tracheal epithelium of *Panx1*^{-/-}/*2*^{-/-} mice. Arrows indicate LacZ staining in the nuclei of Trpm5+ cells. (** p<0.01, ** p<0.01, *** p<0.001)

2) Only 13 cells were analyzed in Fig. 1F?

Reply: We analysed 13 cells for the *ChAT*eGFP mice and 19 cells from the *Trpm5*^{-/-} *ChAT*eGFP mice. The cells were isolated from n=5 *ChAT*-eGFP or n=4 *Trpm5*^{-/-} *ChAT*eGFP mice. The represented data are from at least four different independent experiments. We included this information in the figure legend of revised Fig. 1G (line 855-859). It reads now as follows: "The statistics of maximal fluorescence intensity of all ATP sensor cells responding to TC stimulation in *Trpm5*^{+/+} mice (n=13 cells/5 mice/5 experiments) and *Trpm5*^{-/-} mice (n=19 cells/4 mice/4 experiments). Control: baseline fluorescence of ATP sensor cells, TC Ca²⁺_i: fluorescence in ATP sensor cells after TC stimulation with intracellular high Ca²⁺_i; paired Student's *t*-test"

3) Brush cells should be replaced by tuft cells for nomenclature coherence through the text (legends of Fig. S2 and S4).

Reply: We agree with the reviewer and have replaced the term "brush cell" with "tuft cell" throughout the manuscript.

4) Patch-clamp quantification should be shown for Fig. 1D. Please normalize the current intensity to the capacitance (pA/pF). Differences in cells' size (membrane surface) may explain the variability observed in Fig. 2C.

Reply: We appreciate the reviewer's comment. In response, we have normalized the currents of all recorded cells to their membrane capacitances, as shown in Fig. 2C. While some variation is present, the currents were sensitive to TPPO. During the recordings, denatonium was focally applied to the bath for 3 to 5 minutes once the cell currents were stable, followed by the application of TPPO. The observed variation in currents may be attributed to differences in Trpm5 channel expression in tuft cells. To date, we have recorded more than 50 tuft cells (also in other projects) and consistently detected Trpm5 currents in all analyzed tuft cells. Additionally, we normalized the current density at +100 mV for all cells 10 minutes after membrane rupture, as shown in Supplementary Fig. S1A. We observed that the currents peaked within 3 to 4 minutes as intracellular Ca²⁺ ions diffused into tuft cells. This was followed by ATP release, as confirmed through Ca²⁺ imaging experiments

Fig. S1: (A) Trpm5 current density of tuft cells at 100 mV over time after breaking the cell in the whole cell mode. (n=11 cells)

Fig. 2: (C) The current densities recorded at 100 mV were increased by denatonium in TCs of *Trpm5*^{+/+} mice compared to before treatment and to currents in the presence of 100 μ M TPPO (n=4, paired Student's *t*-test). The currents at 100 mV of *Trpm5*^{-/-} TCs (n=11) perfused with 1 mM denatonium were reduced compared to TCs from *Trpm5*^{+/+} mice (unpaired Student's *t*-test).

5) The closing sentence in L.153-154 is an overstatement as the Ca²⁺ data are not clear cut.

Reply: We thank the reviewer for this comment. We have included patch clamp experiments with primary TCs (*ChATeGFP* or *Trpm5*^{-/-}*ChATeGFP*) in Figure 1B and C. The data show that the increase in intracellular Ca²⁺ in wild-type TCs induces currents which are not detectable in *Trpm5*-deficient TCs (Fig. 1B). The quantification of these experiments is shown in Fig. 1C. Together with the results shown in the new Fig. 2G, we have shown that the increase of Ca²⁺ in the intracellular patch clamp solution increases the currents at 100 mV in wild-type TCs and not in *Trpm5*-deficient TCs. Moreover, these currents could be abolished by the application of CBX and probenecid, both inhibitors of Panx1 in the concentration in which we used them. Taken together, we find that our conclusion that there are functional Panx1 channels in TCs that are connected to *Trpm5* channel activity are supported by the presented data.

6) It is surprising that a small proportion of Tuft cells can release sufficient amount of ATP, despite the dilution imposed by the Ussing system, to activate purinergic-induced Isc currents. Please show traces of the recordings.

Reply: We have now included representative traces of the recordings from a wild-type (*Panx1*^{+/+}) mouse trachea and a *Panx1*^{-/-} mouse trachea in the revised version of Supplementary Fig. S5 (Fig. S5B), showing the denatonium-induced current changes. We also think that the robust effect that we detected in our Ussing chamber experiments is very intriguing. Recently Perniss et al., (2023, PMID: 37531421), described that a long-range Ca²⁺ wave spreading radially over the tracheal epithelium stimulated tuft (brush) cells then triggered by a stimulation of TCs with succinate. Succinate led to release of ACh from TCs, which excites nearby cells via muscarinic acetylcholine receptors. From there, the Ca²⁺ wave propagates through gap junction signaling, reaching also distant ciliated and secretory cells. Single-cell RNA sequencing data from Plasschaert et al., (Plasschaert et al., 2018, PMID: 30069046, Perniss et al., 2023, PMID: 37531421) revealed abundant expression of purinergic receptors (P2rx2, P2rx4, P2rx7, P2ry1, and P2ry2) in secretory cells and P2rx 4 and P2rx7 in ciliated cells. Here, we show that stimulation of TCs with denatonium leads to ATP release and acts most probably in a paracrine manner on ciliated or secretory cells.

[text redacted]

Nevertheless, at this point we can only speculate about this and hope our explanations help to mitigate the reviewer's concerns.

Fig. S5: Tuft cell released ATP modulates transepithelial ion transport. (A) Schematic drawing of the action of denatonium on ATP release. (B) Representative current traces (I_{sc}) of a wildtype ($Panx1^{+/+}$) and of a $Panx1^{-/-}$ mouse trachea upon apical application of 1 mM denatonium (den). (C) In $Panx1^{-/-}$ ($n=4$) and $Panx1^{-/-}2^{-/-}$ mice ($n=4$) the denatonium-induced current (1 mM, apical, ΔI_{sc}) was reduced compared to wild-type controls ($Panx1^{+/+}$, $n=11$, *** $p<0.001$), while the denatonium-induced effect in $Panx2^{-/-}$ mice ($n=4$) was similar to wildtype mice. (one-way ANOVA) (D) The CALHM1-inhibitor Ruthenium Red (RuR, 20 μ M, apical and basolateral) did not change the denatonium-induced current compared to control conditions ($n=4$, ns: not significant, paired Student's t -test). (E) Application of the pannexin inhibitor probenecid (200 μ M, apical and basolateral) led to a significant reduction of the denatonium-induced current in wildtype mice ($n=5$, * $p<0.05$, paired Student's t -test). (F) In the presence of the non-selective P2 purinergic receptor antagonist suramin (100 μ M, basolateral) the denatonium-induced effect was diminished ($n=6$, * $p<0.05$, paired Student's t -test).

7) Line 171-172: this sentence does not seem at the right place!

Reply: We have better contextualised this sentence now by adding more information. The two sentences now read (line 223-225): "To study the TC-induced effects on innate immunity longitudinally we first established a model to specifically activate TCs *in vivo*. First, we performed Ca^{2+} -imaging experiments in freshly isolated primary TCs from *Trpm5*-DREADD mice to confirm their specific activation by CNO."

8) Fig. S4: how TCs were identified, as they were not eGFP-positive? Please clarify how Ca^{2+} recording was made as not only TCs are likely monitored! The CNO concentration is different from the one mentioned in the methods (60 μ M vs 100 μ M).

Reply: We thank the reviewer for this comment. We have now corrected the CNO concentration for the Ca^{2+} -imaging experiments with primary single tuft cells in the supplemental methods to 60 μ M. We used a concentration of 100 μ M CNO in our *in vivo* and *ex vivo* tracheal preparations experiments. In the original version of the submitted manuscript, we stained the cover slips with the *Trpm5*-DREADD expressing tuft cells after the Ca^{2+} -imaging experiments in order to identify them. In the revision process we bred a

Trpm5-DREADD-tGFP mouse strain (Yu et al., 2023 PMID: 36949050). In these mice, tuft cells can be easily identified by their green fluorescence prior to performing Ca^{2+} -imaging experiments. Therefore, we repeated these experiments with this new mouse strain and exchanged the panel in the figure. As previously observed, tuft cells, now identified by their green fluorescence, reacted to 60 μM CNO with an increase in intracellular Ca^{2+} . Please find the new data in Supplementary Fig. S6.

Fig. S6: Ca^{2+} -Imaging in *Trpm5*-DREADD mice stimulated with CNO. (A) Representative curve of $[\text{Ca}^{2+}]_i$ levels of a tuft cell reacting to CNO (60 μM). (B) CNO (60 μM) led to a significant transient increase in $[\text{Ca}^{2+}]_i$ levels in tuft cells from *Trpm5*-DREADD-tGFP mice (n=10 cells from 3 mice). (***) $p < 0.001$, paired Student's *t*-test)

9) Why TRPM5-DTA mice were not described or used before. This model would have been useful to support previous conclusions.

Reply: We agree with the reviewer that *Trpm5*-DTA mice are a useful model. We described the model in Hollenhorst et al., (2022, PMID: 35503420). In this work, we used the model to study the involvement of tuft cells in the induction of protective neurogenic inflammation in respiratory tract as a first line innate immune response to bacterial infection.

10) The loading control (GAPDH or b-actin) is missing in Fig. 6A.

Reply: We apologize for not providing an image for the loading control. In response to your feedback, and to enhance the clarity and presentation of our manuscript, we have now incorporated an image from Actin as a loading control in Supplementary Figure S3. A band for Actin at the expected size of approximately 42 kD was detected in all samples. As visible from this image, the amount of Actin in the *Chat^{fl/fl}; Trpm5^{Cre}* mice is comparable to the protein amount in the wild-type mice. Please see the revised version of Supplementary Fig. S3A.

Fig. S3A: Validation of the *ChAT^{fl/fl}:Trpm5^{cre}* mouse model. (A) Immunoblot for ChAT in *ChAT^{fl/fl}:Trpm5^{cre}* and wild-type mice (WT) showed ChAT protein in wild-type mice at a size of approximately 72 kD, but not *ChAT^{fl/fl}:Trpm5^{cre}* mice. A band for Actin at the expected size of approximately 42 kD was detected in all samples.

11) The scenario for the resolution of inflammation does not sound realistic.

Reply: The scenario for the role of ATP or rather its metabolite adenosine in inflammation is hypothetical. However, we believe that tuft cells are also involved in the limitation of inflammation and not only in the induction. Therefore, we added this hypothesis to our manuscript. There are several hints for this, although the mechanism remains to be elucidated. We have previously observed that tuft cell stimulation led to a prominent increase in serpins (published in Hollenhorst et al., JCI, 2022, PMID 35503420), some of which may have anti-inflammatory effects e.g. by alpha-1-antitrypsin by inhibiting neutrophil elastase.

[text redacted]

Additionally, we cannot rule out that other tuft cell-derived mediators might also be involved or responsible for

this. This still remains to be elucidated in future studies. In order to make this clearer, we have added to the manuscript that this scenario is hypothetical (page 21 line 503).

We hope this revision meets your expectations and provides a clearer context for our findings. Thank you for your patience and understanding.

Reviewer 4:

Reviewer #4 (Remarks to the Author):

Reply: We thank the reviewer for co-reviewing our manuscript.

Reviewer 3:

1) Abstract, l.49: "stimuli".

Reply: We apologize for this typing error and corrected it in the revised version of the manuscript (please see page 3 line 48).

2) Fig.1A: thank you for normalizing the current in the I-V shown in A (and some other Figures). The point of normalization is to compare I-V relations in TCs with different genotypes for quantitative analysis. Please correct Fig.2E, 2F and 2G.

Reply: We thank the reviewer for this comment. Since the cell size remains the same in our experiments, we didn't normalize the currents in the original figures in our manuscript. Nevertheless, we agree with the reviewer that a reliable comparison of whole-cell currents is possible only after normalization of currents to pF (cell size). To address his point, we recalculated the data in Figure 1C and Figure 2E, F and G show them as pA/pF. A corresponding sentence has been added to MM (please see page 3 line 5-10). It now reads: "The data are represented as F/F_0 . F_0 refers to the fluorescence intensity of the recorded ATP sensor cells in the beginning (baseline in experiments performed without simultaneous patch clamp recordings or before whole-cell configuration of tuft cells in measurements with simultaneous patch clamp recordings). F refers to the fluorescence intensity of ATP sensor cells after stimulation at different time points or the maximum response depending on the respective experiment." Now, all figures showing whole-cell currents are presented as current densities, calculated by dividing the measured current by the respective cell membrane capacitance.

Please find the revised Figure 1 and 2 and below:

Fig. 1 Tuft cells (TCs) release ATP and express pannexin 1. (A) Representative I-V relationships of TRPM5 current amplitudes normalized to cell size (pA/pF) measured in a TC at indicated time points after breaking the cell. (B) Representative current-voltage (I-V) relationships of whole-cell currents measured in a tracheal TC from a *Trpm5*^{+/+} mouse or a *Trpm5*^{-/-} mouse in the presence of 1 μ M intracellular Ca^{2+} . (C) Statistical analysis (unpaired Student's t-test) of currents illustrated in (B). Current amplitude densities measured at -100 mV and +100 mV in *Trpm5*^{-/-} TCs (n=9 cells) and *Trpm5*^{+/+} TCs (n=10 cells) are shown. (D) The bright field of a patched TCs and ATP sensor cells shown in E-F. Green cells = GFP-expressing TC. (E-F) Representative images from a recording of the membrane fluorescence of two ATP sensor cells before (E) and after (F) application of Ca^{2+} containing intracellular solution into a TC cell through the patch pipette. Arrows point to the cell membranes of ATP sensor cells (G) The fluorescence intensity of all ATP sensor cells over the whole-cell recording of *Trpm5*^{+/+} (black, n=11) or *Trpm5*^{-/-} TCs (blue, n=9). (H) Statistical analyses of maximal fluorescence intensity of all ATP sensor cells responding to TC stimulation in *Trpm5*^{+/+} mice (n=13 cells/5 mice/5 experiments) and *Trpm5*^{-/-} mice (n=19 cells/4 mice/4 experiments). Control = baseline fluorescence of ATP sensor cells, TC Ca^{2+} i = fluorescence in ATP sensor cells after TC stimulation with intracellular high Ca^{2+} ; paired Student's t-test. (I) Supernatants from denatonium-treated tracheae of *Trpm5*^{+/+} mice (n=5) revealed higher ATP levels than those from *Trpm5*^{-/-} mice (n=5) and vehicle-treated tracheae (n=4-5), quantified by ELISA assays. Supernatants from clozapine-N-oxide (CNO, 100 μ M)-treated *Trpm5-DREADD* mice (n=5) contained higher ATP concentrations after stimulation compared to vehicle-treated controls (n=5). Unpaired Student's t-test. (J) Supernatants from denatonium-treated tracheae from *Panx1*^{-/-} mice (n=3) revealed the same ATP levels as vehicle-treated controls (n=3), as quantified by ELISA. Tracheal supernatants from denatonium-treated *ChAT*^{fl/fl}:*Trpm5*^{cre} mice (n=3)

contained higher ATP levels than vehicle-treated tracheae (n=3). Supernatants from denatonium-treated tracheae from *Trpm5*-DTA mice (n=4) had the same ATP levels as vehicle-treated controls (n=4). Unpaired Student's t-test. (K-O) Immunohistochemistry for Trpm5 (TC marker) combined with LacZ-staining suggests partial but not exclusive co-expression of Trpm5 (brown) and Panx1/2 (perinuclear blue signal) in the tracheal epithelium of *Panx1*^{-/-}/*2*^{-/-} mice. Arrows indicate LacZ staining in the nuclei of Trpm5+ cells. ** p<0.01, *** p<0.001, ns: not significant.

Fig. 2 Whole-cell patch clamp recordings of mouse tracheal tuft cells (TCs). (A) Application of denatonium (1 mM) in *Trpm5*^{+/+} TCs led to an outwardly rectifying current that was abolished by TPPO (100 μ M). (B) Application of denatonium (1 mM) had no effect on the I/V curve recorded in *Trpm5*^{-/-} TCs. Red = denatonium, black = baseline. (C) The current densities recorded at 100 mV were increased by denatonium in TCs of *Trpm5*^{+/+} mice compared to before treatment and to currents in the presence of 100 μ M TPPO (n=4, paired Student's t-test). The currents at 100 mV of *Trpm5*^{-/-} TCs (n=11) perfused with 1 mM denatonium were reduced compared to TCs from *Trpm5*^{+/+} mice (unpaired Student's t-test). (D) Representative current traces at a holding potential of +30 mV hyper-polarized to -60 mV upon application of 60 mM KCl and addition of 300 μ M probenecid (prob). (E) KCl-induced current densities were inhibited by carbenoxolone (CBX, 10 μ M) or probenecid (prob, 300 μ M) in TCs from *Trpm5*^{+/+} and *Trpm5*^{-/-} mice. *Trpm5*^{+/+}: CBX n=3, prob n=9, *Trpm5*^{-/-}: CBX n=8, prob n=8; paired Student's t-test. (F) The KCl-induced current densities in *Trpm5*^{+/+} (n=13) and *Trpm5*^{-/-} (n=16) mice were identical (unpaired Student's t-test). (G) Analyzed current densities at 100 mV upon a ramp depolarization. Carbenoxolone (CBX, 10 μ M)-sensitive

currents upon the activation of Trpm5 channels with 110 nM $[Ca^{2+}]_i$ were reduced in *Trpm5*^{+/+} but not in *Trpm5*^{-/-} mice. Trpm5 currents at 110 nM $[Ca^{2+}]_i$ and 2.4 μ M $[Ca^{2+}]_i$ were sensitive to probenecid (prob, 300 μ M) in *Trpm5*^{+/+} but not in *Trpm5*^{-/-} mice. CBX: *Trpm5*^{+/+} n=9, *Trpm5*^{-/-} n=10; prob, 110 nM Ca^{2+} : *Trpm5*^{+/+} n=12, *Trpm5*^{-/-} n=7; prob, 2.4 μ M Ca^{2+} : *Trpm5*^{+/+} n=13, *Trpm5*^{-/-} n=8; paired Student's t-test. (H) The Ca^{2+} -induced current densities in TCs (ctrl) were not inhibited by the P2X7 receptor inhibitor AZ10606121 (20 μ M) (n=8). Paired Student's t-test. * $p < 0.05$, ns: not significant.

3) L.127-129: It is stated that Ca^{2+} changes are recorded in the ATP sensor cells while performing patch-clamp experiments on TCs. I guess you mean recording of FOF1-ATPase fluorescence changes upon ATP binding, not Ca^{2+} . Please explain or correct. The legend of Fig.1E is also misleading. An additional control/representative image using *trpm5*^{-/-} TCs should be shown. The quality of the images is very low, so that the fluorescence at the membrane is barely visible.

Reply: We apologize for the misleading description of the experimental result. We agree with the reviewer that the fluorescence changes of the ATP sensor were measured, not the increase in intracellular Ca^{2+} . Upon ATP binding, the fluorescence of the GFP coupled to the ATP sensor increases (Lobas et al. 2019, PMID: 30755613). We now corrected this information in the text (please see page 6 line 128) as well as in the figure legend of Figure 1E. This now reads: "(E-F) Representative images from a recording of the membrane fluorescence of two ATP sensor cells before (E) and after (F) application of Ca^{2+} containing intracellular solution into a TC cell through the patch pipette. Arrows point to the cell membranes of ATP sensor cells." We have now included representative images of a recording of ATP-sensor cells with *Trpm5*^{-/-}-*ChaT-eGFP* TCs in the Supplementary Figure 1C, D and E. We agree with the reviewer that the changes in the fluorescence at the cell membrane are delicate, which is expected since the fluorescence increase is highly specific and localized. These changes can be noticed and measured in experiments with *Trpm5*^{+/+}-*ChAT-eGFP* tuft cells. Please see the revised and Supplementary Figure 1 below.

Fig. S1: (A) Trpm5 current density of *Trpm5*^{+/+} (black) or *Trpm5*^{-/-} (blue) tuft cells at 100 mV over time after breaking the cell in the whole-cell mode. *Trpm5*^{+/+}: n=11 *Trpm5*^{-/-}: n=3. (B) A representative IV curve of Trpm5 currents in a tuft cell induced by 1 μ M Ca²⁺ (black curve) and in the presence of the Trpm5 specific antagonist TPPO (red curve). (C) The bright field image of a patched tuft cell of a *Trpm5*^{-/-}-*ChAT*-eGFP mouse and ATP sensor cells (arrows). Green cells = GFP-expressing TC. (D-E) Representative images from a recording of the membrane fluorescence of two ATP sensor cells before (D) and after (E) stimulation of a TC with intracellular Ca²⁺. No changes in fluorescence were observed. Arrows point at cell membranes of the ATP sensor cells at the same time point as depicted in Fig. 1F.

4) Legend of Fig.1H: there are no marked cells in C). Does the trace represent the average of two recordings? Data are shown as F1/F0 while they are expressed as deltaF in supplemental Fig.S2. Which one is correct as it does not mean the same. Does time 0 represents when the whole-cell configuration was established in TCs? The legend needs to be improved or remove panel H.

Reply: We thank the reviewer for this comment. We have now completely revised this panel. It now displays the F/F₀ of 11 ATP sensor cells from recordings with *Trpm5*^{+/+} TCs (black) and 9 ATP sensor cells from recordings with *Trpm5*^{-/-} TCs (blue). The data depicted in Figure 1H and supplemental Fig.S2 were recorded on two different setups by two different scientists. The fluorescence intensity changes were calculated as F/F₀ (fluorescence after/fluorescence before treatment) in Figure 1H and as Δ IF (fluorescence after–fluorescence before treatment) in Figure S2. In principle, both calculation methods are correct. However, to ensure consistency in data presentation, we have adjusted the presentation of the values in Figure S2 to align with those in Figure 1H. Please see the revised version of Figure S2 below. Additionally, we have performed changes in the supplementary MM section of the manuscript (please see page 3 / line 5-10). It now reads: “The data are represented as F/F₀. F₀ refers to the fluorescence intensity of the recorded ATP sensor cells in the beginning (baseline in experiments performed without simultaneous patch clamp recordings or before whole-cell configuration of tuft cells in measurements with simultaneous patch clamp recordings). F refers to the fluorescence intensity of ATP sensor cells after stimulation at different time points or the maximum response depending on the respective experiment.”

Time point 0 represents the time after breaking the cell membrane to achieve whole-cell configuration.

Fig. S2: (A) Dose-response curve of ATP sensor cells (HEK293 cells transfected with the ATP sensor iATPSnFR). The cells showed an EC₅₀ of 83.9 μ M when stimulated with a defined ATP concentration. **(B)** Supernatants of tracheas treated with 1 mM denatonium (den, *Trpm5*^{+/+}; n = 290) or vehicle (*Trpm5*^{+/+}; n = 169) revealed an increase in fluorescence (F/F₀) in ATP sensor cells transfected with the sensor iATPSnFR1.1 after denatonium treatment. (* p<0.05, unpaired Student's *t*-test)

5) Line 169-172. Can you speculate on how 40% of TCs expressing Panx1 lead to such an increase in ATP release? Is there any evidence that Panx1 expression is regulated by denatonium?

Reply: Although there seems to be only a subpopulation of tuft cells that express Panx1, this subpopulation is responsible for the ATP effects observed in our study. There are few possibilities which appear plausible in light with the current knowledge and the results from our study. Though, the exact mechanism warrants thorough investigations. When analyzing Panx1 currents in *Trpm5*⁺ cells by whole-cell patch clamp we observed even a higher percentage of cells that displayed Panx1 currents. 13 out of 18 tuft cells showed Panx1 antagonist-sensitive (either probenecid or carbenoxolone) currents to low intracellular Ca²⁺ levels (physiological levels) and 8 out of 13 tuft cells displayed Panx1 antagonist-sensitive currents to high intracellular Ca²⁺ levels, both with maximum at 100 mV. For better visualization of these results, we included in this response letter an additional figure (please see below), showing the percentage of whole-cell currents sensitive to probenecid or carbenoxolone. Thus, depending on the respective stimulus used, 60% to 70% of the patched tuft cells showed Panx1 currents and thus, express Panx1. We think that the differences in the percentage of Panx1-expressing tuft cells estimated by the LacZ staining and by the functional assay (patch clamp) might be caused by 1) the localized perinuclear staining of the LacZ in cells (some cells might appear LacZ negative when this region is present in the tissue section), 2) the Panx1 expression might not be detected when the expression level is low and 3) the higher sensitivity of the patch clamp technique to record even weak currents. However, it is tempting to speculate that a mechanism that could comprise for the ATP effects after the activation of a small subpopulation of tuft cells might be a Ca²⁺ wave that rapidly spreads throughout the epithelium and thereby activates other epithelial cells and thus, augments the signal induced by the ATP release. Supportive of this hypothesis, Perniss et al. (2023, PMID: 37531421) described a similar mechanism, by which acetylcholine is released from tuft cells, which then induces a paracrine Ca²⁺ wave in adjacent epithelial cells, which spreads throughout the epithelium, originating from the cells surrounding the tuft cells. In line with this, we also observed a paracrine effect of the tuft cell-released ATP in our recordings of transepithelial ion currents, since the denatonium-induced currents were reduced with suramin. Please refer to Supplementary Figure S5F. So far, there is no evidence that denatonium also upregulates expression of Panx1. Since ATP release was observed as an acute response to tuft cell stimulation (please see Fig. 1) we do not assume that the effects are due to changes in Panx1 expression. Moreover, ATP release from tuft cells was also observed after a rise in intracellular Ca²⁺ in patched tuft cells, which is independent from denatonium stimulation. Nevertheless, we cannot exclude the possibility that tuft cell stimulation by denatonium or other stimuli can evoke expressional changes in the following hours or days after stimulation. This is certainly an interesting aspect for future studies.

Additional Figure 1: Percentage of whole-cell currents that were sensitive to carbenoxolone (CBX) and probenecid (prob) at a low intracellular Ca²⁺ concentration (110 nM, black, n=18) or at high intracellular Ca²⁺ concentrations (2.4 µM, blue, n=13) at 100 mV. Unpaired Student's t-test.

6) Supplemental Methods, ATP sensor experiments, I.10: what means exactly tracheas were homogenized? In the reply to Reviewer 3, you state that tracheas are not homogenized for extracellular ATP measurement. Are they or not homogenized? Please, clarify.

Reply: We apologize for this error. As mentioned previously, tracheas used for ATP measurements were not homogenized for the ELISA measurements in order to be able to differentiate between intra- and extracellular ATP. For the ATP sensor experiments shown in Figure S2B tracheas were homogenized in order to measure the complete ATP concentration in the trachea after stimulation with denatonium. All other experiments conducted with ATP sensor experiments were performed with supernatants e.g. without homogenizing the tracheae.

7) L.136-138: I believe that tracheal supernatants were not treated with denatonium but tracheas were, and supernatants collected. Please correct sentence.

Reply: We apologize that the wording of this sentence was not clear. Indeed, the explanted tracheas were treated with denatonium, not the supernatants. We changed this sentence now. It now reads: "ATP release from explanted denatonium-treated wild-type tracheae was also verified by the response of ATP sensor cells treated with these supernatants (Supplementary Fig. 2B)".

8) It is stated that activation of tuft cells leads to trpm5-dependent K efflux, resulting in depolarization of the plasma membrane and opening of Panx1, independently of P2X7 receptor activation (answer to Reviewer 2). If I am correct, K efflux should lead to hyperpolarization, not depolarization of the cell membrane, so this mechanism does not hold. The link between Trpm5 and Panx1 is likely more complex than proposed here. Please explain.

Reply: We apologize if our previous response to Reviewer 2 was unclear to you. To the best of our knowledge, Panx1 can be activated by various stimuli, including high intracellular Ca^{2+} , KCl (K^+ extracellular), etc. (López et al., 2021, PMID: 34301850). In Figure 2, we used KCl solely to demonstrate Pan1 activity in tuft cells. The use of high KCl to activate Panx1 channels is widely accepted approach (Wang et al., 2014, PMID: 25056878). Thus, we see no inconsistency in our findings or proposed model.

We did not claim that K^+ efflux is downstream of Trpm5 activation. However, K^+ efflux occurs during Trpm5 activation in other cells (Prawitt et al., 2003 PMID: 14634208). We stated that ATP released via Panx1 has been reported to activate the purinergic receptor P2x7, leading to additional K^+ efflux and further activation of Panx1 (line 198-200, Xu et al. 2020, PMID 33424864). Therefore, we tested this hypothesis experimentally. The last sentence of the paragraph should read “These findings suggest that P2x7 does not significantly contribute to the Ca^{2+} -induced currents in TCs, which are required for the Trpm5 activation that drives Panx1-dependent currents” (line 207-208).

From our findings, it became evident that high intracellular Ca^{2+} triggered by stimulation of tuft cells (e.g., via Tas2R GPCRs), does not lead directly to the opening of Panx1 and involves Trpm5 activation (Figure 2G). It is well established that the Ca^{2+} -dependent opening of TRPM5 leads to Na^+ influx and plasma membrane depolarization, potentially influencing voltage-sensitive ion channel activity (Hofmann et al. 2003, PMID: 12842017, Kaske et al., 2007, PMID: 17610722). Whether voltage-sensitive ion channels contribute to channel activation remains is beyond the scope of this manuscript and warrants further studies.

9) The synergy between Trpm5 and Panx1 is not fully supported by the experiment shown in Fig. 2G. Why $2.4\mu\text{M}$ intracellular Ca^{2+} was applied here instead of $1\mu\text{M}$, as used in Fig.1B? How can you distinguish between Trpm5 and Panx1-mediated currents using this protocol? Why CBX reduced the current in low calcium and probenecid only in high calcium? Better contextualization and interpretation of these experiments are needed.

Reply:

Activation of the Panx1 channel can be obtained by various stimuli. The majority of the physiological stimuli involve ligands, including ATP itself, glutamate, α -adrenergic agonists, bradykinin, binding to their receptors and leading to opening of Panx1 channels and ATP release (Dahl 2015, PMID: 26009770, Dahl 2018, PMID: 29802622, Mim et al., 2021, PMID: 33835130). Other work indicates, that Panx1 channels can be activated by Gq protein-coupled receptors (GPCRs), i.e. P2Y purinergic receptors and ion channels. In the most cases the channel activation is mediated by an increase in $[\text{Ca}^{2+}]_i$ (Yang et al., 2022 PMID: 35163442).

In our work, we used two different Ca^{2+} concentrations to measure Panx1 currents, as it has been reported that voltage-dependent stimuli can evoke low conductance of Panx1 channels while high intracellular Ca^{2+} and high extracellular K^+ can activate high conductance of Panx1 channels in concentration-dependent manner (Yang et al., 2022, PMID: 35163442). As mentioned in our reply to comment 5, we observed Panx1 currents in 60-70% of the tuft cells when we applied low or high Ca^{2+} concentrations. However, high extracellular K^+ could induce Panx1 currents in all recorded cells (Figure 2E). Therefore, we conclude that Panx1 could be functionally activated by various mechanisms. Moreover, in our experiments, we found that Trpm5 is crucial for Panx1 activation in tracheal tuft cells. Depletion of Trpm5 abolished the currents at low (110 nM, physiological) and at high (2.4 μM) intracellular Ca^{2+} concentrations. Nevertheless, KCl was able to induce Panx1 currents in tuft cells from *Trpm5*^{-/-} mice, which is supportive for a voltage-gating of Panx1 in tracheal tuft cells. In line with our observations, several groups have reported ATP release and/or activation of Panx1 currents after exposure of cells to high extracellular K^+ concentrations (Silverman et al., 2009, PMID: 19416975; Santiago et al., 2011 PMID: 21949881; Heinrich et al., 2012, PMID: 22394324; Suadicanì et al., 2012, PMID: 22499153; Michalski and Kawate, 2016, PMID: 26755773; Qu et al., 2020). However, a lack of K^+ -mediated activation of Panx1 has also been reported (Chiu et al., 2018, PMID: 29233884; Nielsen et al., 2020, PMID: 31698505). Moreover, in type II taste cells deficient for Trpm5, taste stimulation elicited robust Ca^{2+} transients but failed to trigger ATP secretion (Huang and Roper, 2010, PMID: 20498277). Thus, Ca^{2+} signal alone was unable to liberate ATP. Nevertheless, these the cells were still capable of releasing ATP on high KCl (Huang and Roper, 2010, PMID: 20498277). Taken together, it is tempting to speculate that opening of Trpm5 leads to membrane depolarization of the cells and activates voltage-gated ATP-permeable channels, thereby initiating ATP release.

Since previously the probenecid-sensitive currents in the presence of 110 nM intracellular Ca^{2+} (ramp from -100 mV to +100 mV) were very close to significance ($p=0.08$), we performed additional experiments. After increasing the number of experiments, we observed a significant inhibition of the current by probenecid. All currents observed with the low (110 nM, physiological) or the high Ca^{2+} concentration (2.4 μM) are now sensitive to carbenoxolone or probenecid upon application of the respective substance. Therefore, we can conclude that the observed currents are Panx1 currents. Moreover, the 2.4 μM intracellular Ca^{2+} solution did not induce the reduction of the success rate of the whole-cell configuration. Therefore, the high Ca^{2+} concentration was not harmful for the cells.

We used a higher concentration than in Fig. 1B, because here we wanted to induce the maximal Trpm5 current. Our rationales were that this would also induce strong Panx1 currents which would become evident after the application of the CBX and prob. In Fig. 1B we used moderate Ca^{2+} concentration which indeed leads to induction of Trpm5 currents. Nevertheless, the Trpm5 currents induced by denatonium were stronger than those induced by 1 μM Ca^{2+} which again led us to the assumption that currents induced by 2.4 μM Ca^{2+} might mimic denatonium stimulation (GPCRs) and better represent the physiological *in vivo* situation.

Using a ramp protocol and Ca^{2+} , along with the application of inhibitors and the use of *Trpm5*^{-/-} mice, we can distinguish between Trpm5 and Panx1 currents in tuft cells. In *Trpm5*^{+/+} mice, Panx1 currents could be inhibited by the inhibitors prob and CBX.

Additionally, KCl induced a current in tuft cells of *Trpm5*^{-/-} mice, which was completely sensitive to prob and CBX. Interestingly, in *Trpm5*^{-/-} mice and after TPPO application, *Trpm5* currents as well as *Panx1* (and probably also to date unknown *Trpm5*-dependent channels) were inhibited. Nevertheless, since the IV curve for *Trpm5* in our experiments with denatonium and 1 μ M Ca^{2+} aligns with the published IV curve for the channel (Hoffmann et al., 2003, PMID: 12842017, Prawitt et al., 2003, PMID: 14634208), we assume that in this experimental setting, we primarily detect *Trpm5* currents. To the best of our knowledge, there is no better way to provide a direct evidence of channel activity in primary freshly isolated cells.

While the term synergy, which we explicitly did not use in the manuscript, might be a very strong word, the data presented in our manuscript support the idea that *Trpm5* is needed for *Panx1*-dependent ATP release. In Figure 2G the *Panx1*-dependent currents were absent in *Trpm5*^{-/-} mice. Additionally, there was no ATP release after tuft cell stimulation observed in *Trpm5*^{-/-} mice in whole-cell patch clamp as well as in ELISA measurements.

10) Fig.S5 is bringing even more confusion. Denatonium-induced *I*_{sc} integrates responses from many channels and transporters, including *Trpm5* and *Panx1*. Probenecid will block *Panx1* current, and thus denatonium-induced *I*_{sc} will be smaller; this does not imply (and does not exclude either) a paracrine effect of released ATP on purinergic receptors. Similarly, the effect of suramin on denatonium-induced *I*_{sc} does not imply that extracellular ATP comes from *Panx1* channel. These data using whole-mount tracheal preparations, which are nice by the way, do not confirm that ATP is released via pannexin channel after TC stimulation, as stated in l.206-207. The data are consistent with the idea that ATP released via pannexins acts in a paracrine manner. I would suggest toning down some of the conclusions reported in this paragraph.

Reply: In our opinion, the recordings of the transepithelial ion current after tuft cell stimulation with denatonium strongly suggest that ATP is released from tuft cells via *Panx1*. The conclusion is supported by the observation that denatonium-induced *I*_{sc} changes are almost completely abolished in tracheae from *Panx1*^{-/-} mice and by suramin, a general inhibitor of receptors targeted by ATP (purinergic receptors). Furthermore, the experiments using probenecid to inhibit *Panx1* we observed a reduced current, which, in our opinion supports the results from *Panx1*^{-/-} mice. The specificity of this finding is underpinned by the experiments using ruthenium red, which show that CALHM1 channels, involved in ATP release from taste buds, do not contribute to the ATP-induced *I*_{sc} changes in the tracheal epithelium following tuft cell stimulation. Nevertheless, we tuned down the sentence in line 205-207 and it now reads: "Additionally, we further investigated tuft cell-dependent *Panx1* function and ATP signaling in the tracheal epithelium by recording currents after TC stimulation using whole-mount tracheal preparations".

11) Fig.S3: Panel B is not convincing at all! If the shadow fluorescent signal in WT is considered positive for ChAT, why it is not in *Chat*^{fl/fl}:*Trpm5*^{-/-} tracheas? I can see very similar background fluorescence.

Reply: While we appreciate the comment from the reviewer, we do not share the same opinion. The mouse strain that is mentioned by the reviewer is not *Chat^{fl/fl}:Trpm5^{-/-}* but *Chat^{fl/fl}:Trpm5^{Cre}*. In this mouse strain ChAT is depleted in Trpm5-expressing cells. In our opinion, we have clearly shown that there is no ChAT protein in the whole tracheal epithelium, as no bands was observed in homogenates from tracheal epithelium by Western blotting. Additionally, the immunofluorescence for ChAT is clearly overlapping with Trpm5 in wild-type tracheal epithelium. In contrast there is no signal for ChAT in the Trpm5+ immunoreactive cells in *ChAT^{fl/fl}:Trpm5^{Cre}* mice (arrowheads). The slight background fluorescence of the tissue is due the use of epifluorescence imaging, which was intentionally chosen to provide better orientation within the images. For better contextualization, we made changes to the figure legend. The figure legend now reads: “(A) Immunoblot with homogenates from whole tracheal epithelium from *ChAT^{fl/fl}:Trpm5^{Cre}* and wild-type mice (WT) for ChAT showed ChAT protein in wild-type mice at a size of approximately 72 kD, but not in *ChAT^{fl/fl}:Trpm5^{Cre}* mice.”

12) The gating strategy for flow cytometric (please correct spelling in legend, as well as in legends of Fig.S11 and S12) identification is shown in Fig.S8; it should be shown as Fig.S7 to be coherent with the Figure numbering in the text (Fig.S8 is mentioned after Fig.S6 and before Fig.S7).

Reply: We thank the reviewer for this comment. We have corrected all errors in the figure legends related to the gating strategies. We have also changed the orders of the figures to align coherently the text and switched the order of the Supplementary Figures S7 and S8.

13) Fig.3 and Fig.4: Please explain how the data are normalized as counts per “trachea” or per “lung”.

Reply: Depicting FACS values as total cell counts per organ as we have done in our study is well accepted and widely used (Izumi et al. 2021, PMID: 34413303, Nakano et al. 2013, PMID: 23168837, Zhao et al. 2006, PMID: 16861385). We are happy to explain how we evaluated the cell counts per lung or trachea. The numbers presented in Figure 3 and Figure 4 are absolute cell counts either per a trachea or per a lung. Tracheas of all mouse strains used in this study were dissected from the first cartilaginous tracheal ring until the last, at the level of the bifurcation. Since the mice used in our study were of comparable age, their tracheas and lungs had approximately the same size. Following digestion, the cell suspension was spun down and then resuspended in the same volume (200 µl). The cell counts per trachea or lung were determined for 20 µl cell suspension using an automated cell counter (NucleoCounter® NC-200™, Chemomatec, Kaiserslautern, Germany). The total cell count was then calculated for the whole volume, respectively whole trachea or lung. Please find an example of the total cell counts per trachea and lung in *Trpm5^{+/+}* and *Trpm5^{-/-}* mice below (Additional Figure 2). FACS analyses were subsequently used to determine the cell counts for the different immune cell populations which were then recalculated for the whole sample volume i.e. counts per trachea or lung.

Additional Figure 2: Example for total cell counts from tracheae (A) and lungs (B) from *Trpm5*^{+/+} and *Trpm5*^{-/-} mice.

14) Controls for Fig. 5J are missing.

Reply: We thank the reviewer for this comment. We have now compared the IL17A cytokine levels in *Panx1*^{+/+} and *Panx1*^{-/-} not only to each other but also to their corresponding untreated controls. We included these results in the revised version of Figure 5J. Please find the updated panel 5J below.

Fig. 5J: IL-17A levels in plasma samples of control (n=5) or denatonium-treated *Panx1*^{+/+} mice (n=5) or *Panx1*^{-/-} mice (n=3-4). One-way ANOVA; * p<0.05, ns: not significant.

15) Fig.S7: The effects of denatonium at 24h, 72h and 7d were compared to vehicle at 24h only. This is not correct and matched controls should be used. Please speculate on the mechanisms by which TCs activation by denatonium induces their intrinsic expansion within 24 h. DCAMKL1 and CD11c legends are shown in black then not visible on the images!

Reply: We thank the reviewer for this comment. We have now performed additional *in vivo* control experiments of wild-type mice treated with vehicle and evaluated the dendritic and tuft cell numbers after 72h and 7d. These results have been included in the revised version of Supplementary Figure S7. Please find the revised figure and

figure legend below. As visible from the figure, neither the tuft cell numbers (DCAMKL1 positive) nor the dendritic cell numbers (CD11c positive) changed after 72h or 7d and, thus, the treatment and our experimental design had no impact on the observed results. Vehicle-treated wild-type mice had significantly lower number of tuft cell or dendritic cell compared to denatonium-treated. In accordance with the '3R' principle for the protection and use of laboratory animals and especially the principles for reduction in the number of used animals, we refrained from performing vehicle-treatment experiments with *Trpm5*^{-/-} mice. Since there were no changes in the dendritic and tuft cell numbers in *Trpm5*^{-/-} mice after denatonium-treatment, and we did not observe any changes in vehicle-treated wild-type mice, we did not get a permission from the authorities for conduction of animal experiments for performing these experiments.

Considering findings from the literature, there might be two possible mechanisms underlying the observed tuft cell expansion in our study, which yet have to be elucidated. On the one hand the expansion of tuft cells after 24h might be regulated by CysLTs released from tuft cells, as previously suggested for tuft cell expansion in the airways after stimulation with aeroallergens (Bankova et al. 2018, PMID: 30291131). On the other hand the observed tuft cell expansion in our study might be similar to the gut. In the gut, IL-25 released from tuft cells regulates expansion indirectly by activating ILC2s, which produce IL-13. The IL-13 then acts on gut crypt epithelial progenitor cells (Von Moltke et al. 2016, PMID: 26675736). We agree with the reviewer, that it would be interesting to delineate the mechanism underlying the tuft cell expansion observed in our study. Indeed, we are currently addressing this issue in a separate study. Since the underlying mechanism does not appear to impact the findings of the current study, we would prefer to keep this information confidential and not include it into the manuscript.

We apologize for the black labeling of DCAMKL1 and CD11c and changed the color of the labeling in the revised version of the figure. Please find the revised version of supplementary Figure S8 (former Figure S7) below.

Fig. S8: Dendritic cell numbers correlate with tuft cell numbers. **(A)** Immunofluorescence staining for tuft cells (DCAMKL1) in tracheal cross sections and quantification of DCAMKL1⁺ cells in *Trpm5*^{+/+} and *Trpm5*^{-/-} mice treated with 1 mM denatonium after 24h, 72h, 7d or with vehicle (PBS) after 24h (*Trpm5*^{+/+} and *Trpm5*^{-/-} mice), 72h or 7d (*Trpm5*^{+/+} mice) (n=20 sections of 4 mice for each experimental condition). **(B)** Immunofluorescence staining of dendritic cells (CD11c) in tracheal cross sections and quantification of CD11c⁺ cells in *Trpm5*^{+/+} and *Trpm5*^{-/-} mice treated with 1 mM denatonium after 24h, 72, 7d or with vehicle (PBS) after 24h (*Trpm5*^{+/+} and *Trpm5*^{-/-} mice), 72h or 7d (*Trpm5*^{+/+} mice) (n=20 sections of 4 mice for each experimental condition). **(C)** Correlation analysis of tuft cell and dendritic cell numbers of *Trpm5*^{+/+} mice revealed a moderate positive linear correlation with Pearson's $r=0.4454$. **(D)** There was no correlation between tuft cell and dendritic cell numbers in *Trpm5*^{-/-} mice ($r=0.0432$). A, B: * $p<0.05$, ** $p<0.01$, *** $p<0.001$, one-way ANOVA.

16) Reply to Reviewer 3 regarding TRPM5-DTA mice. At least, the effect of stimulation of TRPM5-DTA tracheas with denatonium on ATP release should be shown.

Reply: We thank the reviewer for this valuable comment. We have now measured the ATP release after treatment of *Trpm5*-DTA tracheae with 1 mM denatonium and vehicle using the same ELISA kit employed for the ATP measurements in the samples from the other mouse strains used in this study. Denatonium stimulation failed to induce a release of ATP from *Trpm5*-DTA mouse tracheae, highlighting the significant role of tracheal tuft cells in ATP release following stimulation. The data have now been included in figure 1J. Please find the revised version of Figure 1J below.

Fig. 1: (J) Supernatants from denatonium-treated tracheae from *Panx1*^{-/-} mice (n=3) revealed the same ATP levels as vehicle-treated controls (n=3) quantified by ELISA assays. Tracheal supernatants from denatonium-treated *ChAT*^{fl/fl}; *Trpm5*^{cre} mice (n=3) contained higher ATP levels than vehicle treated tracheae (n=3). Supernatants from denatonium-treated tracheae from *Trpm5*-DTA mice (n=4) had the same ATP levels as vehicle-treated controls (n=4). Unpaired Student's *t*-test. * $p < 0.05$, ns: not significant.

17) Thank you for sharing additional experiments on DC proliferation. Since the representative images are not very clear and some wounds even enlarge over time, controls should be presented to support the migration data. Maybe a line in Methods can be added to exclude the possibility of DC proliferation. Please align the positioning of bottom legends (*Trpm5*^{+/+}...) in Fig. 7E to the corresponding conditions.

Reply: We thank the reviewer for this comment. We have now included representative images from the control experiments with all mouse strains used to study tuft cell-induced migration of dendritic cells. They can now be found in the Supplementary Figures S20 and S21 (please see below). Additionally, we added a description of the experiment that was performed to differentiate between proliferation and migration of dendritic cells in our migration assay in the final sentence of the supplementary methods. This now reads as following (page 18/ line 9-12): "To estimate the proliferation of the plated dendritic cells, the plated cells were first washed with DPBS (gibco) and then dissociated with 0.5 % trypsin (gibco). Cell counts and the percentages for their viability were then determined using NucleoCounter (Chemomatec)."

Additionally, we have also changed the positioning of the labelling (of the mouse strains) in Figure 7E. We hope that this is now more understandable for the readers.

Fig. 7: (E) Quantification of (D). DCs treated with supernatants from unstimulated tracheae of *Trpm5*^{+/+}, *Trpm5*-DREADD, *Trpm5*^{-/-} and *Trpm5*-DTA mice (control) as well as with supernatants from tracheae of *Trpm5*^{+/+} mice stimulated with 1 mM denatonium (den) and with den with apyrase (5 U/ml), from denatonium-treated *Trpm5*^{-/-} and *Trpm5*-DTA tracheae, and from *Trpm5*-DREADD tracheae treated with CNO (100 μ M) and with CNO and apyrase. Wound closure was evaluated 24 h after induction of the scratch (n=3). One-way ANOVA or unpaired Student's *t*-test. * p<0.05, *** p<0.001, ns: not significant.

Fig. S20: Migration assay with dendritic cells. Representative images of dendritic cells before (0 h) and after stimulation (24 h) with supernatants obtained from denatonium- (1 mM, SN den) or vehicle-treated (control) tracheae of *Trpm5*^{-/-} and *Trpm5*-DTA mice. Scale bar = 250 µm.

Fig. S21: Migration assay with dendritic cells. Representative images of dendritic cells before (0 h) and after stimulation (24 h) with supernatants obtained from CNO- (100 µM, SN CNO) or vehicle-treated or CNO and apyrase (5 U/ml) treated tracheae of *Trpm5*-DREADD mice.